

# Towards an understanding of the controls on $\delta O_2/N_2$ variability in ice core records

Romilly Harris Stuart[1], Amaëlle Landais[1], Laurent Arnaud[2], Christo Buizert[3], Emilie Capron[2], Marie Dumont[4], Quentin Libois[5], Robert Mulvaney[6], Anaïs Orsi[1,7], Ghislain Picard[2], Frédéric Prié[1], Jeffery Severinghaus[8], Barbara Stenni[9], and Patricia Martinerie[2]

[1]Laboratoire des Sciences du Climat et de l'Environnement, UMR8212, CNRS – Gif sur Yvette, France
[2]Université Grenoble Alpes, CNRS, INRAE, IRD, Grenoble INP, IGE, 38000 Grenoble, France
[3]College of Earth, Ocean, and Atmospheric Sciences, Oregon State University, Corvallis, OR 97331, USA
[4]Univ. Grenoble Alpes, Université de Toulouse, Météo-France, CNRS, CNRM, Centre d'Etudes de la Neige, 38000 Grenoble, France
[5]CNRM, Université de Toulouse, Météo-France, CNRS, Toulouse, France
[6]British Antarctic Survey, Natural Environment Research Council, Madingley Road, Cambridge CB3 0ET, UK
[7]The University of British Columbia, Department of Earth, Ocean and Atmospheric Sciences, Vancouver, Canada
[8]Scripps Institution of Oceanography, University of California, San Diego, La Jolla, CA 92093, USA
[9]Ca' Foscari University of Venice, Department of Environmental Sciences, Informatics and Statistics, Venezia, 30172, Italy

**Correspondence:** Romilly Harris Stuart (romilly.harris-stuart@lsce.ipsl.fr)

**Abstract.** Processes controlling pore closure are broadly understood yet defining the physical mechanisms controlling associated elemental fractionation remains ambiguous. Previous studies have shown that the pore closure process leads to a decrease in concentration of small-size molecules (e.g., $H_2$, $O_2$, Ar, Ne, He) in the trapped bubbles. Ice core $\delta(O_2/N_2)$ records – the ratio of $O_2$ to $N_2$ molecules in bubbles trapped in ice cores relative to the atmosphere – are therefore depleted owing to this $O_2$ loss

and show a clear link with local summer solstice insolation making it a useful dating tool. In this study, we compile $\delta(O_2/N_2)$ records from 14 polar ice cores and show a new link between $\delta(O_2/N_2)$ and local surface temperature and/or accumulation rate, in addition to the influence of the summer solstice insolation. We argue that both local climate-driven and insolation forcings are linked to the modulation of snow physical properties near the surface. Using the Crocus snowpack model, we perform sensitivity tests to identify the response of near-surface snow properties to changes in insolation, accumulation rate, and air

temperature. These tests support a mechanisms linked to snow grain size, such that the larger the grain size for a given density, the stronger the pore closure fractionation, and hence, lower $\delta(O_2/N_2)$ values. Our findings suggest that local accumulation rate and temperature should be considered when interpreting $\delta(O_2/N_2)$ as an insolation proxy.

## 1  Introduction

Ice cores store crucial information for our understanding of past climate variability and atmospheric composition. Interpreting

ice core gas records first requires an understanding of the evolution of snow into ice via the firnification processes. Firn is the name given to the layer of unconsolidated snow which makes up the top 50-120 m of ice sheets. Atmospheric air moves through porous networks within the firn until a critical depth where vertical diffusion effectively stops, and the pores gradually



become sealed off from the atmosphere to form bubbles trapped within the ice. This depth is called the lock-in depth (LID) and is largely controlled by local accumulation rate, temperature, and possibly the degree of density layering (Schwander et al., 1997; Martinerie et al., 1994; Mitchell et al., 2015).

Measurements of entrapped air can be used to reconstruct past atmospheric compositions, as well as to date the ice cores. One such dating technique, used primarily for deep ice cores from low accumulation sites is orbital dating, which uses insolation curves at a given latitude directly calculated from astronomical variables (Laskar, J. et al., 2004). Records of total air content (Raynaud et al., 2007), $\delta^{18}O$ of atmospheric $O_2$ (Extier et al., 2018), and $\delta(O_2/N_2)$ (Kawamura et al., 2007; Suwa and Bender, 2008a; Landais et al., 2012; Bouchet et al., 2023) were found to be strongly anti-correlated with insolation curves, and thus, allow for ice core dating using peak matching techniques. The term $\delta(O_2/N_2)$ – hereafter, simply $\delta O_2/N_2$ - describes the relative difference between the ratio of $O_2$ to $N_2$ molecules trapped within the ice and that of the standard atmosphere and is expressed in the delta notation commonly used for stable isotope ratios.

The use of $\delta O_2/N_2$ for ice core dating was first proposed by Bender (2002) after observations of an anti-correlation with local summer solstice insolation (hereafter SSI). Data from the Vostok ice core showed that high SSI corresponds to low $\delta O_2/N_2$ values and vice versa (Bender et al., 1994). A similar relationship was then observed at numerous other sites such as Dome Fuji (Kawamura et al., 2007) and EPICA Dome C (Landais et al., 2012) in Antarctica, and GISP2 (Suwa and Bender, 2008b) in Greenland. Over orbital timescales, the correlation between $\delta O_2/N_2$ and local SSI largely improved when matched on the ice-age timescale rather than the gas-age (Bender, 2002), suggesting that the impact of the insolation on the ice properties occurs at the surface rather than the pore closure depth.

Parallel firn air studies of the open porosity revealed an enrichment in $O_2$ and other small molecules, such as Ar, Ne and He, compared to air within the closed porosity at the close-off depth (COD), providing further evidence of size-dependent fractionation during pore-closure (Battle et al., 1996; Huber et al., 2006; Severinghaus and Battle, 2006). While the physical mechanisms controlling the amount of fractionation are not fully understood, it is believed that the main processes by which smaller molecules escape during pore closure are; 1) molecular diffusion through the ice lattice, or permeation, resulting from pressure gradients between recently closed pores and neighbouring open pores (Tomoko Ikeda-Fukazawa and Hondoh, 2004; Huber et al., 2006; Severinghaus and Battle, 2006), and 2) diffusion through small openings in the ice matrix, with a threshold of 3.6 Å, allowing only molecules with a diameter below 3.6 Å to pass through (Huber et al., 2006; Severinghaus and Battle, 2006). Both processes are facilitated by the pore network's capacity to export the fugitive gases back to the atmosphere, which is required for the observed depletion in bulk ice $O_2$ in bubbles (Fujita et al., 2009).

The simultaneous observations of $O_2$ depletion in entrapped air bubbles and $O_2$ enrichment in open porosity, alongside the strong correlation between SSI and $\delta O_2/N_2$ on the ice-age scale, led Bender (2002) to develop the hypothesis that the dependence of $\delta O_2/N_2$ on local insolation was the result of snow metamorphism near the surface. They, and many subsequent studies, proposed that strong summer insolation drives temperature gradient metamorphism, thus increasing near-surface grain size which propagates through the firn during the firnification process down to the COD (Bender, 2002; Severinghaus and Battle, 2006; Suwa and Bender, 2008a; Fujita et al., 2009; Hutterli et al., 2009). In addition, Fujita et al. (2009) proposed that $\delta O_2/N_2$ would be decreased under high SSI conditions due to enhanced density stratification in the deep firn. They argue that



'summer' layers – characterised by large grains and relatively low density in the deep firn – close-off deeper and take longer to do so than neighbouring 'winter' layers, which are denser and have smaller grains (Picard et al., 2012, 2016). 'Summer' layers
therefore remain permeable for longer, allowing the $O_2$ enriched air in open porosity to be exported to the atmosphere, and hence, reducing bulk ice $\delta O_2/N_2$ under high SSI conditions (Fujita et al., 2009). While the proposed mechanisms are posited to explain the SSI imprint on $\delta O_2/N_2$, they are also influenced by local climate conditions such as temperature, accumulation rate and wind-speed. Indeed, there is a substantial amount of evidence linking local climate conditions with both firn physical properties (McDowell et al., 2020; Casado et al., 2021; Inoue et al., 2023) and deep firn layering (Hörhold et al., 2011).

There is a growing body of evidence for a local climatic imprint on ice core $\delta O_2/N_2$ records. Firstly, spectral analysis has revealed climate related 100-ka cyclicity at EPICA Dome C (Bazin et al., 2016) - although such a signal is not apparent at Dome F (Kawamura et al., 2007). Secondly, millennial scale variability in $\delta O_2/N_2$ records from GISP2 appeared in-phase with local temperature fluctuations driven by Dansgaard-Oeschger events (Suwa and Bender, 2008b). In parallel, Kobashi et al. (2015) evidenced an anti-phase effect of accumulation rate on $\delta Ar/N_2$ records at GISP2 over the last millennia. Like $O_2$, Ar
is a smaller molecule than $N_2$ so that the same driving mechanisms are invoked for the $\delta Ar/N_2$ and $\delta O_2/N_2$ variations, but the $\delta Ar/N_2$ anomaly tends to be half as large as the $\delta O_2/N_2$ anomaly (Bender et al., 1995; Buizert et al., 2023). Kobashi et al. (2015) proposed a direct effect of accumulation rate or temperature on the $\delta Ar/N_2$ ($\delta O_2/N_2$) variations through the firn depth. The higher the accumulation rate or the lower the temperature, the higher the firn weight and hence the overloading pressures in microbubbles preferentially expelling $O_2$ and Ar in the LIZ. In contrast, Severinghaus and Battle (2006) proposed that the
higher the accumulation rate, the more rapid the burial of bubbles, allowing less time for gases to escape during pore closure.

Using a combination of data and snowpack modelling, we aim to develop our understanding of the formation of the of $\delta O_2/N_2$ records, by first, determining the role of local climate parameters, accumulation rate and temperature, on $\delta O_2/N_2$ variability, and second identifying potential mechanisms related to snow physical properties using snowpack sensitivity tests. We use a compilation of datasets from 14 ice cores from both Antarctica and Greenland to identify spatial and temporal patterns
in $\delta O_2/N_2$ depending on local surface conditions. The impacts of SSI and local climate on snow properties are then assessed using the SURFEX-ISBA-Crocus detailed snowpack model (Vionnet et al., 2012). We aim to constrain the influence of near-surface snow properties on $\delta O_2/N_2$ variability, potentially contributing to a mechanistic explanation for elemental fractionation during pore closure.

## 2 Methods

### 80 2.1 Ice core sites

We compiled $\delta O_2/N_2$ from 18 ice cores from Antarctica and Greenland but use data from 14 of those sites for reasons explained in Section 2.4. Previously published data were measured on ice cores from: Dome Fuji (DF), EPICA Dome C (EDC), Greenland Ice Core Project 2 (GISP2), Law Dome DE08 and DSSW20k, North Greenland Ice core Project (NGRIP), Roosevelt Island Climate Evolution (RICE), Siple Dome (SD), South Pole (SP), Talos Dome (TALDICE), Vostok (VK), and the West Antarctic
Ice Sheet Divide (WAISD) ice cores (references for all datasets are presented in Table S2). We also present unpublished data





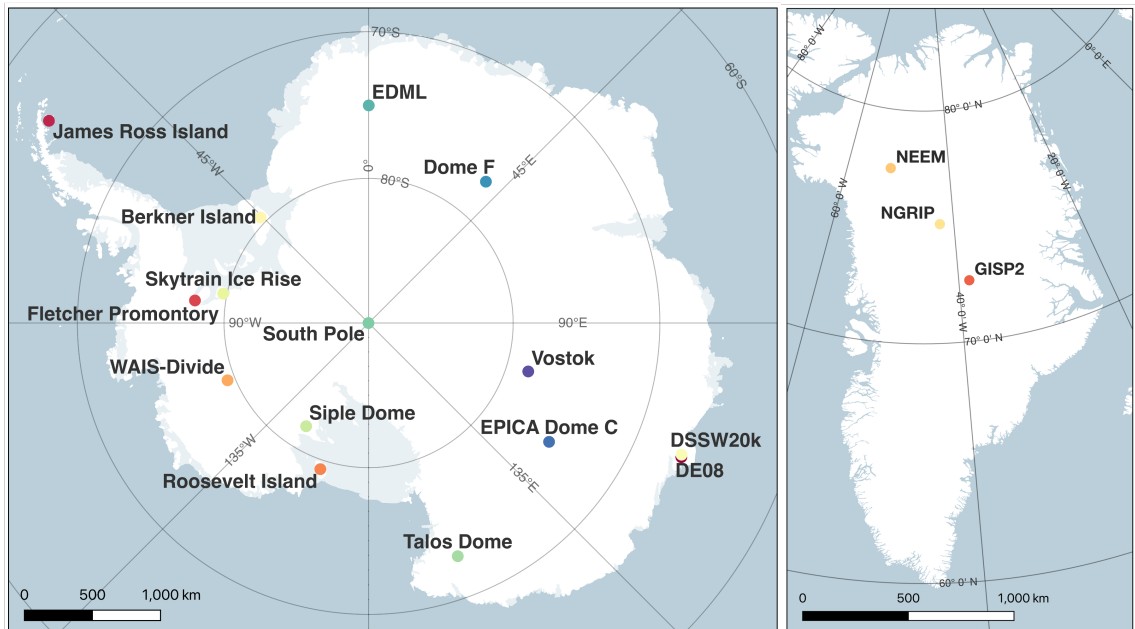

**Figure 1.** Locations of each ice core site initially included in our study. Maps were made in Quantarctica and QGreenland (Matsuoka et al., 2018; Moon et al., 2023).

from Berkner Island (BI), EPICA Dronning Maud Land (EDML), Fletcher Promontory (FP), GISP2, James Ross Island (JRI), North Greenland Eemian Ice Drilling (NEEM), Skytrain Ice Rise (SIR) and Talos Dome (TALDICE) ice samples (Table S1). Table 1 provides an overview of the site characteristics.

## 2.2 Analytical techniques for previously unpublished data

The previously unpublished $\delta O_2/N_2$ datasets were measured at the Laboratoire des Sciences du Climat et de l'Environnement (LSCE), with the addition of some GISP2 data measured at Scripps Institution of Oceanography (Scripps). At LSCE, two techniques have been used to extract the trapped air from ice core samples and have been described in previous studies (Landais et al., 2003; Capron et al., 2013; Bazin et al., 2016). In short, the first uses a melt-refreeze technique where ice samples are placed into glass flasks at -20°C while the atmospheric air is evacuated. Samples are then left to slowly melt to release the

trapped gases, before the samples are refrozen with liquid nitrogen. The extracted air samples are then individually introduced into the line and passed through a $CO_2$ and water vapour trap, before being trapped in the stainless-steel dip-tube submerged in liquid helium (Landais et al., 2003). The second uses a semi-automated extraction (melt-extraction) line which removes the need for refreezing of the samples (Capron et al., 2010; Bazin et al., 2016). For all datasets measured at LSCE, the average analytical uncertainty for $\delta O_2/N_2$ is 0.5‰.

Unpublished data from the GISP2 core were measured at the Ice Core Noble Gas Laboratory of the Scripps Institution of Oceanography using the melt and refreeze technique (Sowers et al., 1989; Petrenko et al., 2006). Additional information on the



**Table 1.** Overview of ice core site characteristics. The final three columns show the average annual accumulation rate, the average annual air temperature, and the average surface snow density at each site for present-day conditions.

| Site | Latitude | Longitude | Elevation | Brittle zone | Accumulation rate | Temperature |
|---|---|---|---|---|---|---|
| | (°N) | (°E) | (m) | (m) | (cm w.eq. a$^{-1}$) | (°C) |
| **BI** | -79.55 | -45.68 | 890 | 450-940[1] | 16.2[1] | -26[1] |
| **DF** | -77.32 | 39.7 | 3810 | 450-1200 | 2.6[2] | -58[3] |
| **EDC** | -75.1 | 123.35 | 3233 | 600-1200 | 2.8[4] | -55[5] |
| **EDML** | -75 | 0.04 | 2892 | 500-1050 | 6.4[6] | -45[7] |
| **FP** | -77.9 | -82.61 | 873 | -[8] | 38[8] | -27[8] |
| **GISP2** | 72.6 | -38.5 | 3200 | 650-1400 | 24[9] | -31.4[10] |
| **JRI** | -64.2 | -57.69 | 1542 | -[8] | 58[11] | -14[11] |
| **LD DE08** | -66.72 | 113.2 | 1250 | 525-1200 | 110[12] | -19[12] |
| **LD DSSW20k** | -66.77 | 112.81 | 1370 | | 15[13] | -20.7[14] |
| **NEEM** | 77.45 | -51.6 | 2450 | 609-1281 | 20[15] | -28.9[15] |
| **NGRIP** | 75.1 | -42.32 | 3090 | 790-1200 | 17.5[16] | -31.5[16] |
| **RICE** | -79.36 | -161.71 | 550 | 650-1300 | 21[17] | -23.5[18] |
| **SD** | -81.65 | -148.81 | 621 | 400-1000 | 12.4[19] | -25.4[20] |
| **SIR** | -79.74 | -78.55 | 784 | -[21] | 13.5[22] | -26[22] |
| **SP** | -89.99 | -98.16 | 2835 | 619-1078 | 8[23] | -49[24] |
| **TALDICE** | -72.82 | 159.07 | 2315 | 667-1002 | 8[25] | -41[25] |
| **VK** | -78.47 | 106.87 | 3488 | 300-720 | 2.2[26] | -57[26] |
| **WAISD** | -79.47 | -112.09 | 1766 | 650-1300 | 20.2[27] | -31.1[27] |

[1]Mulvaney et al., 2007; [2]Oyabu et al., 2023; [3]Fujita et al., 1998; [4]Frezzotti et al., 2004; [5]Stenni et al., 2004; [6]Oerter et al., 2000; [7]EPICA community members, 2006; [8]Mulvaney et al., 2014; [9]Alley et al., 1993; [10]Alley and Koci, 1988; [11]Capron et al., 2013; [12]Etheridge and Wookey, 1988; [13]Sturrock et al., 2002; [14]Rubino et al., 2013; [15]Buizert et al., 2012; [16]NGRIP project members, 2004; [17]Winstrup et al., 2019; [18]Bertler et al., 2018; [19]Severinghaus et al., 2001; [20]Hamilton, 2002; [21]Mulvaney et al., 2021;[22]Hoffmann et al., 2022; [23]Lazzara et al., 2012; [24]Mosley-Thompson et al., 1999; [25]Stenni et al., 2002; [26]Arnaud et al., 2000; [27]Fegyveresi et al., 2011. All brittle zones are those presented by Neff (2014) unless otherwise stated (and references therein).

GISP2 measurements is available in Martin et al. (2023) where they use the $\delta^{15}$N of N$_2$ from the same samples. An overview of all datasets, both previously published and unpublished, can be found in Table S1 and Table S2, respectively.

### 2.2.1 New $\delta$O$_2$/N$_2$ measurements

*Berkner Island*

Measurements were performed on bubbly ice from the Berkner Island ice core every 55 cm (every bag) between 631 m



and 680 m, corresponding to 10,269 – 21,350 yr BP (Capron et al., 2013). Replicate samples were prepared at LSCE using the melt-refreeze method and measured on a 10-collector Thermo Delta V Plus between March 2010 and March 2011.

*EDML*

Nine samples were measured on bubbly ice from the EDML ice core over five depth levels between 328-473 m (327.8 m (4.51 ka BP), 354.2 m (4.95 ka BP), 381 m (5.43 ka BP), 467 m (7.04 ka BP), and 473 m (7.16 ka BP)) (Bazin et al., 2013). The samples were prepared using the melt-refreeze method and measured on the 10-collector Thermo Delta V Plus at LSCE. Where possible, the final value of each sample is the average of two replicate measurements at each depth

level.

*Fletcher Promontory*

In January 2015, 39 depth levels were measured from the Fletcher Promontory ice core, retrieved in 2012. Measurements were performed approximately every 3 m starting at 289 m down to 388 m. There is currently no published age-scale for the FP ice core. All samples were prepared using the melt extraction method and then measured using a 10-collector

Thermo Delta V Plus as LSCE. The final value of each sample is the average of at least two replicate measurements at each depth level.

*GISP2*

New GISP2 $\delta O_2/N_2$ data were analysed at Scripps Institution of Oceanography from 643 depths between 1740 and 2400 m (13 to 50 ka BP), with the $\delta^{15}N$ data from these analyses reported previously (Martin et al., 2023). Most of

the samples were analysed in replicate. Measurements were performed in several campaigns between 2017-2020 and referenced to La Jolla pier air. Measurements were performed using a melt-refreeze method (Petrenko et al., 2006).

*James Ross Island*

Between February and March 2011, measurements were performed at 16 depth levels on the James Ross Island Ice core. The depth resolution varied between 2-50 m starting at 52 m until 363 m. Samples were prepared using the melt extraction

method and then measured using a 10-collector Thermo Delta V Plus as LSCE. The final value of each sample is the average of at least two replicate measurements at each depth level.

*NEEM*

Clathrate ice from the NEEM ice core was measured between February and April 2011, a year after the core was retrieved. A total of 119 depth levels were sampled were measured at varying resolutions over the following intervals:

55 cm intervals (every bag) between 1757-1773 m (38.127 – 39.735 ka B2k), 5.5 m intervals (every 10 bags) between 2205 and 2370 m (108.56 - 120.237 ka B2k), every 2 bags from 2375-2434 m (no published age-scale available below these depths), and 5.5 m intervals (every 10 bags) between 2436 and 2519 m (Gkinis et al., 2021; Rasmussen et al., 2013). In total, samples from 119 depth levels were prepared using the melt-refreeze method and measured using a 10-collector





Thermo Delta V Plus as LSCE. The final value of each sample is the average of at least two replicate measurements at
each depth level.

*Skytrain Ice Rise*

Measurements were performed on bubbly ice from the Skytrain Ice Rise ice core between March and April 2021. Samples were taken sporadically (1-15 m intervals) at 16 depth levels between 307 and 436 m depth (4.707-11.696 ka BP) (Mulvaney et al., 2023). Each sample was prepared at LSCE using the melt extraction method and subsequently mea-
sured on a 10-collector Thermo Delta V Plus. The final value of each sample is the average of at least two replicate measurements at each of the 16 depth levels.

*TALDICE*

Numerous measurements have been done on bubbly and clathrate ice from TALDICE between 2008 and 2022 at LSCE. A total of 308 depth levels were measured at varying intervals starting at 155 m down to 1617 m. Published age-scales
reach 1548 m, giving an age range of 1.55-343 ka for TALDICE samples (Buiron et al., 2011; Crotti et al., 2021). All samples were prepared using the melt extraction technique and measured on the 10-collector Thermo Delta V Plus. Some measurements between 1356-1620 m depth have been published previously and are available in Crotti et al. (2021).

### 2.2.2    Corrections

Chemical and pressure imbalance corrections are applied to the measurements during data processing (Landais et al., 2003).
In addition, all data are corrected for gravitational fractionation in the firn using $\delta^{15}N$ of $N_2$ from the same samples.

$$\delta O_2/N_2 \, grav = \delta O_2/N_2 - 4 \cdot \delta^{15}N \tag{1}$$

Gas loss effects during coring and ice core storage are well documented to modify $\delta O_2/N_2$, causing significant depletion in $O_2$ in clathrate ice stored above -50°C (Ikeda-Fukazawa et al., 2005; Kawamura et al., 2007; Landais et al., 2012). Ikeda-Fukazawa et al. (2005) proposed an equation to correct for gas loss effects during storage at different temperatures. However,
given the incomplete storage history for all ice cores we do not attempt to correct for storage gas loss, but rather define rejection criteria outlined in Section 2.2.3.

### 2.2.3    Data rejection criteria

At the transition zone between bubbly ice and clathrate ice (hereafter the brittle zone) strong elemental fractionation occurs whereby the air in the gas phase has a very different composition to that in the clathrate hydrates, thus making the interpretation
of gas measurements unreliable at these depths (Bender, 2002). Measurements from brittle ice tend to be characterised as having increased mean $\delta O_2/N_2$ (usually in excess of 0‰) and strong data scattering, expressed as a high standard deviation. All measurements from Berkner Island fall within the reported brittle zone but show no scattering, and are therefore included in our analysis. This is also observed at other sites – where some depths defined as the brittle zone show no evidence of scattering – and is expected to be due to the approximation of the brittle zone depths. To avoid adding biases to our analysis, measurements



from brittle ice from all other sites are removed, leaving the measurements from bubbly ice above, and the clathrate ice below the brittle zone. Additional scattering in elemental ratios, characterised by a standard deviation of 6.2‰ compared to 1.8‰ in non-brittle ice, is observed below the brittle zone in the WAIS-Divide record between 1300-1500 m (Shackleton, 2019). Similar effects were observed on the EDC and TALDICE ice cores (Lüthi et al., 2010) and the Dome Fuji ice core (Oyabu et al., 2021). Data influenced by this scattering effect was also removed from our analysis.

The different storage histories of ice used for each measurement campaign need to be considered before interpreting the data to account for gas loss effects (Section 2.2.2). Successive $\delta O_2/N_2$ measurements from TALDICE and GISP2 clathrate ice samples show strong depletion of $O_2$ through time (Supplement S1), which is consistent with observations from Dome C (Bouchet et al., 2023). Having separated the datasets into bubbly ice and clathrate ice – by taking the data above and below the brittle zone – we systematically reject measurements from clathrate ice stored at -20°C for over 3 years, or at -36°C for more

than 4 years. Bubbly ice stored at these same temperatures appear largely unaffected by gas loss (Supplement S1), with the exception of Vostok (Suwa and Bender, 2008a). Applying these criteria to the datasets results in the removal of all data from NGRIP and Vostok, as well as sections of data from other sites. The remaining 14 datasets are presented in Table 3 and were used to analyse the drivers of $\delta O_2/N_2$ variability.

## 2.3   Modelling near-surface snow properties

The second component of our study addresses the modelled response of snow physical properties to perturbations in SSI, accumulation rate, and temperature with the aim of identifying which properties may be influencing elemental fractionation during pore closure. We use the SURFEX-ISBA-Crocus detailed snowpack model (Crocus hereafter) to simulate snowpack evolution (Vionnet et al., 2012). Crocus simulates changes in snow physical properties induced by surface metamorphism and the evolution of these properties with depth. The model is forced by ERA5 reanalysis data (Hersbach et al., 2020), and

the snowpack is initialised with measurements of snow density, effective optical radius of snow grains and snow temperature. Optical radius is defined as the radius which snow grains would have for their surface area-to-volume ratio if they were spherical (Domine et al., 2006). Optical radius is thus directly linked to specific surface area (SSA), defined as the surface area of snow at the ice-air interface per unit mass (units $m^2 kg^{-1}$) (Legagneux et al., 2002), via the following equation:

$$SSA = \frac{3}{r_{opt} \cdot \rho_{ice}} \tag{2}$$

Where $r_{opt}$ is the optical radius and $\rho_{ice}$ is the density of ice (Gallet et al., 2014). We use this model to assess changes in snow physical properties near the surface which are invoked to explain $\delta O_2/N_2$ variability. Dome C is used as the test site given the abundance of snowpack observations as well as high resolution $\delta O_2/N_2$ data.

### 2.3.1   Crocus model description

Crocus is a 1-dimensional model which simulates the evolution of snow properties with time and depth on a layer-by-layer

basis, i.e., in a Lagrangian framework (Vionnet et al., 2012). A detailed description of the model can be found in Vionnet et al. (2012). Briefly, the initial number of layers is defined by the user, with the thickness of each layer allowed to change along





the simulation (layer thickness ranging from millimetres to metres thick). The maximum number of layers available in the model was increased from 50 to 80 to account for the higher number of thin layers forming at Dome C than at Alpine sites. Once the simulated snowpack consists of 80 layers, the aggregation scheme merges internal neighbouring layers with similar

properties allowing a new surface layer to form. The key physical processes incorporated into Crocus for dry snow conditions are accumulation of snowfall, snow metamorphism, compaction of snow by the wind, compaction due to the weight of the overlaying layers, absorption of solar radiation, heat diffusion, and surface energy budget.

For our study, two fundamental user-defined model components are the snow metamorphism and radiative transfer schemes. We use the semi-empirical model from Flanner and Zender (2006) (F06) for the metamorphism scheme which describes

the evolution of optical radius with time. F06 was found to be the most appropriate formulation for Dome C conditions (Carmagnola et al., 2014; Libois et al., 2014). To successfully reproduce the snow temperature profile – vital for realistically simulating snow metamorphism – the Two-streAm Radiative TransfEr in Snow model (TARTES) is used to account for vertical distribution of absorbed solar radiation in the snowpack (Libois et al., 2013). TARTES also considers the effect of impurities on snow temperature via albedo. For Dome C, we include black carbon content which is set to $3\,\mathrm{ng\,g^{-1}}$ (Warren et al., 2006;

Libois et al., 2015). In this study, we assess the simulated snow density, snow temperature and snow SSA from Crocus model outputs.

### 2.3.2 Dome C specific Crocus configuration

Crocus was initially developed for alpine or sub-polar regions with seasonal snowpacks. Libois et al. (2014) modified multiple components of the Crocus model to improve its suitability to high latitude sites with low accumulation rates - specifically for

Dome C. The modifications are extensively described in Libois et al. (2014) and were implemented into the current version of Crocus for this study. The changes are as follows:

1. *Fresh snow properties*: The parameterisation of fresh snow density is based on temperature and wind-speed which results in an unrealistically low density for Dome C conditions ($50\,\mathrm{kg\,m^{-3}}$). Fresh snow density is fixed to a minimum of $170\,\mathrm{kg\,m^{-3}}$ - the lowest fifth percentile from Dome C observations (Libois et al., 2014). Similarly, fresh snow SSA is

set to $100\,\mathrm{m^2\,kg^{-1}}$ instead of $65\,\mathrm{m^2\,kg^{-1}}$ used in the standard version of the model (Grenfell et al., 1994; Libois et al., 2014).

2. *Wind-induced compaction*: At low-accumulation sites, the snow can remain at the surface for prolonged periods of time. The long exposure time to surface winds facilitates compaction, and hence increases density. The maximum surface snow density is increased from $350\,\mathrm{kg\,m^{-3}}$ to $450\,\mathrm{kg\,m^{-3}}$ to account for this effect (Albert et al., 2004; Libois et al.,

230 2014).

3. *Aggregation scheme*: The formation of a new snow layer requires a minimum amount of snowfall. Due to the low accumulation rate at Dome C, the amount of snowfall needed to form a new layer was decreased from $0.03\,\mathrm{mm\,h^{-1}}$ to $0.003\,\mathrm{mm\,h^{-1}}$. In the instance when the snowpack has the maximum number of layers (80) at the time a new snow layer is formed, layers with similar properties will be aggregated, resulting in a smoothed signal. The aggregation scheme



was disabled for the top 6 layers to resolve realistic near-surface snow temperature profiles and gradients, required to accurately simulate snow metamorphism.

### 2.3.3  Model initialisation

The snowpack was initialised with density and optical radius profiles measured in January 2010 at Dome C down to 20 m (Champollion et al., 2019), and snow temperature data from a probe installed at Dome C in 2012 with 5 cm resolution near

the surface, coarsening with depth down to 12 m. ERA5 reanalysis data from Dome C was used to force the model at 3-hourly resolution over the period between 1st January 2000 and 1st December 2020 (Hersbach et al., 2020). The model requires atmospheric forcings for air temperature, accumulation rate, wind speed and direction, incoming shortwave and longwave radiation, and specific humidity. ERA5 gives a mean annual snowfall rate between 2000 and 2020 of $2.3\,\mathrm{cm\,w.eq.\,a^{-1}}$, and as such, the snowfall rate was multiplied by 1.2 to match the observed mean annual accumulation rate of around $2.8\,\mathrm{cm\,w.eq.\,a^{-1}}$

(Frezzotti et al., 2004; Libois et al., 2014). To ensure that at least the top 1 m consists of accumulated snow, a 100-year spin up was used by running the forcing file 10 times between 2000 and 2010, followed by the period from 2000 to 2020. The outputs from 2010 to 2020 were then used for analysis.

### 2.3.4  Sensitivity tests

The sensitivity of snowpack properties to perturbations in surface forcings are tested by modifying one of three forcing pa-

rameters: incoming shortwave radiation, accumulation rate, or 2 m air temperature. The magnitude of the perturbation to each parameter correspond to minimum and maximum values reconstructed over the last 800 ka. We use shortwave radiation as a proxy of insolation and scale the values in proportion to the SSI values. A total of seven simulations are used to perform sensitivity analysis and are outlined in Table 2. The model configuration and initial snow profile were kept constant for each simulation; only the tested parameter in the atmospheric forcing file was modified as follows:

**Table 2.** Overview of modifications made to forcing test parameter in Crocus snowpack sensitivity test scenarios.

| Simulation | | Reference | **SSI min** | **SSI max** | **A min** | **A max** | **T min** | **T max** |
|---|---|---|---|---|---|---|---|---|
| Incoming SWR | (Scaled) | 100% | **85%** | **111%** | 100% | 100% | 100% | 100% |
| Accumulation rate | $(\mathrm{cm\,w.eq.\,a^{-1}})$ | 2.8 | 2.8 | 2.8 | **1.0** | **4.1** | 2.8 | 2.8 |
| Air temperature | (°C) | -55 | -55 | -55 | -55 | -55 | **-65** | **-51** |

*Summer solstice insolation (SSI)*: Over the last 1000 years, the average SSI at 75.1°S was $544\,\mathrm{W\,m^{-2}}$ (Laskar, J. et al., 2004), compared to $462\,\mathrm{W\,m^{-2}}$ and $601\,\mathrm{W\,m^{-2}}$, corresponding to minimum and maximum SSI over the past 800 ka. To translate to forcing perturbations, the incoming shortwave radiation (SWR) is scaled by 85% and 111%, respectively to reach the target values ($462\,\mathrm{W\,m^{-2}}$ and $601\,\mathrm{W\,m^{-2}}$). No additional modifications are applied to annual distribution of SWR.

*Annual mean accumulation rate (A)*: Present-day accumulation rate at Dome C is set to $2.8\,\mathrm{cm\,w.eq.\,a^{-1}}$ in the ERA5 forcing.

Hereafter, accumulation rate is expressed as ice equivalent centimetres per year. ERA5 snowfall was scaled by 36%





to reach the target accumulation rate of $1.0\,\mathrm{cm\,w.eq.\,a^{-1}}$, representing the 800 ka minimum, and 146% to produce an accumulation rate of $4.1\,\mathrm{cm\,w.eq.\,a^{-1}}$ which corresponds to the 800 ka maximum (Bazin et al., 2013).

*Annual mean air temperature (T)*: Snowpack sensitivity to air temperature is tested by decreasing the 2 m air temperature by 10°C for glacial conditions (Jouzel et al., 2007), and applying a 4°C, increase to represent 800 ka maximum tempera-
tures. We note, however, that borehole temperature measurements and delta-age are more consistent with a 5°C cooling (Buizert, 2021). Furthermore, these temperature modifications do not include changes in seasonal temperature variability but suffices for the purpose of identifying bulk changes in the snow properties. The average seasonal cycle is kept constant with an average amplitude of 35°C.

It is important to highlight that at polar sites accumulation rate is dependent on temperature, and temperature is influenced
by insolation, such that these parameters are not independent. However, we use the model to constrain the influences of each forcing parameter in an independent manner to understand the mechanisms, even if, in reality, these parameters are inter-dependent.

## 3 Results

### 3.1 Influence of SSI and local climate on $\delta O_2/N_2$ variability in ice cores

Figure 2 shows $\delta O_2/N_2$ versus SSI for EDC, Dome F and South Pole. These three sites are used owing to their long temporal range and high-resolution $\delta O_2/N_2$ measurements, but with negligible gas loss. The regression slopes vary between $-0.09 \pm 0.006\,‰.\mathrm{m}^2.\mathrm{W}^{-1}$ for South Pole to $-0.06 \pm 0.005\,‰.\mathrm{m}^2.\mathrm{W}^{-1}$ for EDC. The regression for Dome F falls within 2 standard deviations ($2\sigma$) of the regression for EDC, but the regression for South Pole falls just outside the $2\sigma$ uncertainty. Furthermore, the $\delta O_2/N_2$ is shifted for South Pole compared to EDC, evidenced by the increased y-intercept from 18.9‰ at
EDC to 40.6‰ at South Pole. Inter-site differences in the dependence of absolute $\delta O_2/N_2$ on SSI suggest additional factors are influencing the records, such as accumulation rate – which at South Pole is over double that of both EDC and Dome F – or integrated summer insolation. In the following sections, we provide evidence for the influences of accumulation rate and air temperature on $\delta O_2/N_2$, in addition to SSI, using both spatial (inter-site) and temporal (EDC ice core) variability of $\delta O_2/N_2$.

### 3.1.1 Inter-site comparison of mean $\delta O_2/N_2$

In addition to the $\delta O_2/N_2$ datasets, we compile accumulation rate and temperature reconstructions from each site covering the same depth range as the $\delta O_2/N_2$ data. Due to the inclusion of measurements from varying depths, and thus, ages, between sites, we include the range of accumulation rates and temperatures to indicate the climate histories for each site (Table 3). The data are presented in Figure 3 for a) 1000 year averaged SSI, b) accumulation rate, and c) temperature. Error bars in panel a) and b) indicate the range of values with the exceptions of; Berkner Island and James Ross Island, where the minimum, maximum,
and mean accumulation rate and temperature values are approximated as last glacial maximum values (Capron et al., 2013), the present day values, and the mean of these two values, respectively; and, Law Dome sites DE08 and DSSW20k, where





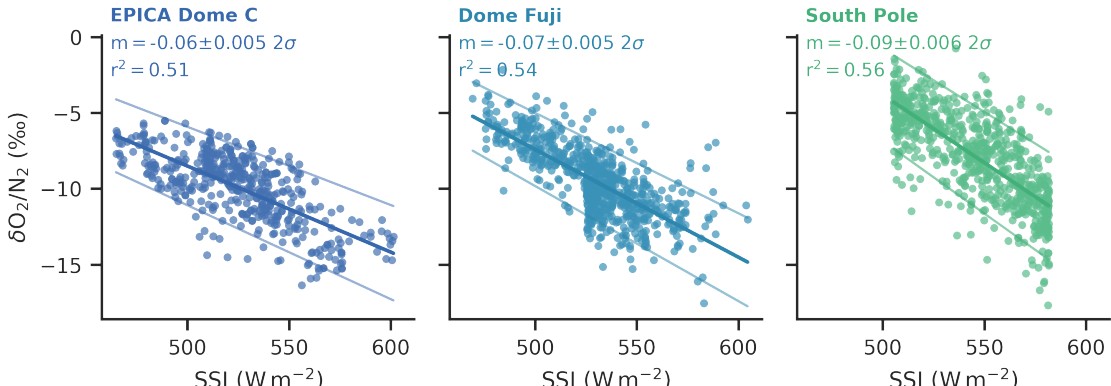

**Figure 2.** Scatter plots showing the negative correlation between SSI and $\delta O_2/N_2$. Significant negative correlations (over 99% confidence) are observed using high resolution data from Dome C (dark-blue; Bouchet et al., 2023), Dome F (mid-blue; Kawamura et al., 2007; Oyabu et al., 2021), and South Pole (green; Severinghaus et al., 2019). The slope (m, units $‰.m^2.W^{-1}$) and $r^2$ values are presented for each site.

only the present-day values are used. We acknowledge that the former approximation may introduce a bias towards cold, low-accumulation conditions.

As expected, mean $\delta O_2/N_2$ is anti-correlated with SSI (r=-0.47, p<0.1) with a slope of -0.07, the same as observed in Figure 2. However, in addition to the influence of SSI, we find a stronger, more significant correlation between mean $\delta O_2/N_2$ and both temperature (r=0.72, p<0.001) and the natural log of accumulation rate (r=0.81, p<0.001) – the logarithmic dependence suggesting increased sensitivity at low accumulation rates. The linear model in Figure 2b indicates that a doubling of accumulation rate would result in a 1.5‰ increase in $\delta O_2/N_2$ ($\delta O_2/N_2 = 2.2 \cdot log(A) - 12$). However, it is important to note that temperature and the logarithm of accumulation rate are strongly correlated in Antarctica, such that the correlations seen in
panels (a) and (b) of Figure 3 are dependent on one another.

Large residuals in Figure 2a can partially be attributed to the use of 1000-year average SSI instead of averages over the same time periods as the $\delta O_2/N_2$ data. This would require age-scales for all sites. Deviations from the $\delta O_2/N_2$-SSI regression line may also be attributed to discrepancies in site latitude resulting from the ice flow speed at different sites. Indeed, data from NEEM were measured on ice between 1757 and 2525 m depth (approximately 38-130 ka), when the ice would have been
upstream of the current site at a lower latitude (Rasmussen et al., 2013; Members, 2013). Berkner Island, NEEM and Siple Dome fall below the regression lines in panels b) and c). For Berkner Island, this may be linked to the method used to determine the mean accumulation rate and temperature values. Other explanations may be linked to procedural artefacts such as gas loss during storage.

### 3.1.2    Temporal variability of $\delta O_2/N_2$ at Dome C

High resolution data is required to investigate the temporal variability in $\delta O_2/N_2$ as a function of accumulation rate, temperature, and SSI. The $\delta$-deuterium ($\delta D$) record from water isotope measurements is used as a qualitative proxy for accumulation





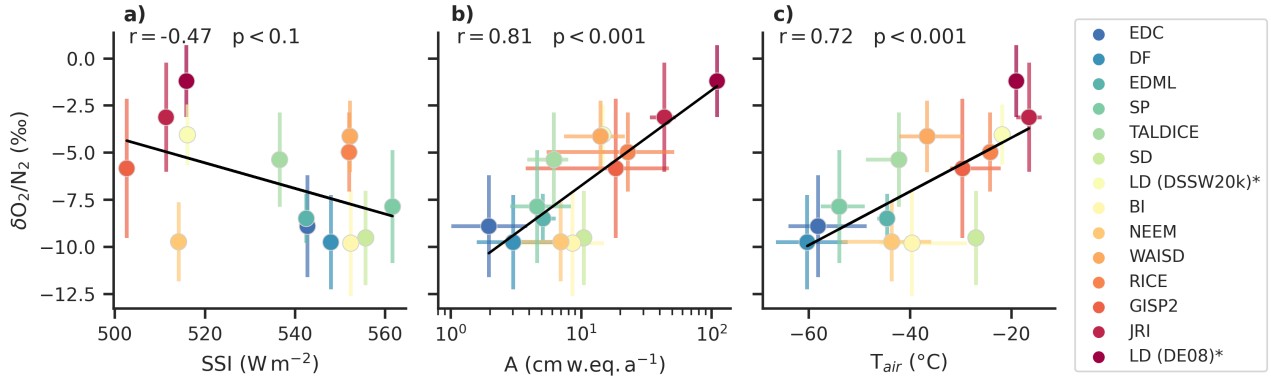

**Figure 3.** Scatterplots showing the dependence of $\delta O_2/N_2$ on the a) SSI, b) accumulation rate (A), and c) annual average temperature ($T_{air}$). Each point represents the mean values for each site over the depth interval of included $\delta O_2/N_2$ data for each site (Table 3). Error bars represent the range of values over the depth interval. Error bars on y-axis show the standard deviation of $\delta O_2/N_2$ measurements, and the x-axis in b) and c) show the range in accumulation rate and temperature. In each panel, the black line shows the linear regression between $\delta O_2/N_2$ and each parameter, along with the associated correlation coefficient (r) and p-value. Data shown here are presented in Table 3, while information on the individual datasets can be found in Table S1 and S2 in the supplement. *Present-day accumulation rate and temperature have been used or the two Law Dome sites; DE08 and DSSW20k.

rate and temperature, whereby higher $\delta D$ values are generally associated with increased accumulation rate and temperature in ice cores from the East Antarctic plateau (Jouzel et al., 2007; Parrenin et al., 2007). The following analysis uses high res-olution $\delta O_2/N_2$ and $\delta D$ measurements from the EPICA Dome C (EDC) ice core (Jouzel et al., 2007; Bouchet et al., 2023),

both on the AICC2012 ice-age scale (Bazin et al., 2013). $\delta O_2/N_2$ records used for orbital dating require the filtering of noise and millennial-scale variability (Kawamura et al., 2007; Landais et al., 2012). However, no filtering is applied here in order to resolve variability over shorter timescales. The new measurements cover five distinct sections of the core between 111 and 539 ka BP (Bouchet et al., 2023). We primarily focus on the longest section between 180-259 ka BP (1980-2350 m) which covers MIS 7.

In Figure 4, the $\delta O_2/N_2$ curve is dominated by the SSI cyclicity, as has been documented previously (e.g., Landais et al., 2012). Superimposed onto this signal are millennial-scale peaks in $\delta O_2/N_2$ which appear to coincide with peaks in $\delta D$, high-lighted by grey bars in Figure 4c. To evaluate additional climatic signals in the EDC $\delta O_2/N_2$ records, we first remove the SSI signal from the record using the residuals of the linear regression between $\delta O_2/N_2$ and SSI (Figure 2b). We observe that, in addition to the millennial-scale variability, some long-term variability remains which would not be expected if $\delta O_2/N_2$ was

controlled by SSI alone. Indeed, there is a stronger significant positive correlation between $\delta D$ and SSI-$\delta O_2/N_2$ residuals on the ice-age scale between 190-260 ka BP (Figure 4d; r=0.68, p<0.001), than $\delta D$ and $\delta O_2/N_2$ (Figure 4e; r=0.43, p<0.001). This suggests a positive correlation with accumulation rate and temperature and shares analogy with the spatial positive correlation between $\delta O_2/N_2$ and both accumulation rate and temperature (Figure 3). Applying the same analysis to the other periods of





**Table 3.** Information on $\delta O_2/N_2$ datasets from each site after removing measurements influenced by gas loss. Presented are; the $\delta O_2/N_2$ mean ($\mu$), standard deviation ($\sigma$), and number of $\delta O_2/N_2$ measurements depths (N); the depth range of $\delta O_2/N_2$ measurements; the mean ($\mu$), minimum (min) and maximum (max) accumulation rate (A), and the mean ($\mu$), minimum (min) and maximum (max) temperature (T) over the same depth range as the $\delta O_2/N_2$ data; and site latitude. The depth ranges have been rounded.

| Site | $\delta O_2/N_2$ (‰) | | | Depth (m) | A (cm w.eq. a$^{-1}$) | | | T (°C) | | | Lat (°N) |
|---|---|---|---|---|---|---|---|---|---|---|---|
| | $\mu$ | $\sigma$ | N | | $\mu$ | min | max | $\mu$ | min | max | |
| *BI[1] | -9.81 | 2.8 | 64 | 609-694 | 8.6 | 2.3 | 14.9 | -39.6 | -46.0 | -28.7 | -79.6 |
| DF[2] | -9.76 | 2.5 | 913 | 113-449, 1410-2500 | 3.0 | 1.6 | 7.2 | -60.3 | -66.5 | -52.3 | -77.3 |
| EDC[3] | -8.91 | 2.7 | 823 | 1421-3189 | 2.0 | 1.0 | 3.9 | -58.2 | -64.0 | -48.5 | -75.1 |
| EDML[4] | -8.5 | 1.3 | 5 | 596-860 | 5.1 | 5.1 | 6.4 | -44.5 | -46.5 | -43.7 | -75.0 |
| GISP2[5] | -5.84 | 3.7 | 182 | 73-648, 1515-2428 | 18.4 | 3.8 | 47.0 | -29.6 | -32.1 | -22.1 | 72.58 |
| *JRI[6] | -3.13 | 2.9 | 16 | 52-364 | 43.5 | 33.8 | 53.3 | -16.5 | -19.0 | -14.0 | -64.2 |
| **DE08[7] | -1.21 | 1.9 | 8 | 175-218 | 110 | - | - | -19.0 | - | - | -66.72 |
| **DSSW20k[8] | -4.06 | 1.6 | 4 | 61-63 | 14.7 | - | - | -21.8 | - | - | -66.77 |
| NEEM[9] | -9.74 | 2.1 | 119 | 1757-2525 | 7.0 | 2.6 | 10.5 | -43.6 | -53.5 | -35.8 | 77.42 |
| RICE[10] | -4.98 | 2.1 | 387 | 60-344 | 22.8 | 5.4 | 51.6 | -24.3 | - | - | -79.36 |
| SD[11] | -9.53 | 2.5 | 68 | 69-400 | 10.5 | 9.9 | 11.0 | -27.0 | -28.2 | -26.1 | -81.65 |
| SP[12] | -7.87 | 3 | 691 | 125-617, 1078-1751 | 4.6 | 2.9 | 98.3 | -54.0 | -57.7 | -48.9 | -89.99 |
| TALDICE[13] | -5.38 | 2.5 | 68 | 155-669, 1003-1402 | 6.2 | 3.9 | 8.0 | -42.2 | -48.7 | -41.1 | -72.82 |
| WAISD[14] | -4.14 | 1.9 | 433 | 80-648, 1602-3397 | 14.1 | 7.4 | 21.7 | -36.6 | -42.1 | -29.5 | -79.47 |

References for $\delta O_2/N_2$ data, accumulation rate, and temperature correspond to a, b, and c, respectively. [1]This study[a], Capron et al., 2013[b,c]; [2]Kawamura et al., 2007[a], Oyabu et al., 2021[a], Watanabe et al., 1999[b], Uemura et al., 2018[c]; [3]Bouchet et al., 2023[a], Extier et al., 2018[a], Bazin et al., 2013[b], Jouzel et al., 2007[c]; [4]This study[a], Bazin et al., 2013[b], Stenni et al., 2010[c]; [5]This study[a], Suwa and Bender, 2008b[a], Cuffey and Clow, 1999[b], Clow, 1999[c]; [6]This study[a], Capron et al., 2013[b,c]; [7]Buizert et al., 2020[a]; Rubino et al., 2013[b], Etheridge and Wookey, 1988[c]; [8]Buizert et al., 2020[a], Rubino et al., 2013[b], Morgan et al., 1997[c]; [9]This study[a], Rasmussen et al. (2013)[b,c]; [10]Lee et al., 2020[a], Winstrup et al., 2019[b], Bertler et al., 2017[c]; [11]Severinghaus, 2009[a], Buizert, 2021[b,c]; [12]Severinghaus, 2019[a], Kahle et al., 2020[b,c]; [13]This study[a], Bazin et al., 2013[b,c]; [14]Severinghaus, 2015[a], Fudge et al., 2017[b], White et al., 2019[c].

*JRI and BI using LGM and present day values (Capron et al., 2013) as the maximum and minimum.

** Present-day A and T are used for both Law Dome sites.

high-resolution measurements results in significant positive correlations for all but the period between 409 and 449 ka BP
(Figure S3 in Supplement).





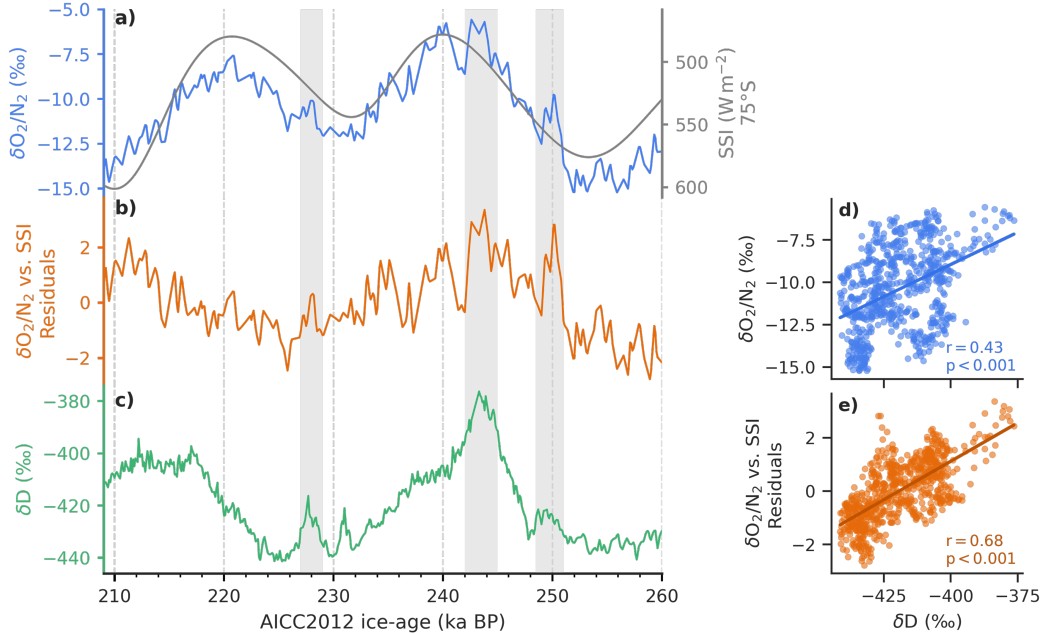

**Figure 4.** Evolution of $\delta O_2/N_2$, SSI and $\delta D$ on the AICC2012 ice age timescale from Dome C (Bazin et al., 2016; Bouchet et al., 2023). Panel a) presents the $\delta O_2/N_2$ (blue) with SSI on the right y-axis (grey), b) the $\delta O_2/N_2$-SSI residuals (orange), and c) $\delta D$ (green), over the period between 210 and 260 ka BP. Correlations between $\delta D$ and $\delta O_2/N_2$, and between $\delta D$ and the $\delta O_2/N_2$-SSI residuals are presented in d) and e), respectively.

## 3.2 Crocus model results

### 3.2.1 Crocus model evaluation for Dome C

The Crocus model outputs are first evaluated by comparing the reference simulation (Ref in Table 2) to observational data from Dome C. Simulated density and SSA profiles are compared to data from Libois et al. (2014), measured daily between
$23^{rd}$ November 2012 and $16^{th}$ January 2013 at two sites within 600 m of Concordia station, Dome C. Density was measured at 2.5 cm resolution down to 25 cm, while SSA was measured at 1 cm depth intervals down to 50 cm. Snow temperature at Dome C has been continuously measured since 2012 at 30-minute intervals. Simulated snow density and SSA were interpolated to a fixed grid of 1 cm depth resolution and a 24-hour timestep between 2010-01-01 and 2020-12-01, while snow temperature was interpolated onto a 1 mm by 6-hourly grid over the same time period.
Density and SSA outputs presented in Figure 5a and b are averaged values over the measurement period ($23^{rd}$ November 2012 and $16^{th}$ January 2013). For the most part, the observations fall within the range of the simulations. The simulated SSA profile is consistently within one standard deviation of the measurements below 10 cm, above this depth, simulated SSA is overestimated by up to $10 \, m^2 \, kg^{-1}$. In contrast, density is well-simulated throughout the top 25 cm. Small standard deviations





associated with simulated density and SSA, suggests that variability is not well reproduced. As discussed in Libois et al. (2014),
the standard version of Crocus is unable to reproduce density and SSA variability with depth due to its one-dimensional nature.
For the purpose of our study, we consider the standard version sufficient to assess the overall sensitivity of snowpack properties
to perturbations in forcings.

Snow temperature in Figure 5c covers the period between $23^{rd}$ November 2019 and $16^{th}$ January 2020. A 3°C cold bias
is apparent, but the mean falls within $1\sigma$ of the observations. Figure 5d presents distributions of the stacked January snow
temperatures between 2016 and 2020, further showing this 3°C cold bias during summer (-37.1±3.6°C and -34.4±4.1°C for
Crocus outputs and observations, respectively). However, stacked July snow temperatures are well simulated with a 1 m mean
of -64.7±4.5°C and -63.9±5°C from Crocus outputs and observations, respectively. The overestimation of SSA in the top
10 cm may be linked to the summer cold bias in near-surface snow temperature, reducing the rate of snow metamorphism in
near-surface grains.

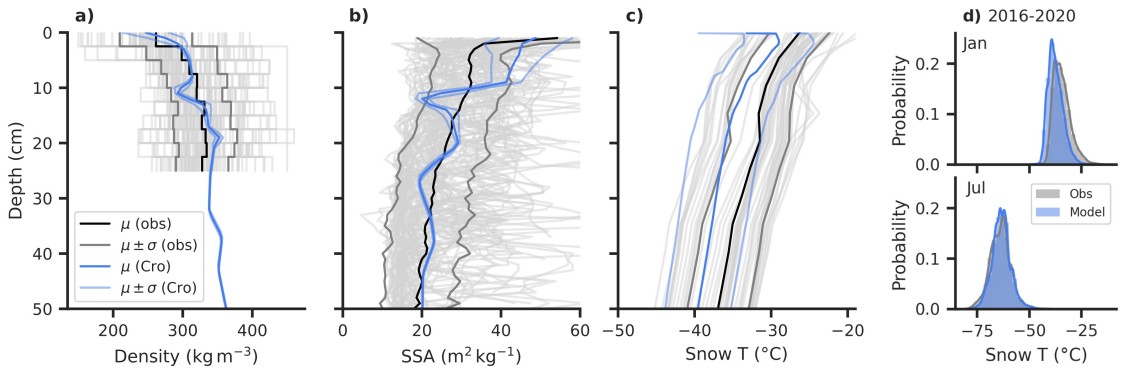

**Figure 5.** Comparison of observations and Crocus simulated snowpack profiles. SSA (a) and density (b) profiles represent the average from
$23^{rd}$ November 2012 and $16^{th}$ January 2013, to cover the measurement period (Libois et al., 2014). The snow temperature (c) profiles both
from Crocus and the observations are the average from $23^{rd}$ November 2019 and $16^{th}$ January 2020. Faded grey lines represent individual
profiles from observations, and the shaded bands show the standard deviations. Snow temperature distributions (d) from January and July
between 2016 and 2020.

### 3.2.2 Simulated response of snowpack properties to surface perturbations

Sensitivity tests were run using the Crocus model to assess the response of near-surface snowpack properties to perturbations
in local surface forcings. The following analysis uses optical radius as a measure of grain size – which is directly linked to
SSA and the density of ice (Section 2.3.1) – and focuses on the response of near-surface snow density and grain size to the six
scenarios outlined in Table 2. We firstly assess the bulk changes in physical properties before looking at the variability with
depth.



### 3.2.3 Bulk snowpack sensitivity

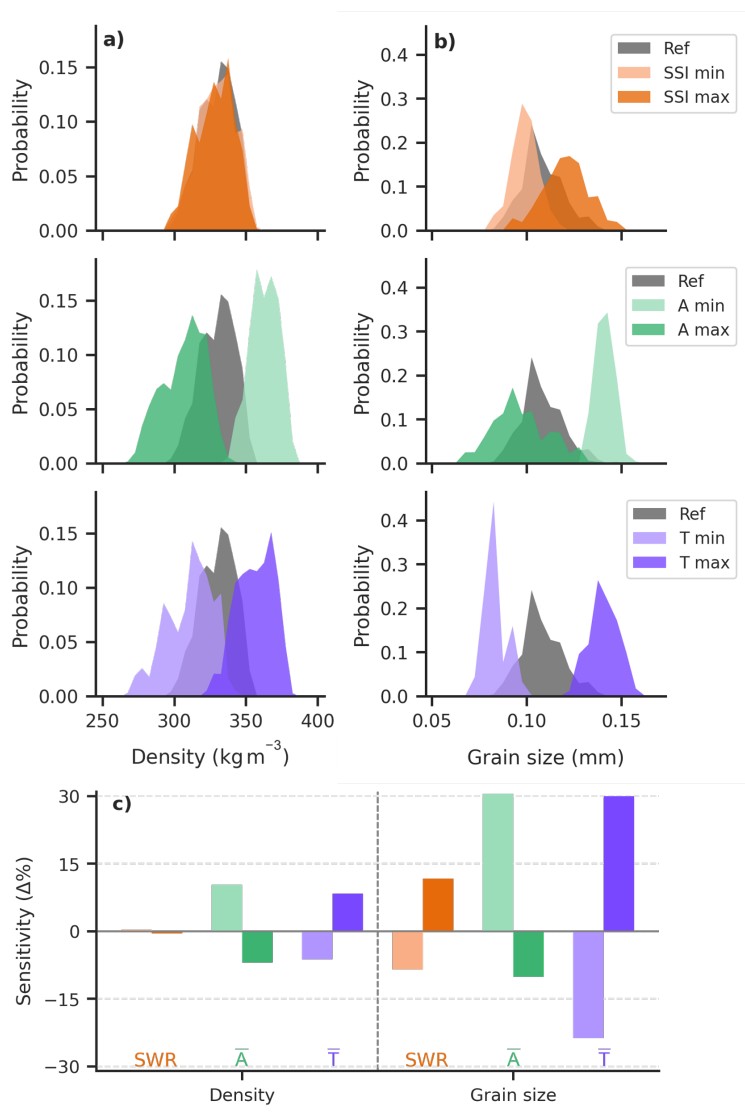

**Figure 6.** Comparison of density and grain size ($r_{opt}$) over the top 20 cm from Crocus sensitivity simulations. Distributions of a) density and b) grain size outputs from each test simulation are compared the reference simulation (bin size is 5 kg m$^{-3}$, and 0.005 mm). In panel c) bars represent the percentage change in mean density and mean grain size for perturbations in SSI (orange), accumulation rate (green), and temperature (purple); with the decreased scenarios represented by the faded colour, and the increased by the bold colour.

Numerous studies have suggested that modifications in near-surface density and grain size are key parameters influencing elemental fractionation during pore closure (e.g., Bender, 2002; Fujita et al., 2009). Figure 6c and 6d show the mean difference in density and grain size from the Dome C reference simulation (Ref) and each of the test scenarios (outlined in Table 2).





Overall, grain size is more sensitive to changes in surface forcing than density in the upper 20 cm. The sensitivity tests reveal that a 15% decrease in SSI (SSI min) deceases grain size by 8% and an 11% increase in SSI (SSI max) causes a 12% increase in grain size. Both directions of perturbation result in <1% change in density. The magnitudes of change in both density and grain size are much larger under accumulation rate and temperature perturbations than under SSI perturbations.

Accumulation rate and temperature have the opposite effect on density and grain size. A 4°C increase in temperature (T max)
increases density and grain size by 8% and 29%, respectively, while an increasing the accumulation rate to 4.1 cm w.eq. a$^{-1}$ (A max) results in a 7% decrease in density and a 10% decrease in grain size. These opposing influences of accumulation rate and temperature on snow properties at first appears to contradict the observations in Figure 3, whereby $\delta O_2/N_2$ increases with both accumulation rate and temperature. This can be attributed to co-linearity between accumulation rate and temperature. We also note the non-linear response of density and grain size to perturbations in all forcings – most evident is the sensitivity of grain
size to accumulation rate. This is in line with the dependence of $\delta O_2/N_2$ to the logarithm of accumulation rate documented in Figure 3.

### 3.2.4 Depth variability sensitivity

An alternative (although possibly complimentary) explanation for the mechanistic control of snow properties on elemental fractionation links to stratigraphic layering due to seasonality (Fujita et al., 2009). Here we explore the influence of layering
by looking at the depth variability in density and grain size as a qualitative measure of stratification (Hörhold et al., 2011),

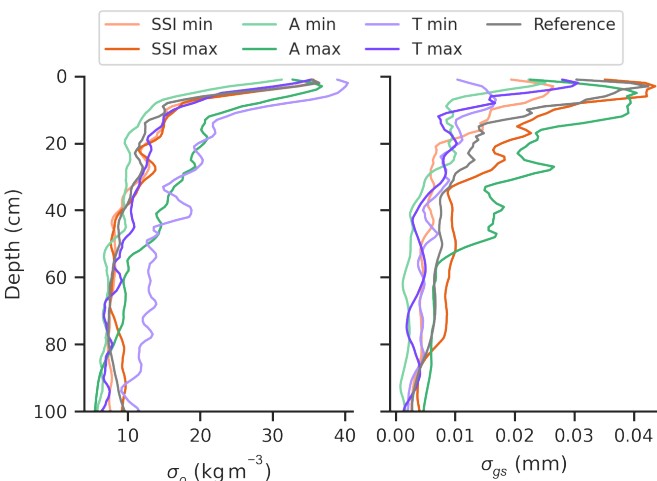

**Figure 7.** Variability in density and grain size with depth over the top 1 m. Each pair of simulations is represented by a colour, SSI in orange, accumulation rate in green, and temperature in purple. The faded line of each pair represents the 800-ka minimum simulation, and the bold line represents the 800-ka maximum simulation.



assuming that higher variability indicates stronger layering. Variability is defined as the standard deviation of each depth interval (denoted $\sigma$) over the period between January 1st 2010 and December 1st 2020.

The $\sigma$ values for each simulation are presented in Figure 7 for the top 1 m of snowpack. In all runs, $\sigma$ peaks near the surface and decreases with depth for both density and grain size ($\sigma_\rho$ and $\sigma_{gs}$). Density variability for four out of the six test simulations is largely similar to the reference. High-accumulation rate (A max) and low-temperature (T min) are the exception, with an increase in $\sigma_\rho$ of up to $5\,\mathrm{kg\,m^{-3}}$ compared to all other runs. Surprisingly, the increased $\sigma_\rho$ values correspond to a decrease in mean density in the A max and T min simulations (Figure 6). The spread in $\sigma_{gs}$ between simulations is broader than for $\sigma_\rho$ over the top 50 cm. , with T min, SSI min, A min, and T max all resulting in reduced variability compared to the reference run. $\sigma_{gs}$ appears to increase with SSI and accumulation rate throughout the top metre.

To summarise, outputs from the Crocus model indicate that insolation modifies mean grain size but has negligible effect on mean density in the top 20 cm. On the other hand, higher air temperature and lower accumulation rate result in increases in both mean density and mean grain size, although this change is not necessarily proportional. The depth variability of the two snowpack parameters is strongly influenced by increased accumulation rate, with decreased temperature also having a large effect on density variability, and increased SSI causing an increase in grain size variability. At 100 cm, there is no significant difference in the $\sigma_\rho$ and $\sigma_{gs}$ for the different forcing scenarios, which is discussed in Section 4.3.2.

## 4 Discussion

### 4.1 Evidence for non-SSI dependence of $\delta O_2/N_2$

Our compilation of different deep ice core $\delta O_2/N_2$ records show the widely documented anti-correlation between SSI and $\delta O_2/N_2$ on the ice-age scale (Kawamura et al., 2007; Landais et al., 2012; Suwa and Bender, 2008a; Oyabu et al., 2021; Extier et al., 2018; Bouchet et al., 2023). However, a comparison of EDC, Dome F, and South Pole (as the only 3 cores with sufficient long and high resolution $\delta O_2/N_2$ records not affected by gas loss for this study) reveals an additional latitudinal dependence of the slope of the linear regression, whereby the highest latitudes have the steepest slope (SP 90°S = -0.09‰.m$^2$.W$^{-1}$, DF 79°S = -0.06‰.m$^2$.W$^{-1}$, EDC 75°S = -0.05‰.m$^2$.W$^{-1}$; Figure 2). This suggests an additional influence of site conditions such as integrated summer insolation (ISI) (Huybers and Denton, 2008), which is not surprising given the mechanistic overlap with total air content (TAC) variability (Fujita et al., 2009). Inter-site analysis reveals a dependence of mean $\delta O_2/N_2$ on SSI (average over the last 1000 years) with both temporal (Figure 2) and spatial (Figure 3c) data falling on the same regression slope (mean -0.07‰.m$^2$.W$^{-1}$). We also observe new evidence that mean $\delta O_2/N_2$ is significantly dependent on local temperature and accumulation rate, most apparent in the dependence of mean $\delta O_2/N_2$ on the natural log of accumulation rate (Figure 3b).

Further investigation into drivers of $\delta O_2/N_2$ variability at Dome C revealed a significant dependence on $\delta D$ (a proxy for accumulation rate and temperature on the long-time scale considered here). Between 190 and 260 ka BP at Dome C, the residuals of the $\delta O_2/N_2$-SSI regression are strongly correlated with $\delta D$ (r=0.62, p<0.001), both on the AICC2012 ice-age scale. This relationship is maintained in four out of five sections of high-resolution measurements from the EDC core (Figure S3), with the age-range of the outlying section (409–449 ka BP) coinciding with onset of MIS 11 (424–374 ka BP).



The temporal analysis from Dome C initially contradicts the findings from Kobashi et al. (2015), who show an anti-

correlation between accumulation rate and $\delta Ar/N_2$ (and thus, $\delta O_2/N_2$) at GISP2, Greenland. However, this anti-correlation

is observed when $\delta Ar/N_2$ is drawn on a gas-age scale while our observations are performed on $\delta O_2/N_2$ drawn on the ice-age

scale. Moreover, (Suwa and Bender, 2008b) found a positive correlation between millennial-scale $\delta O_2/N_2$ variability and lo-

cal temperature at GISP2. This apparent contrast in the link between $\delta O_2/N_2$ (or $\delta Ar/N_2$) and accumulation rate variability

is also reflected in TAC records. Superimposed to orbital variations driven by ISI (Lipenkov et al., 2011), TAC also exhibits

shorter, millennial-scale variations which are correlated with accumulation rate at SP, Antarctica (Epifanio et al., 2023) but

anti-correlated at NGRIP, Greenland (Eicher et al., 2016) – all considered on the gas-age scale. The observations from Kobashi

et al. (2015) and Eicher et al. (2016) suggest additional mechanisms may be at play at high accumulation sites in Greenland,

which are evident in the $\delta O_2/N_2$ and TAC records over millennial timescales. Our overarching aim is to improve the use of

$\delta O_2/N_2$ as a dating tool, and therefore, we focus on the mechanisms controlling $\delta O_2/N_2$ variability at Antarctica sites - or

rather, low accumulation sites - where accumulation increase leads to an increase in $\delta O_2/N_2$.

## 4.2 Proposed mechanisms controlling $\delta O_2/N_2$ variability

Several mechanisms have been proposed to explain variations in $\delta O_2/N_2$ and TAC and could be considered in light of our

new findings; namely, 1) the effect of residence time in the LIZ, specifically referring to the time taken for pores to close-off

(Severinghaus and Battle, 2006; Lipenkov et al., 2011), 2) transient effects from rapid climatic changes driving variations in

overburden pressure on the closing bubbles (Kobashi et al., 2015; Eicher et al., 2016), and 3) effects of near-surface snow

properties and layering which persist throughout the firn (Bender, 2002; Fujita et al., 2009; Severinghaus and Battle, 2006;

Gregory et al., 2014).

1) Severinghaus and Battle (2006) stated that at low accumulation sites, pores will close-off deeper and take longer to fully

close, thus, experiencing more total gas loss and, presumably, more elemental fractionation. Moreover, low accumulation sites

are usually characterised by a thin lock-in zone which facilitates diffusivity and fugitive gas loss, as is evidenced by young

gas ages in deep firn open porosity (Landais, 2004; Witrant et al., 2012). It is plausible that this 'residence time' effect could

partially explain our observations of a positive correlation between accumulation rate and $\delta O_2/N_2$. However, we would expect

this signal to be imprinted on the gas-age scale – or more likely, an integrated signal over the entire firn column – and as such,

this effect alone cannot account for the much stronger correlation between accumulation rate/temperature and $\delta O_2/N_2$ on the

ice-age scale at Dome C.

2) Short-term (millennial-scale) variations of $\delta O_2/N_2$ ($\delta Ar/N_2$) and TAC with accumulation rate have been linked to transient

effects in the firn column, resulting from rapid climatic changes (Kobashi et al., 2015; Eicher et al., 2016). During the initial

stage of a Dansgaard-Oeschger event, rapid increases in accumulation rate cause overburden pressure to increase, while more

time is needed for the firn column temperature to respond (Eicher et al., 2016). Eicher et al. (2016) suggest that these transient

effects explain the anti-correlation observed between accumulation rate and TAC on the gas-age scale at high-accumulation

sites. On the other hand, Kobashi et al. (2015) propose an alternative microbubble mechanism to explain the $\delta Ar/N_2$ response to

rapid climatic change. Such effects are expected to be negligible at low-accumulation sites such as Dome C where accumulation





variations are gradual. Moreover, the associated direction of change in $\delta O_2/N_2$ during high accumulation conditions is opposite to our observations, potentially indicating that the dominant mechanisms vary between low- and high-accumulation sites, as
suggested by Epifanio et al. (2023) with regard to TAC.

3) Near-surface snow properties and layer stratification have been invoked to modulate $\delta O_2/N_2$ variability since the first study of Bender (2002). The signal of near surface snow metamorphism - imprinted in the grain properties - is generally understood to persist throughout the densification process to the close-off depth, influencing pore closure processes (Bender, 2002). Kawamura et al. (2007) also suggested a link between the strength of metamorphism at the surface and the existence of
the lock-in zone leading to $\delta O_2/N_2$ fractionation. In parallel, Raynaud et al. (2007) proposed that the metamorphism induced by stronger insolation accelerates grain growth at the surface which later controls porosity in the lock in zone. Fujita et al. (2009) were the first to focus explicitly on the physical mechanisms governing both elemental fractionation during pore closure and total air content. They provided microstructural data – such as grain size, anisotropy, and a qualitative measure of permeability – from a Dome Fuji firn core and suggested a link between SSI and deep firn density stratification and permeability. They
proposed that gas transport via permeation would be more efficient when there is a strong density stratification; such that, increased SSI leads to bulk ice with lower $\delta O_2/N_2$ - which agrees with what we observe. However, these studies do not explicitly explore the link between local climate (accumulation rate and temperature) and $\delta O_2/N_2$ variability.

### 4.3 Towards a mechanistic understanding of $\delta O_2/N_2$ variability at low-accumulation sites

While a link between climate and snow metamorphism has been evidenced in previous studies - whereby snow metamorphism
is enhanced during summers with very low accumulation rates (Picard et al., 2012; Casado et al., 2021) – the link to pore closure processes has received less attention. Our analysis aims to bridge this gap by focusing on both local climate parameters and SSI, their influence on near-surface snow metamorphism, and how this might modulate elemental fractionation during pore closure. Crocus snowpack sensitivity tests were used in this study as a first step towards understanding a mechanistic link between physical properties and $\delta O_2/N_2$ variability. Our findings support a density-dependent grain size mechanism linking
snow metamorphism and elemental fractionation at close-off. Indeed, Calonne et al. (2022) showed that grain size has a strong influence on permeability, such that for a given density, permeability is increased with grain size. Moreover, Gregory et al. (2014) found that permeability is increased in high-density, large-grain size firn due to a less complex pore structure. The following sections utilise the qualitative differences in bulk snow properties and stratification obtained from the Crocus sensitivity tests to develop a mechanistic explanation for the role of snow properties on pore closure fractionation.

### 4.3.1 The role of bulk snow properties

SSI sensitivity tests (S min and S max in Table 2) show an increase in grain size under increased SSI, attributed to both increases in near-surface temperature gradients (Appendix B in Vionnet et al., 2012) and increased snow temperature (Figure S4 in supplement) during summer. The absence of a density response to perturbations in SSI forcing is unclear. In reality, negligible change, or even a decrease, in near-surface density may be attributed to mass loss via sublimation (Inoue et al.,



2023). However, vapour transport is not modelled in Crocus and it is therefore more likely that this artifact is linked to the compaction scheme.

The sensitivity of bulk grain size to accumulation rate can be linked to the indirect effect of the residence time of a snow layer in the upper centimetres of the snowpack, where temperature gradients are strongest (Vionnet et al., 2012; Picard et al., 2012). Shorter residence time with increased accumulation rate (A max) impedes snow metamorphism, resulting in smaller 485 grain size, and vice versa. A corresponding decrease in snow density under A max facilitates grain growth (Eq. B1 in Vionnet et al., 2012), but this effect is minor compared to the residence time. Increased grain size in T max can be attributed to an increase in the rate of snow metamorphism with temperature, independent of changes in temperature gradient (Legagneux et al., 2003).

Based on the results presented in our study, the anti-correlation between SSI and $\delta O_2/N_2$ is coherent with an increase in 490 near-surface grain size for a given density, leading to bulk ice with decreased $\delta O_2/N_2$. The opposite – a decrease in grain size for a given density – would thus result in bulk ice with increased $\delta O_2/N_2$. Under this rationale, our results suggest a dominant influence of accumulation rate over temperature, given the decreased grain size under A max conditions. We note, however, that accumulation rate and temperature co-vary and thus this interpretation is oversimplified. Regardless, a stronger correlation between $\delta O_2/N_2$ and accumulation rate than temperature in Figure 3 supports the dominant role of accumulation rate at low 495 accumulation sites. However, increased grain growth with temperature (Figure 6) contradicts this interpretation, highlighting a sensitive balance between temperature driven snow metamorphism, and accumulation driven burial rates. Our findings suggest a link between bulk ice $\delta O_2/N_2$ and density-dependent grain size near the surface, but also suggest the importance of snow/firn residence time, both near the surface, and in the LIZ.

### 4.3.2 The role of depth-dependent variability as a proxy for layering

The role of SSI and local climate on deep firn layering is considered by determining the sensitivity of density and grain size variability near the surface. Making the link between variability near the surface and variability in deep firn is not straightforward. Indeed, a study by Hörhold et al. (2011) used a compilation of firn cores from numerous polar sites to show that density variability in deep firn is positively correlated with local accumulation rate and temperature, but anti-correlated with near-surface density variability. Although, this anti-correlation was not observed by Inoue et al. (2023) when comparing firn 505 cores in the Dome Fuji area where the range of accumulation rate (temperature) was narrower. Additional consideration is required for density inversions in the firn column - whereby low-density layers located in the upper part of the firn become high-density layers below the density inversion depth, due to preferential deformation in the upper firn (Freitag et al., 2004; Fujita et al., 2009). Moreover, stratification at depth may also be influenced by impurity content, with impurity rich layers being more susceptible to densification (Hörhold et al., 2012).

Increased grain size variability in SSI max in Figure 7 supports the conclusions of Fujita et al. (2009) that layering is enhanced with SSI due to increased seasonality in snow temperature and temperature gradients, and vice versa. Conversely, decreased grain size variability in both the increased and decreased temperature simulations is likely linked to the way in which temperature forcing is perturbed. By applying a constant increase in air temperature, the strength of metamorphism is increased





during winter due to higher temperatures (Legagneux et al., 2003; Flanner and Zender, 2006), while snow temperatures during
summer are less effected (Figure S4) – due to the dominant influence of insolation on summer snow temperature. The weaker
effect during summer results in homogeneity in the snowpack. Opposingly, a decrease in temperature would suppress meta-
morphism throughout the year – as evidenced by a decrease in mean grain size (Figure 6) - resulting in relatively decreased
variability in the snowpack. The combined influence of accumulation rate and temperature perturbations is expected to result in
a complex response in both $\sigma_\rho$ and $\sigma_{gs}$. Moreover, the variability – and bulk mean – differences are likely to be very sensitive to
the ascribed glacial and interglacial accumulation rate and temperature values, leading to potentially inaccurate interpretations.
This is particularly important for glacial temperature reconstructions which are debated to have been overestimated by up to
5°C (Buizert, 2021).

Extracting concrete conclusions from the variability analysis and extrapolating these into the deep firn is inhibited by both
the ascribed forcing perturbations and the aggregation scheme of the model, which is particularly sensitive to changes in
accumulation rate. We thus conclude that the simulations performed with the Crocus model can support a mechanism of bulk
grain size on $\delta O_2/N_2$ in ice but not a mechanism implying grain size variability. However, our conclusion does not rule out the
effect of layering or grain size variability on $\delta O_2/N_2$ variability but highlights a limitation in our study, as explained in the next
section.

## 4.4 Perspectives and limitations

As noted in Section 3.2.1, there is a cold bias in simulated surface temperature compared to observations during summer, which
is expected to be consistent between simulations. We therefore argue that this should not influence our qualitative interpretations
of the anomalies in density and grain size (relative to the reference), especially considering that Crocus accurately reproduces
density and grain size profiles (Figure 5). The variability, however, is not fully captured by the model and can be explained
by the absence of snow transport by the wind in the standard version of Crocus (Libois et al., 2014). Libois et al. (2014) were
able to improve the variability reproducibility in the top 50 cm using a multi-patch approach to account for snow transport
by the wind. Further, Inoue et al. (2023) found that wind speed was the major factor controlling density variability from 6
cores near Dome Fuji. While the effects of wind on snow properties are important (Pinzer and Schneebeli, 2009; Dadic et al.,
2015; Inoue et al., 2023), the absence of wind transport from our simulations is not expected to influence the results from the
sensitivity tests. Moreover, from a paleo-climatological perspective, we are limited in our understanding of winds throughout
the Quaternary. In parallel, the aggregation of layers with depth in Crocus make it difficult to focus on both the fine-layered
near-surface snow, and the propagation of stratified layers into the deep firn. Accurate assessment of the layering effect would
require a new dedicated snow model preserving individual snow layers and properties over a large depth range, from the surface
to LIZ, and at high resolution.

We also reiterate that accumulation rate depends on temperature which in turn is linked to local SSI. We maintain that
the single-parameter sensitivity tests presented here provide useful insights for understanding physical mechanisms, but do
not account for complex compound effects expected in reality. Indeed, additional tests perturbing multiple forcing parameters
simultaneously indicate that snowpack properties are very sensitive to the ascribed accumulation rate and temperature values.



Because this modelling approach also has weaknesses, as detailed above, we did not use it for interpretation of compound effects in this study.

A number of commonly proposed mechanisms to explain $\delta O_2/N_2$ variability were considered in our analysis. However, it is important to note that there are other possible explanatory mechanisms. The influence of microstructural properties beyond grain size has not been discussed but poses an alternative mechanistic explanation for elemental fractionation during pore closure. Hutterli et al. (2009) theoretically showed that changes in near surface temperature gradients under varying climate states modulate the anisotropy of the ice due to variations in temperature gradient metamorphism. This theory was recently

confirmed by Leinss et al. (2020) who found that snow anisotropy was predominantly driven by vertical water vapour fluxes in the near-surface snow. Periods of high SSI facilitate temperature gradient metamorphism in the vertical, resulting in elongated pores allowing more fractionation (Hutterli et al., 2009; Leinss et al., 2020). Similarly, low accumulation rates will prolong the residence time of snow layers near the surface where temperature gradients are strongest, thus facilitating snow metamorphism (e.g. Inoue et al., 2023). While assessing pore shape and anisotropy is outside the scope of this study, the direction of change is

consistent with our results, such that periods of high accumulation rate, and temperature, would reduce near-surface temperature gradient metamorphism, allowing less fractionation resulting in increased $\delta O_2/N_2$.

     We lastly consider the bias towards low accumulation sites in our study. The majority of the datasets included in this study were relatively low-resolution, which limited the analysis of non-orbital temporal variability. While peaks in $\delta O_2/N_2$ corresponding to Dansgaard-Oeschger events are apparent in data from GISP2, which was previously observed by (Suwa and

Bender, 2008b), much of the data was either of too low resolution or influenced by storage gas-loss to perform additional analysis. The limited availability and temporal range of $\delta O_2/N_2$ records from Greenland cores meant that our study is slightly biased towards Antarctic sites, which tend to be characterised by low accumulation rates. Given this bias, we were unable to explore the opposing millennial-scale behaviour observed at low and high accumulation sites for both $\delta O_2/N_2$ and TAC (Kobashi et al., 2015; Eicher et al., 2016). Future studies would therefore benefit from obtaining high resolution measurements

from sites with different characteristics.

## 5   Conclusions

We present a compilation of $\delta O_2/N_2$ records measured on multiple ice cores from Greenland and Antarctica, to improve the mechanistic explanation for $\delta O_2/N_2$ variability. Analysis of both spatial (multi-site) and temporal (single-site) variability in $\delta O_2/N_2$ revealed new evidence of a dependence on local climate (accumulation rate and temperature), in addition to the

well-documented insolation dependence. High resolution measurements from the EDC ice core hinted to a millennial-scale variability in $\delta O_2/N_2$ behaving in-phase with $\delta D$ records when both parameters are plotted on the AICC2012 ice-age scale. The inter-site analysis revealed an increase in mean $\delta O_2/N_2$ for sites with higher accumulation rate and temperature, which is analogous with the temporal analysis from EDC showing $\delta O_2/N_2$ to increase together with $\delta D$.

     We argue for a dominant firn physical properties mechanism which links both the influence of SSI and local climate to

$\delta O_2/N_2$ variability on the ice-age scale by modulating near-surface snow properties. Sensitivity tests using the Crocus model



show that grain size is very responsive to perturbations in SSI, accumulation rate and air temperature, while density responds to all but SSI perturbations. Our findings support a grain size mechanism partially controls elemental fractionation during pore closure, such that increased grain size for a given density facilitates $O_2$ expulsion via enhanced permeability. We argue that the presence, or lack thereof at sites such as Dome Fuji (Kawamura et al., 2007), of a local climatic signal in $\delta O_2/N_2$ variability

is due to the delicate balance between the counter-effects of accumulation rate and temperature on grain properties. However, the inter-site results suggest that low accumulation, low temperature sites experience stronger elemental fractionation, having a comparable effect to high insolation.

While our findings from the $\delta O_2/N_2$ data compilation can be supported by the Crocus sensitivity tests, we acknowledge that there may be more complex mechanisms at play. In particular, the influence of deep firn layering – itself linked to surface snow

metamorphism – could not be tested fully in this study but is believed to play a major role on bulk ice $\delta O_2/N_2$. Determining the relative influence of stratification, firn physical properties and residence time in the lock-in zone, using firn models would be useful for future studies.

*Code availability.* The Crocus model is open-source, and the code is available at https://opensource.umr-cnrm.fr/projects/snowtools_git/ wiki/Procedure_for_new_users. The version used is labeled as Surfex V8_1.

*Data availability.* All unpublished $\delta O_2/N_2$ data measured at LSCE will be made available online. Published datasets are available online at the references in Table S2 in the supplement.

*Author contributions.* AL, EC and FP performed measurements/produced the unpublished datasets measured at LSCE - on ice provided by RM and BS - and CB and JS provided unpublished datasets measured at Scripps. Snow temperature data from Dome C was acquired and shared by LA and GP. RHS ran the Crocus simulations, with the support of MD and QL. RHS and AL prepared the manuscript with

contributions from all co-authors.

*Competing interests.* At least one of the authors is a member of the editorial board of The Cryosphere.

*Acknowledgements.* This publication was generated in the frame of DEEPICE project. The project has received funding from the European Union's Horizon 2020 research and innovation programme under the Marie Sklodowska-Curie grant agreement No 955750. The measurements leading to these results has also received funding from the European Union's H2020 Programme (H2020/20192024)/ERC

grant agreement no. 817493 (ERC ICORDA). EC acknowledges the financial support from the French National Research Agency under the "Programme d'Investissements d'Avenir" (ANR-19-MPGA-0001). MD has received funding from the European Research Council (ERC)



under the European Union's Horizon 2020 research and innovation program (IVORI; grant no. 949516). We also thank Matthieu Fructus for providing his expertise and support in the Crocus model.

*Financial support.* This research has been supported by the Horizon 2020 research and innovation programme (grant no. 955750 and grant

no. 817493).



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
