# Peer review of "$\delta\text{O}_2/\text{N}_2$ datasets and gas loss effect"

_EGUsphere, 2023_

## Referee Comment (RC2)

Review for Harris Stuart et al. 2023: ***Towards an understanding of the controls on δO₂/N₂ variability in ice core records*** by Jochen Schmitt, Bern

The paper, led by Romilly Harris Stuart, aims to improve our process understanding of a key ice core parameter - the O2/N2 ratio - that is used to orbitally date Antarctic ice core records. Decades ago it was discovered that the O2/N2 ratio resembles the orbitally-controlled solar insolation at the drill site. Already at that time, it was speculated how the upper firn layer is modulated by the amount of sunlight during summer. To affect the archived O2/N2 ratio in the bubbles, firn surface properties need to travel through the firn column to influence gas-specific (size-dependent) gas loss processes during the pore closure at the bottom of the firn. Over the years, a large number of studies suggested ideas to explain the observed O2/N2 variations, but we still lack an overall process understanding. While insolation apparently contributes a large fraction of the measured O2/N2 ratio, local temperature and accumulation rate modulate the orbital signal and lead to noise and uncertainty in the orbital tuning. At this target, Harris Stuart et al. place their study, which consists two approaches. Their study contributes to an important question relevant to the readers of The Cryosphere. Mostly, the paper is written clearly and provides the right depth of information and the figures are well-crafted and provide a visual support for the text. Overall, I support the publication of this study after minor revisions.

Their first approach is to apply an existing snowpack model to see if and to what extent differences in the solar radiation lead to changes in firn properties that might explain the observed O2/N2 ratios. Since the snowpack model was originally designed for alpine firn, applying it to low accumulation sites in Antarctica sets limitations. The authors became well aware of several limitations of their model (1-dimension, no wind compaction, merging box design) and thus interpreted their results with care. I got the impression that they used the model as much as it was possible for this study and then realized that no further insight could be obtained with this setup and that the model would need a significant improvement to capture the situation of low accumulation sites.

Their second approach is data-based and it was certainly a large effort to collect and screen all available O2/N2 records. The screening and data evaluation of the different cores and measurement campaigns is an important step and it would be crucial to provide a figure or two to allow the reader to see and understand the underlying problems of that step. Since it likely took a long time to collect all the records it would be helpful for others and the next generations of scientists to have easy access to these data sets and their meta information. So please spend some hours (perhaps more realistically days) of your time to bring all these data sets to a public database (both the already published and the new data). The analyses done on these 14 selected ice core sites conclude that factors other than insolation (accumulation rate and local temperature) have a sizable effect on the observed O2/N2 records and set limits to the precision and accurate of orbital tuning. This is a valuable outcome, but I feel that - in an ideal world with more time and resources - more can be done to disentangle the interplay between accumulation rate and temperature. As for the length of the diffusive firn column (i.e. d15N-N2), it might be the location on an accumulation vs temperature plot that determines if the firn column gets longer or shorter, or if the grain size within the first meter of firn increases or decreases. Since the temperature and accumulation rates are either known from present-day conditions or are output parameters of models (e.g. can be derived from delta age etc.), the team of this study might want to look a bit deeper into the interplay of temperature and accumulation rate in modulating O2/N2 ratios.

I also wondered if more process understanding can be gained from analyzing the O2/N2 data from firn air studies. At least there should be some O2/N2 data from some drill sites available. The authors mention several times that one modulating factor of the O2/N2 imprint in the archived air bubbles is the degree by which the O2-enriched air that was expelled by the closing pores is advected upwards or diluted in firn. In other words, the O2/N2 fractionation during pore closure is only seen if it happens in an open system, ie. if the O2-rich air is removed from that layer. See e.g. lines 433 – 438. Perhaps using a full firn model that allows the simulation of permeability in the deep firn could help here?

Further suggestions and technical comments:

Line 3: "trapped bubbles". I guess you want to say that the air in the bubble is sealed off from the open pore space; you can just say bubble since bubbles are closed anyway.

Line 4: write "… N2 molecules in extracted ice core air relative to the **modern** atmosphere - "

Line 6: write "…and show a new additional link…" delete: ", in addition to the influence of the summer solstice insolation"

Line 8: "… forcings modulate snow physical properties near the surface "

Line 10: "**a** mechanism**s**.."

Line 16: firn…unconsolidated snow? Firn is the consolidated snow

Line 18: rewrite "become sealed off from the firn air to form bubbles within the ice.

Line 18: "lock-in depth (LID)" actually you never use LID throughout the paper while you often use lock-in zone.

Line 22: komma after sites?

Line 21 to 28: perhaps restructure this a bit. Essentially you describe two different kinds of dating approaches. O2/N2 and TAC are due to local effects of the firn column, thus these parameters are highly site-specific. On the other hand, d18O of O2 is a globally mixed atmospheric gas parameter that is not site-specific, and all ice cores yield the same record. Thus it can be used to wiggle-match different records but also relate the record to a certain orbital parameter.
Would be good to mention these two different approaches
Ideally, you could mention that d18O2 is used to date the gas phase of the ice core while O2N2 is an ice age parameter

Line 28: delete "trapped within the ice" so it gets a bit more general

Line 30/31: delete "vice versa"

Line 31: you could delete "numerous" as you already name quite a few sites…

Line 34/35: you might rewrite this to convey that the modification due to insolation happens at the snow surface but the process that effectively alters the archived O2N2 ratio happens at the depth where the pores close off

Line 37: (COD) is just used twice …just write it out in both cases

Line 40: replace ; with :

Line 41: Why cite also her first name Tomoko?

Line 48: Why "They"? you refer to Bender (2002) so technically just Michael Bender although he acknowledges at the end of his paper that he profited a lot from the discussion with many giants in this field

Line 85: WAISD would be a new abbreviation, commonly used is WD or WAIS

Line 102 Table 1 (and other tables): for better visibility please align numbers in columns on the right side, e.g. Table 3 in Petrenko et al. 2016 http://dx.doi.org/10.1016/j.gca.2016.01.004

Line Table 1: If possible and available please also add other site characteristics to this table, e.g. close-off depth or ice age at close-off depth (delta age) they might be useful as well

Line 157: "gas loss during coring", can you explain a bit more here?

Line 163: Note that the brittle zone does not always correspond to the BCTZ, while for most of the ice cores, this is the case. I guess some ice cores have a technically defined brittle zone while they do not have the conditions to form clathrates at a certain set of depth or temperature; thus, without this coexistence of clathrate and bubbles, there shouldn't be a strong fractionation. Perhaps elaborate shortly on that.

Line 165: O2/N2 measurements within the brittle (or BCTZ) ice are not per se unreliable; it requires a post-coring gas loss, so the fractionated air in the bubbles escapes and thus induces scattered results. Also, small sample sizes resolve individual layers of bubbes vs clathrates

Line 167: see above, does Berkner have a chlathrate zone? Perhaps this explains good data within the brittle zone.

Line 176: post-coring gas loss to differentiate between the gas loss happening during pore closure in the ice sheet

Line 214: you mention the black carbon content. How sensitive is the model to the black carbon? What about a similar effect of mineral dust during glacial times (OK, mineral dust is mostly light quartz but there are also darker particles…)

Line 277: are the dots after the permil and the m2 correct?

Line 282: "integrated summer insolation": can say a few words on the difference between integrated summer insolation and SSI and why you use SSI?

Line 290: Figure 2 caption: you don't need to say that Dome C is plotted in dark-blue and Dome F in mid-blue because you can indentify each panel with their name already. Please add a), b), c) as you do in Fig. 3

Line 290: Figure 2 caption: why do you use r2 here, while in Fig. 3a, you use r for the same type of plot? perhaps always use r (as r2 can be calculated from that)

Line 312: Figure 3: I very much like your colour scheme, but here, it would also help to provide more visual hints to distinguish between some sites, e.g. LD and BI have quite similar colours (same for NEEM and WAIS). You could additionally use squares and diamonds.

Line 328: Table 3: The 5 EDML samples (596 – 860) are from the brittle zone. Are there no other samples measured at EDML, why just in the brittle zone?

Line 331: Fig. 4: Would the residuals look different if it would be plotted on the AICC2023?

Line Figure. 4 caption: the respectively structure always requires the reader to go to the end of the sentence while the classical way "Panel d shows the correlation" is often quicker to access

Line 354 Figure 5d: since there is no overlap between Jan and Jul, you could put both distributions into a single panel

Line 381 Figure 7: it is not easy to see the difference between the faded line and the max line, perhaps increase the thickness of the line or use dashed lines etc

Line 399:  the long list of references affects a bit the readability ....not sure if you need all the references here in the discussion section, perhaps write e.g. and two refs

Line 406: Fig. 3c, I guess you mean Fig. 3a showing as well O2/N2 vs SSI while Fig. 3c shows temperature. Where is the slope for Fig. 3a to compare it with the slopes of Fig. 2?

Line 438-440: I am not so sure if this argument holds that the O2/N2 signal would then be on the gas age scale. Still it happens in the lock-in zone due to a process that was imprinted originally at the surface.

Line 490: I am puzzled a bit about the term bulk ice…

Line 490: "The opposite – a decrease" not sure if this sentence describing the opposite effect is necessary I guess the sensitivity of grain size for a given density works in both directions

Line 493: yes, temperature and accumulation rate do generally covary, but they are not super tightly correlated, and there are sites that are above or below the expected line for the temperature–accumulation relationship. Perhaps you can derive some useful information from the deviations from this temperature-accumulation relation, i.e. a site that has too little accumulation rate for a given temperature. A scatter plot showing all sites with their accumulation vs temperature might help to identify sites that deviate from others in Figure 3. This requires O2/N2 data for the present-day conditions for accumulation and surface temperature that are likely more accurate than the reconstructed values based on modelling via water isotopes.

Line 500: this sentence is a bit unclear to me

Line 545: could you spend a few words on how the local SSI at EDC is linked to local accumulation rates since this is a larger-scale weather phenomenon and involves low-pressure systems entering the continent, etc.? Perhaps elaborate a bit on that?

Line 575: "local climate (accumulation rate" I understand what you mean but accumulation rate might be largely determined by the circulation patterns in the Southern Ocean region.

Line 576: I guess this statement also holds for AICC2023 (although there seems to be a small circularity hidden into that because the age scale is constructed using the O2/N2 orbital tuning)

Line 582: this sentence misses some words…support the idea…

---

## Author Comment (AC1)

**Response to anonymous Referee #1**

**Harris Stuart et al.: Towards an understanding of the controls on $\delta O_2/N_2$ variability in ice core records**

We are grateful to the reviewer for their time and effort in providing valuable feedback on the manuscript. These constructive comments have contributed to the improvement of the study. We have addressed all comments below and propose to implement the changes in a revised version of the manuscript.

Black = reviewer comment
blue = author's response / *"italic" = revised text.*

This paper presents the relationship between environmental conditions of the ice sheet surface and $O_2/N_2$ fractionation in combination with $O_2/N_2$ records from 14 ice cores and results from a snowpack model. The relationship between $O_2/N_2$ and local summer solstice insolation is well known, while the physical mechanism to create the insolation signal is poorly understood. The paper provides new $O_2/N_2$ data from Greenland and Antarctica. It qualitatively demonstrates the role of accumulation rate, surface temperature, and solar radiation on surface snow properties with the snowpack model, which contributes to understanding the mechanism of close-off $O_2/N_2$ fractionation. Overall, the subject is appropriate to *The Cryosphere*. However, there are several points which require some major revision. It appears that there are several areas with insufficient explanations throughout the manuscript.

**General comments:**

1. The analysis investigating the relationship between $O_2/N_2$ and SSI, A, or T (section 3.1) and the sensitivity experiments with the snowpack model (section 3.2) provide new results that deserve a thorough examination. In the current manuscript, the model results are well discussed, while the discussion connecting the observation of $O_2/N_2$ and the model results (i.e., surface conditions) is weak. The relationship between grain size and $O_2/N_2$ is not well described. I understand that the focus of this study is not to investigate how the surface snow conditions affect the densification processes/metamorphisms in the deeper firn, but readers may expect that the discussion would address the link between surface snow metamorphism and subsequent firn properties and how they are related to $O_2/N_2$ Section 4.1 contains a recitation of the results and gives an example of Greenland, but the main argument of this section is unclear. Section 4.2 is almost entirely a review of previous studies. In Section 4.3, I expect the author to develop their discussion here. The model results are discussed, but the arguments connecting to $O_2/N_2$ are unclear. In the revised manuscript, I expect to add some discussion connecting the results of Sections 3.1 and 3.2, for example, by introducing arguments from previous studies such as Fujita et al. (2009) and Hutterli et al. (2009). See also my specific comments.

Many thanks to the reviewer for this useful suggestion. We acknowledge that the discussion connecting the $\delta O_2/N_2$ and snowpack modelling results ought to be further developed. Given the useful feedback from both reviewers, we propose to reorganise the discussion to enable a deeper look into the interplay between temperature and accumulation rate, and to develop the mechanistic links with $\delta O_2/N_2$, while also improving readability. Proposed developments to discussion are included in response to specific comments, and we will update the neighbouring discussion sections in the revised manuscript to accommodate the new text.

2. I am not satisfied with the analysis of the link between $O_2/N_2$ and water isotope ratios of the EDC core, which is one of the bases for the discussion linking the $O_2/N_2$ fractionation mechanism with the model results. First, the authors found positive correlations between $\delta D$ and $O_2/N_2$ in the EDC core, but they are all separated by relatively short time periods where the amplitudes of the 20 kyr cycle of $\delta D$ are large (interglacials). Therefore, it seems to be not surprising to find correlations within those specific periods because $O_2/N_2$ has a strong precession component. The absence of a positive correlation in MIS11 may be because this interglacial period has a length of 2 precession cycles. If there is indeed a correlation between $O_2/N_2$ and T and/or A, a positive correlation would be expected even in MIS11. What would happen if correlations were examined over longer continuous periods, including periods with a smaller amplitude of 20 kyr cycle, such as MIS6? The validation may be possible by using data from the South Pole and Dome Fuji cores.

It is true that the high-resolution measurements from the EDC core are mainly during deglaciations and interglacial periods, but this was actually the objective of the study. We wanted to see if during the important local change in temperature such as during deglaciations and associated sub-orbital variability (e.g. linked to Heinrich events), there is an $\delta O_2/N_2$ signal which is not only linked to local summer solstice insolation since $\delta D$ itself is not in phase with change in local solstice insolation at EDC (e.g. Raynaud et al., 2023).

1) Using the AICC2023 chronology instead of AICC2012 does not significantly change the results of the SSI-$O_2/N_2$ residual analysis from the original manuscript. This is shown in Figure A below which compares Figure 4 from the original manuscript with data drawn on the AICC2012 time scale to the same analysis with data on the AICC2023 timescale. The $\delta D$ and SSI data were interpolated onto the $\delta O_2/N_2$ scale in both cases.

2) We agree that it would be beneficial to apply this analysis over the entire core, but we are limited by the available high-resolution data which were focused on the important $\delta D$ changes. The entire time series are presented in the figure accompanying the next comment to show the non-uniform measurement resolution. The key observation here is the coherent millennial-scale variability in $\delta D$ and $\delta O_2/N_2$ from the EDC core which is difficult to identify in low-resolution records (such as the older EDC data). Our response to the reviewers following comment will address the sub-orbital scale variability.

3) Applying the same analysis to Dome Fuji and South Pole cores reiterates the sensitivity of $\delta O_2/N_2$ to individual site conditions, as is mentioned below. We will include the analysis at these two sites in the supplement.

[Figure]

**Figure A**. Comparison between the evolution of $\delta O_2/N_2$, SSI, and $\delta D$ from EDC on the AICC2012 ice age chronology (light colours: Bazin et al., 2013), and on the AICC2023 chronology (dark colours: Bouchet et al., 2023). Panel a) presents the $\delta O_2/N_2$ (blues) with SSI on the right y-axis (grey), b) $\delta D$ (green), and c) the $\delta O2/N2$-SSI residuals (orange), between 200 and 260 ka BP. Scatterplots show the correlation between the $\delta O2/N2$-SSI residuals and $\delta D$ using d) AICC2012 chronology, and e) the AICC2023 chronology.

Second, if the authors want to investigate whether the 1000-year scale variations shown in grey shadings in Fig. 4 are related to the $O_2/N_2$ signal, it is necessary to remove orbital variations with appropriate methods. For this purpose, I suggest applying the low-pass filter used in orbital tuning to both $\delta D$ and $O_2/N_2$ and extracting residuals from low-pass filtered curves, or applying a high-pass filter to $\delta D$ and $O_2/N_2$ to cur off the insolation frequency. The current approach can introduce artificial variations due to the potential mismatch between the insolation and ice core ages. If the AICC2012 age scale is perfect in terms of absolute age, the long-term variation in Fig. 4b is real. However, as Extier et al. (2018) pointed out, the AICC2012 chronology tends to deviate from the U-Th chronology, especially during deglaciations. This raises the question about the accuracy of this chronology. Therefore, the orbital-scale variation shown in Fig. 4b may just reflect the phase difference between the AICC2012 chronology and insolation. Using the most recent AICC2023 chronology aligned with the U-Th chronology over the last ~600 kyr (Bouchet et al., 2023), or the DF2021 chronology (Oyabu et al.,

2022) synchronized with the local summer insolation, could potentially alter the appearance of the residuals in $O_2/N_2$.

We thank the reviewer for raising this and appreciate the suggestions. The aim was not necessarily to remove all orbital variations but to investigate the non-SSI drivers of $\delta O_2/N_2$ variability. We do acknowledge that separating the SSI signal from orbital-scale local climate variability is not trivial and this is the reason why we focus on the millennial-scale variability. However, the fact that the results are unchanged with the new chronology suggests a climate signal in the EDC $\delta O_2/N_2$ record, even if this signal is not always present. These results highlight the importance of individual site conditions when using $\delta O_2/N_2$ records as a proxy for local insolation.

Therefore, I suggest to conduct the following analyses, and add the results and corresponding discussion as needed.

1. Apply a low-pass filter to both $\delta D$ and $O_2/N_2$ and extract residuals from low-pass filtered curves to eliminate the potential discrepancy between the ice core and insolation ages. Alternatively, apply a high-pass filter to cut off the insolation signal.

The suggestion from the reviewer to apply a filter is very useful. Following the filter treatment from Kawamura et al. (2007) and Bouchet et al. (2023), a finite-impulse response (FIR) low pass filter is applied to the 100-year interpolated $\delta O_2/N_2$ and $\delta D$ records with a 10-16.7 kyr cut off (using a KaiserBessel20 window with 559 coefficients in Igor Pro) to isolate low frequency signals). The filtered curves are subtracted from the original curve to extract the high frequency (sub-orbital scale) signals, and both curves are smoothed using a 5-point moving average to reduce noise (Figure B).

As in the original paper, we focus on the longest available period of relatively high-resolution data (between 200-260 ka BP) and reach a similar conclusion that rapid changes (on the order of 1000's of years) in $\delta O_2/N_2$ often correspond with changes in $\delta D$. The linear regression of the high pass filtered $\delta D$ vs high pass filtered $\delta O_2/N_2$ is much lower than we found from the original analysis (r = 0.3, p < 0.0001), but this is expected given the absence of any low-frequency coherence in the records.

The same analysis cannot be applied to the entire record given that the filtering requires interpolated data which leads to oversampling during periods with low measurement resolution and impedes statistical analysis on the strength of the correlations. Unfortunately, the high-resolution measurements do not cover the entire of MIS11 (just the onset) and thus we are unable to fully assess the non-SSI variability in the $\delta O_2/N_2$ data. We do acknowledge that the coherence between $\delta O_2/N_2$ and $\delta D$ is far from constant, and we will address this more directly in the discussion of the revised manuscript.

[Figure]

**Figure B.** Time series of δO₂/N₂, δD, and SSI from EDC. Residuals of low pass filtered curves for δO₂/N₂ and δD are shown in the bottom two panels and the dashed box highlights the period of high-resolution data between 190-260 ka. Grey bars indicate age ranges exhibiting a degree of coherence between the two filtered curves and span the same time period as those in Figure 4a.

The following text will be included in results Section 3.1.2:

"*The following analysis uses high resolution δO₂/N₂ and δD measurements from the EDC ice core (Jouzel et al., 2007; Bouchet et al., 2023), covering five distinct periods between 111 and 539 ka BP, both on the AICC2023 ice-age scale (Bouchet et al., 2023). Following Bouchet et al. (2023), a finite-impulse response (FIR) low pass filter is applied to the 100-year interpolated δO₂/N₂ and δD records from each site with a 10-kyr cut-off to isolate the low-frequency signals (using a KaiserBessel20 window with 559 coefficients in Igor Pro). The filtered curves are subtracted from the original curve to remove the low frequency SSI signals, and a 5-point moving average is applied to the residuals to reduce noise but retain millennial-scale variability in the records. We primarily focus on the longest section between 190-259 ka BP (1980-2350 m) which covers MIS 7.*

*Figure 4, firstly shows the dominant SSI signal in the δO₂/N₂ curve, as has been documented previously (e.g., Landais et al., 2012). Superimposed onto this signal are millennial-scale peaks in δO₂/N₂ which appear to coincide with peaks in δD, high-lighted by grey bars in Figure 4c. This suggests a positive correlation with accumulation rate and temperature and shares analogy with the spatial positive correlation between δO₂/N₂ and both accumulation rate and temperature (Figure 3). Applying the same analysis to the Dome Fuji core reveals a similar millennial-scale signal in the high pass filtered δO₂/N₂ curve covarying with the high pass filtered δD (supplemental Figure S).*"

The discussion Section 4.1 will also be slightly modified given the different method; however, the overall result and interpretation remains largely unchanged.

"*Further investigation into drivers of δO₂/N₂ variability in the EDC core reveals a sub-orbital climate signal (from δD - a proxy for accumulation rate and temperature on the long-time scale considered here) in the δO₂/N₂ records. The non-SSI variability superimposed onto the δO₂/N₂ curve is considered by taking the residuals of the low pass filtered curves of δO₂/N₂ and δD from EDC. Our results show a coherence in the timing of millennial-scale anomalies in the filtered δD and δO₂/N₂ curves.*"

2. Use the AICC2023 chronology to examine SSI-residuals.

Please see response to the two previous comments.

3. Use the Dome Fuji $O_2/N_2$ and water isotope data, and use the DF2021 chronology to examine SSI-residuals.

Applying the same analysis to the Dome Fuji records similarly reveals some coherence in millennial-scale variability between residuals from the low-pass filtered curve of $\delta D$ and $\delta O_2/N_2$ records (Figure C below). However, the $\delta D$ maximum around 130 ka (corresponding to MIS 5) is not present in the $\delta O_2/N_2$ data, as may be expected considering findings from Kawamura et al. (2007). During the glacial periods there appears to be some coherence (highlighted by the grey bars in the figure below) but the signal is very weak compared to EDC. Data older than 170 ka are also of too low resolution for analysis of sub-orbital variability.

4. If applicable, use the South Pole core $O_2/N_2$ and water isotope data, and use the SP19 chronology to examine SSI-residuals.

Similarly, no significant link is observed between $\delta O_2/N_2$ and $\delta D$ in the South Pole core. This may be attributed to the relatively short time period (35 ka), and differences in the relative influence of SSI, accumulation rate, and temperature between sites, especially given the relatively high accumulation rate at South Pole compared to Dome F and EDC. We propose to include the analysis from Dome F and South Pole in the Supplement.

[Figure]

**Figure C.** Time series of $O_2/N_2$, $\delta D$, and SSI from Dome F (left) and South Pole (right). Residuals of low pass filtered curves for $O_2/N_2$ and $\delta D$ are shown in the bottom two panels for both sites. Grey bars indicate age ranges exhibiting a degree of coherence between the two filtered curves. Data from the BCTZ and above are not considered here due to strong scattering between 40 and 100 ka BP at Dome F, and anomalously high $O_2/N_2$ values at South Pole between 8-18 ka. BP.

**Specific comments:**

Title: I suggest changing the title to be more specific. This paper focuses mainly on inland Antarctica (Dome C), although the Greenland records are used, and discusses the relationship between $O_2/N_2$ and the surface environments. Thus, I suggest including the idea of "the relationship between $O_2/N_2$ and the ice sheet surface environment" and "inland Antarctica" in the title.

We agree and will change the title accordingly. However, while we acknowledge that the snowpack modelling and temporal analysis focusses on Dome C, our compilation includes sites several Antarctic coastal stations. Therefore, we propose to modify the title to:

"*On the relationship between δO₂/N₂ variability and ice sheet surface conditions in Antarctica.*"

Line 24: Please consider adding two more references: Lipenkov et al. (2011) for the Vostok chronology and Oyabu et al. (2022) for the DF chronology. Both used $O_2/N_2$ for orbital dating.

The suggested references have now been included.

Around line 60: Please consider adding a hypothesis by Kawamura et al. (2007) that the absence of a climatic signal may result from the cancellation of temperature and accumulation effects on $O_2/N_2$.

The hypothesis by Kawamura et al. (2007) has now been included.

Line 60-62: *"There is a growing body of evidence for a local climatic imprint on ice core δO₂/N₂ records. Firstly, spectral analysis has revealed climate related 100-ka cyclicity at EPICA Dome C (Bazin et al., 2016). However, such a signal is not apparent in the Dome Fuji core which Kawamura et al. (2007) attribute to the cancellation of temperature and accumulation effects in the δO₂/N₂ record."*

Line 100: I suggest also providing a brief explanation of the methodology at Scripps in a similar manner as that for LSCE.

The melt-refreeze technique at Scripps is very similar to that used at LSCE. We propose to add the following text to Section 2.2:

Line 93: "*In short, the first uses a melt-refreeze technique based on Sowers et al. (1989), where ice…*"

Line 100: "*Unpublished data from the GISP2 core were measured at the Ice Core Noble Gas Laboratory of the Scripps Institution of Oceanography using the same melt refreeze technique as used at LSCE (Sowers et al., 1989; Petrenko et al., 2006).*"

Line 103: "An over view of ..." the same information is written in Section 2.1.

To clarify, Table 1, referred to in Section 2.1, is referring to the overview of site characteristics. At the end of Section 2.2 we refer to Table S1 and S2 in the supplement which contain information about the datasets used in the study.

Section 2.2.1: It is not clear which samples were measured at LSCE and which at Scripps. If only the GISP2 samples were measured at Scripps, it would be better to switch the order and write it at the end.

Only GISP2 were measured at Scripps and the description for these measurements have been moved to the end of Section 2.2.1.

"measured on the 10-collector Thermo Delta V Plus" is not necessary for all samples. This information should be shown at once in the first paragraph of Section 2.2.

The following sentences have been added at the start of Section 2.2:

"*Samples from the following sites were measured at LSCE, France on a 10-collector Thermo Delta V Plus, unless otherwise stated. In all cases, the values used in this study are the average of at least two replicate measurements.*"

Line 163-165: The brittle zone is a zone of poor ice-core quality and not necessarily consistent with the bubble-clathrate hydrate transition zone (BCTZ) (Neff, 2014). Thus, I suggest using "BCTZ" (or a similar term) as in the supplement text. If you did not find the BCTZ information in some cores and thus employed the brittle zone of Neff (2014), you should write it in the text. Also, "the air in the gas phase has a very different composition to that in the clathrate hydrates" should be "the air in the **bubble** has a very different composition to that in the clathrate hydrates" (e.g., Ikeda-Fukazawa et al., 2001).

Thank you for pointing this out. Information in Table 1 (and corresponding text) now uses BCTZ depth range instead of the more general brittle zone. The majority of measurements within the published 'brittle zones' were also the BCTZ, and those that differ (Berkner Island, Siple Dome, Roosevelt Island, and Law Dome) have been updated in the text and for the data rejection. We have also updated the sentence.

"*At the bubble-clathrate transition zone (BCTZ: where the high hydrostatic pressure in the bubbles cause entrapped gases to form clathrate hydrates (Schaefer et al., 2011)) elemental fractionation occurs due to some gas species being preferentially incorporated into the clathrate structures (Ikeda-Fukazawa et al., 2001), thus making the interpretation of gas measurements unreliable at these depths (Bender, 2002).*"

Line 166: $O_2/N_2$ is not always increased in the BCTZ (e.g., Oyabu et al., 2021).

Many thanks for highlighting this. The sentence has been corrected in the updated version.

"*Measurements from the BCTZ tend to be characterised as having increased mean $\delta O_2/N_2$ (usually in excess of 0‰) or strong data scattering, expressed as a high standard deviation (Oyabu et al., 2021).*"

Lines 178-183: Regarding the data rejection criteria, it is unclear what was rejected with and without gas loss correction, and what was employed with and without gas loss correction. It seems that some of the data you utilized was affected by the gas loss but was included in the analysis because the gas loss correction worked. Also, the data with correction should be written as "corrected" since it is not unaffected by gas loss.

I would like to see a figure that displays all $O_2/N_2$ data, including rejected and employed data with and without gas loss corrections, with different colors or symbols.

Thank you for bringing this to our attention. There was no attempt to apply a gas loss correction given the uncertainty of storage history for all cores, as mentioned in Section 2.2.2. Consecutive measurements over a ~15-year period at LSCE revealed that bubbly ice is largely unaffected by gas loss during storage while clathrate ice becomes strongly depleted in $O_2$ (Fig S1 and S2; Bouchet et al., 2023). Measurements from bubbly ice are therefore not rejected based on the storage time. We do acknowledge that previous studies have suggested that $\delta O_2/N_2$ in bubbly ice is more susceptible gas loss (Yan et al., 2023), however this was not observed at EDC and TALDICE.

Due to the vastly different depth ranges for all the sites we propose to include a figure with two panels, showing all the $\delta O_2/N_2$ data from each site before and after data rejection. Below is an example of the proposed figure, showing boxplots of all data prior to and after data rejection.

[Figure]

**Figure D**. Boxplots showing the median, standard deviation, interquartile range, and outliers, overlain with individual data points for each site. The top panel displays all the data prior to data rejection. The lower panel shows the data for each site after the removal of data influenced by gas loss, data from the BCTZ, and outlier removal (defined as values outside of the interquartile range).

Lines 279-280: I didn't understand what the authors meant.

This sentence refers to the fact that $\delta O_2/N_2$ data from the South Pole ice core are higher than EDC and Dome F. Higher accumulation rate and temperature at South Pole may explain the shift, supported by the positive correlation between $\delta O_2/N_2$ and both temperature and accumulation rate (Figure 3 in paper). The following sentence may be used instead to improve clarity.

"*$\delta O_2/N_2$ values from South Pole are higher than for EDC and Dome F for the same SSI, suggesting additional factors are influencing the records, such as accumulation rate – which at South Pole is over double that of both EDC and Dome F – or integrated summer insolation.*"

Line 288: "1000 year averaged SSI". Did you average for the last 1000 years?

Yes, the values presented in the original paper are the average SSI over the last 1000 years for the latitude of each site. However, we realise that it would be more robust to compare to the mean SSI over the same time period as the $\delta O_2/N_2$ data. Please see response to comment after next for proposed updates relating to mean SSI for each site.

Line 292: "bias toward cold, low-accumulation conditions" is unclear. Does it mean that A and T are averaged between the LGM and the present and do not include past interglacials that were warmer than the present?

A bias towards "cold, low-accumulation conditions" rather refers to the site locations. However, the reviewer makes a useful point in the lack of information of time periods presented in the original version of the manuscript. Where available, we have now included the age-scales and propose to evaluate potential biases relating to the time period (glacial vs interglacial), i.e., whether the deviations of the mean $\delta O_2/N_2$ from the linear regressions - in Figure 3 in the manuscript - are influenced by the time period of the data.

Lines 301-302: I am curious whether this regression would be better if you used SSI for the same time period as the $O_2/N_2$ data. Have you confirmed before?

The regression for Figure 3a is not improved when comparing to SSI over the same time period as the $\delta O_2/N_2$ data. In any case, we propose to use the mean SSI values over the same time period as the data (where age scales are available) for continuity of comparison between variables. Scatter plots showing the original and new $\delta O_2/N_2$ vs SSI values are included below.

[Figure]

**Figure E.** Scatterplots showing the dependence of δO2/N2 on a) the average SSI for each site over the last 100 years, (as in the original manuscript), and b) the mean SSI at each site's present latitude over the time period of the $\delta O_2/N_2$ measurements. Note that any differences in mean $\delta O_2/N_2$ comes from adjustments in the brittle zone definition, and colours differ from Figure D in these responses because the sites in this figure are ordered by accumulation rate.

Proposed updates to the results description of Figure 3 from the manuscript:

"*In addition to the $\delta O_2/N_2$ datasets, we compile SSI, accumulation rate, and temperature reconstructions from each site covering the same age range as the $\delta O_2/N_2$ data (where available). The range of SSI, accumulation rates, and temperatures are included to indicate the climate histories for each site (Table 3). The data are presented in Figure 3 for a) SSI, b) accumulation rate, and c) temperature. Error bars on the x-axis indicate the range of values with the exceptions of Law Dome sites DE08 and DSSW20k, where only the present-day values are used.*"

Lines 315-317: I don't think these descriptions are necessary here. It is obvious that this paper does not use $O_2/N_2$ for orbital dating.

We have removed these sentences and replaced them with the description of the filtering techniques (included in response to general comments) used for the temporal analysis of the EDC records (as well as Dome F and South Pole in the supplement).

Line 317: If all EDC $O_2/N_2$ data were published in Bouchet et al. (2023), I think it would no longer be new in this paper.

The sentence has been modified accordingly.

"*The following analysis uses relatively high resolution $\delta O_2/N_2$ and $\delta D$ measurements from the EDC ice core (Grisart et al., 2021; Bouchet et al., 2023), covering five distinct periods between 111 and 539 kaBP, both on the AICC2023 ice-age scale (Bouchet et al., 2023).*"

Section 2.3: What is the depth range or thickness of each layer considered in this model? Also, the authors mention that the maximum number of layers was increased from 50 to 80; was this a new modification made specifically for this study?

Snow layer thickness in Crocus dynamically evolves through time with densification. In the version of the model used here, the layer thickness ranges from 2 mm at the surface (Libois et al., 2014) to over a metre in some

circumstances. The range is mentioned in Section 2.3.1, but we propose to explicitly state the minimum and maximum in our simulations instead of the current "... millimetres to metres…".

*"Briefly, the initial number of layers is defined by the user, with the thickness of each layer allowed to change along the simulation (layer thickness ranging from 2 mm at the surface to metres thick)."*

For our Dome C simulations where the accumulation rate is very low and simulations are run with a 100-year spin up, the thickest layers are likely to be at base of the snowpack owing to the aggregation of layers when the snowpack is already comprised of 80 layers. We chose to increase the number of layers to 80 to maximise resolution with depth compared to the 50 layers ascribed in Libois et al. (2014).

*"The maximum number of layers available in the model was increased from 50 (Libois et al., 2014) to 80 (this study) to maximise the resolution with depth owing to the higher number of thin layers forming at Dome C than at Alpine sites."*

Line 323:325: As I pointed out above, this long-term variability may just reflect the phase difference between the AICC2012 chronology and insolation.

Please see the response to general comments above.

Lines 342-343, Figure 5a and 5b: What factors contributed to the increased SSA for the top 10 cm and the subsequent dips in both SSA and density around the 10 cm depth in the model results?

There are various possible explanations for the high simulated SSA in the top 10 cm. Three possibilities are included here, and we will mention this in the revised manuscript.
1) One possibility relates to the 3∘C bias in snow temperature in the model compared to observations (Figure 5d), resulting in less grain growth in the upper snowpack.
2) Alternatively, the high values may be linked to large precipitation events adding new snow with high SSA. During large precipitation events, the fresh snow will be buried rapidly and undergo less metamorphism than the topmost snow given that most metamorphism occurs in the top 5 cm where solar radiation drives strong temperature gradients (Picard et al., 2012). This could result in sustained high SSA in the top 10 cm.
3) Finally, the snowpack was initialised using a 100-year spin up where we ran the forcing file 10 times between 2000 and 2010, followed by the period from 2000 to 2020. The distinct snow properties in the top 10 cm may potentially relate to a transition from repeated atmospheric forcing file.

Lines 371-373: "These opposing influences of accumulation rate and temperature on snow properties at first appears to contradict the observation in Figure 3" I agree that the opposing influences appear to contradict, while this is consistent with the hypothesis by Kawamura et al. (2007) (cancellation of temperature and accumulation effects on O$_2$/N$_2$). How about mentioning this consistency?

The consistency with the hypothesis of Kawamura et al. (2007) is now included, although here we are also referring to the dominant effect of accumulation rate over temperature on δO$_2$/N$_2$ between sites. In this case, it appears as though accumulation and temperature effects are not muted by one another, but that accumulation rate is dominant. The cancellation of temperature and accumulation effects on δO$_2$/N$_2$ through time at a given site is indeed important and is now discussed in Section 4.4. The sentences in lines 371-373 now read:

*"The opposing influence of accumulation rate and temperature at first appears to contradict the observations in Figure 3, where δO$_2$/N$_2$ increases with both variables, but is supported by the observations at Dome F and hypothesis of Kawamura et al. (2007) that the effects of temperature and accumulation δO$_2$/N$_2$ are cancelled out, at least in this site."*

Additional text will also be included in Section 4.4:

*"We thus expect that a local climatic signal is only present in $\delta O_2/N_2$ records when there are deviations from the accumulation rate-temperature relationship. Indeed, a cancellation of the accumulation rate and temperature effects were invoked by Kawamura et al. (2007) to explain the absence of a 100-ka periodicity at Dome Fuji."*

Line 373-375: I didn't understand the sentence. "most evident is the sensitivity of grain size to accumulation rate" Why can you say that?

Figure 6c shows that the mean grain size over the top 20 cm (and deeper) is more sensitive to decreased accumulation rate (-10% in grain size compared to the reference simulation) than to increased (31%). This sentence aims to highlight this non-linearity.

*"Mean density and grain size respond non-linearly to perturbations in all forcing parameters–clearly documented by the magnitude of increase in grain size from decreased accumulation rate being 3 times greater than the decrease induced by an increase in accumulation rate. This is in line with the dependence of $\delta O_2/N_2$ to the logarithm of accumulation rate documented in Figure 3."*

Line 386: It is not clear why the decrease in mean density and increase in density variability with A max and T min is "surprising". Some more explanation is needed.

We agree that surprising is not the correct word to use here and it has been removed to avoid confusion.

Line 389: I suggest inserting "increase in" or similar words between "with" and "SSI".

The sentence has been updated and now reads:

*"$\sigma_{gs}$ appears to increase with an increase in SSI and accumulation rate throughout the top metre."*

Lines 421-423: "The observations from…." The sentence is unclear. Please clarify.

The intention of this sentence is to highlight that the dominant mechanisms driving $\delta O_2/N_2$ variability in Antarctica may differ from those in Greenland due to their different climates. Indeed, Kobashi et al. (2015) show a multi-decadal scale variability in $\delta Ar/N_2$ from GISP2 which is anti-correlated with accumulation rate – opposite what we observe on millennial timescales at Dome C between $\delta O_2/N_2$ and $\delta D$ (accumulation rate). In parallel, total air content studies have shown an anti-correlation between TAC and accumulation rate in Greenland (Eicher et al., 2016) but a positive correlation in Antarctica (Epifanio et al., 2023). We propose the following sentence for clarification in the text:

*"Epifanio et al., (2023) proposed that the contradictory behaviour of TAC between NGRIP, Greenland and South Pole, Antarctica may be explained by different responses of the firn to changes in accumulation rate for different sites with different surface climatic conditions. They argue that a grain size mechanism is dominant at low accumulation sites while transient effects from rapid climatic changes are more important at warm, high accumulation sites (Epifanio et al., 2023). The overlap between mechanisms controlling TAC and $\delta O_2/N_2$ (Fujita et al, 2009) indicate that a similar effect may explain the positive correlation we observe between $\delta D$ and $\delta O_2/N_2$ at EDC."*

Lines 423-425: This section is for discussion and I don't think this statement fits here.

We agree and have removed this sentence from the updated manuscript.

Lines 469-470: I would suggest to delete the sentence "Our findings…" and move the contents of lines 470-472 to the last paragraph of section 4.3.1. If you would keep the sentence here, more words are needed (it is unclear what "our findings" and "a density-dependent grain size mechanism" refer to).

Please see response to the comment after next which outlines a proposed modification to Section 4.3.

Lines 473-474: Not necessary. This sentence is a repetition of the sentence in lines 468-469.

The repetition has been removed from the updated manuscript.

Lines 489-492: Hard to understand. "leading to bulk ice with decreased $\delta O_2/N_2$", but the sentence before this phrase alone does not yet clarify the causal relationship. Your analysis of the ice core data shows that $O_2/N_2$ decreases as SSI increases (anti-correlation), and your model results show that grain size decreases as SSI increases. This would mean that there should be an anti-correlation between grain size and $O_2/N_2$. In addition, there seems to be a lack of explanation why/how the increased grain size depletes $O_2/N_2$. You may consider adding discussion, drawing arguments from previous studies as described in the Introduction section, to explain why a decreased (increased) $O_2/N_2$ is associated with a larger (smaller) grain size. One idea may be bring the discussion of Calonne et al. (2022) and Gregory et al. (2014), which appeared in lines 470-473, to here.

To improve clarity on this discussion section we propose to include a flow diagram to illustrate the relationship between the forcing parameters and both grain size and $\delta O_2/N_2$, and to expand the interpretation within the text. Actually, our model results show that grain size increases as SSI increases, thus, a positive correlation between grain size and $\delta O_2/N_2$ would be expected. This supports previous studies such as Bender (2002). We propose to modify Section 4.3.1 as follows:

*"Crocus sensitivity tests show an increase in grain size under increased SSI (S max in Table 2), attributed to both increases in near-surface temperature gradients (Appendix B in Vionnet et al., 2012) and increased snow temperature during summer (Figure S4 in supplement). Previous studies have proposed that near surface grain size determines the density at pore close-off (Gregory et al., 2014), and as such, the pore volume (Goujon et al., 2003). Larger firn grain size is associated with higher density at close-off, higher SSI is associated with larger grain size and lower $\delta O_2/N_2$, hence, firn with large grain size and small pore volume is expected to experience more elemental fractionation during close off (lower $\delta O_2/N_2$). Indeed, Calonne et al. (2022) showed that grain size has a strong influence on permeability, such that for a given density, permeability is increased with grain size. Moreover, Gregory et al. (2014) found that permeability is increased in high-density, large-grain size firn and they specifically attribute this to a less complex pore structure.*

*Near surface grain size is also influenced indirectly by the residence time of a snow layer in the upper centimetres to metres of the snowpack, where temperature gradients are strongest (Vionnet et al., 2012; Picard et al., 2012). Longer residence time with decreased accumulation rate (A min) facilitates snow metamorphism, resulting in larger, more rounded grains (Colbeck, 1983) - as we observe in our sensitivity results in Figure 6. In addition, sites with low accumulation rates are usually characterised by a thin lock-in zone with larger, more rounded grains - associated with a less complex pore structure at close-off (Gregory et al., 2014) - which facilitates diffusivity and the removal of fugitive gases (enriched $O_2$) back to the atmosphere (Landais, 2004; Witrant et al., 2012). A grain size mechanism may therefore also be invoked for low accumulation rates, which is supported by the positive correlation between accumulation rate and $\delta O_2/N_2$ in Figure 3b."*

Lines 498: What is the link between your findings and residence time in the LIZ? The model results show only near the ice sheet surface, and there seems to be no discussion of how the results relate to the $O_2/N_2$ fractionation in the LIZ (deep firn).

Our modelling results do not reveal anything about the LIZ in themselves. This statement was in reference to the mechanism proposed by Severinghaus and Battle (2006), but it is true that our results do not provide a link to the LIZ residence time, and we will clarify this in the updated manuscript.

Lines 519-520: What does "the variability– and bulk mean – differences" refer to? I didn't understand what the authors meant.

"bulk mean" here should be in brackets as we are saying that both the variability, and mean density and grain size values are sensitive to the ascribed accumulation rate and temperature values. This will be corrected in the revised discussion.

Lines 556-567: Need to explain why elongated pores lead to a greater fractionation of $O_2/N_2$.

The following sentence will be added to the discussion:

"*Periods of high SSI facilitate temperature gradient metamorphism, leading to vertically elongated pores in the lock-in zone (Hutterli et al., 2009; Leinss et al., 2020). Vertical diffusivity in the lock-in zone is hypothesised to be enhanced in such cases, leading to greater fractionation of $\delta O_2/N_2$ (Hutterli et al., 2009; Fujita et al., 2009).*"

Lines 583-585 (We argue that the…): I don't see this argument in Discussion. This is the conclusion section and not a good way to introduce a new argument. The argument should be addressed in the Discussion section.

Apologies for the lack of continuity. The cancellation of accumulation and temperature effects in $\delta O_2/N_2$ records will be included in the discussion Section 4.4 Perspectives and limitations.

"*Indeed, additional tests simultaneously perturbing accumulation rate and temperature indicate that snowpack properties are very sensitive to the ascribed accumulation rate and temperature values (i.e., glacial temperature reconstructions which are debated to have been overestimated by up to 5°C (Buizert et al.,2021). We thus expect that a local climatic signal is only present in $\delta O_2/N_2$ records when there are deviations from the accumulation rate-temperature relationship. Indeed, a cancellation of the accumulation rate and temperature effects were invoked by Kawamura et al. (2007) to explain the absence of a 100-ka periodicity at Dome Fuji.*"

Table 3: A max of the Dome Fuji core seems to be too large (even larger than at EDML). The accumulation rate of the Dome Fuji core over the last 720 ka can be found at NOAA Paleo Data Search.

Thank you for pointing out this mistake. The maximum value has been corrected to 4.1 cm w.eq yr$^{-1}$ and the associated reference has been updated (Kawamura et al., 2017).

**Technical corrections:**

Line 18: "LID" only appears here, but "LIZ" appears without abbreviation (e.g., line 69).

Thank you for pointing this out. Line 18 now reads:

"*Atmospheric air moves through porous networks within the firn until a critical depth (known as the lock-in depth) where vertical diffusion effectively stops, and pores gradually become closed off from the atmosphere. The lock-in depth and the depth at which all pores are closed (close-off depth) are largely determined by local accumulation rate, temperature, and possibly the degree of density layering (Schwander et al., 1997; Martinerie et al., 1994; Mitchell et al., 2015). The region between the lock-in depth and close-off depth in known as the lock-in zone (LIZ).*"

Line 41: "Tomoko Ikeda-Fukazawa and Hondoh, 2004" is "Ikeda-Fukazawa et al., 2004".

The citation has now been corrected in the revised manuscript.

Line 154: Add "slope" after "Chemical"

This has been updated in the text.

Line 277 and 3rd line of the Fig. 2 caption: ‰.m$^2$.W$^{-1}$ Remove periods.

The units have been corrected in the updated version.

Line 297: Figure 2b may be 3b.

Yes, apologies, this has been corrected.

Line 300: panels (a) and (b) may be panels (b) and (c).

Many thanks for this pointing this out. We have corrected this is the text.

Line 301: Large residuals in "Figure 2a" should be in "Figure 3a". I suggest replacing "residual" with another term, such as deviation from the regression line.

The figure reference has been updated and the term 'residuals' has been changed.

Line 314: EPICA Dome C is already shortened in Line 82.

EPICA Dome C has been replaced with "EDC".

Line 363: Figure 6c and 6d may be 6a and 6b.

The text has been changed to, "*Figure 6c shows the mean difference in density and grain size from the Dome C reference simulation (Ref) and each of the test scenarios (outlined in Table 2)*."

Line 388: 50cm. ,  Remove period and space.

Corrected in the revised version.

Line 417: (Suwa and Bender, 2008b) -> Suwa and Bender (2008b)

The citation formatting has been corrected.

Table 1: Brittle zone should be bubble-clathrate transition zone (BCTZ) or a similar term.

Changed to BCTZ (also throughout the text).

Figure 5 caption: Density (a) and SSA (b)

We thank you for pointing the out. The caption has been corrected.

**References:**

Bazin, L., Landais, A., Lemieux-Dudon, B., Toyé Mahamadou Kele, H., Veres, D., Parrenin, F., Martinerie, P., Ritz, C., Capron, E., Lipenkov, V., Loutre, M.-F., Raynaud, D., Vinther, B., Svensson, A., Rasmussen, S. O., Severi, M., Blunier, T., Leuenberger, M., Fischer, H., Masson- Delmotte, V., Chappellaz, J., and Wolff, E.: An optimized multi-proxy, multi-site Antarctic ice and gas orbital chronology (AICC2012): 120ndash;800 ka, Climate of the Past, 9, 1715–1731, https://doi.org/10.5194/cp-9-1715-2013, 2013.

Bender, M. L.: Orbital tuning chronology for the Vostok climate record supported by trapped gas composition, Earth and Planetary Science Letters, 204, 275–289, https://doi.org/https://doi.org/10.1016/S0012-821X(02)00980-9, 2002.

Bouchet, M., Landais, A., Grisart, A., Parrenin, F., Prié, F., Jacob, R., Fourré, E., Capron, E., Raynaud, D., Lipenkov, V. Y., Loutre, M.-F., Extier, T., Svensson, A., Legrain, E., Martinerie, P., Leuenberger, M., Jiang, W., Ritterbusch, F., Lu, Z.-T., and Yang, G.-M.: The Antarctic Ice Core Chronology 2023 (AICC2023) chronological framework and associated timescale for the European Project for Ice Coring in Antarctica (EPICA) Dome C ice core, Clim. Past, 19, 2257–2286, https://doi.org/10.5194/cp-19-2257-2023, 2023.

Dome Fuji Ice Core Project Members, Kawamura, K. et al: State dependence of climatic instability over the past 720,000 years from Antarctic ice cores and climate modelling, Sci Adv, 3, https://doi.org/10.1126/sciadv.1600446, 2017.

Eicher, O., Baumgartner, M., Schilt, A., Schmitt, J., Schwander, J., Stocker, T. F., and Fischer, H.: Climatic and insolation control on the high-resolution total air content in the NGRIP ice core, Climate of the Past, 12, 1979–1993, https://doi.org/10.5194/cp-12-1979-2016, 2016.

Epifanio, J. A., Brook, E. J., Buizert, C., Pettit, E. C., Edwards, J. S., Fegyveresi, J. M., Sowers, T. A., Severinghaus, J. P., and Kahle, E. C.: Millennial and orbital-scale variability in a 54 000-year record of total air content from the South Pole ice core, The Cryosphere, 17, 4837–4851, https://doi.org/10.5194/tc-17-4837-2023, 2023.

Extier, T., Landais, A., Bréant, C., Prié, F., Bazin, L., Dreyfus, G., Roche, D. M., and Leuenberger, M.: On the use of δ18Oatm for ice core dating, Quat. Sci. Rev., 185, 244-257, https://doi.org/10.1016/j.quascirev.2018.02.008, 2018.

Fujita, S., Okuyama, J., Hori, A., and Hondoh, T.: Metamorphism of stratified firn at Dome Fuji, Antarctica: A mechanism for local insolation modulation of gas transport conditions during bubble close off, J. Geophys. Res., 114, https://doi.org/10.1029/2008JF001143, 2009.

Hutterli, M., Schneebeli, M., Freitag, J., Kipfstuhl, J., and Röthlisberger, R.: Impact of local insolation on snow metamorphism and ice core records, Physics of Ice Core Records II : Papers collected after the 2nd International Workshop on Physics of Ice Core Records, held in Sapporo, Japan, 2-6 February 2007. Edited by Takeo Hondoh, 2009.2009.

Ikeda-Fukazawa, T., Hondoh, T., Fukumura, T., Fukazawa, H., and Mae, S.: Variation in N2/O2 ratio of occluded air in Dome Fuji antarctic ice, J. Geophys. Res., 106, 17799-17810, https://doi.org/10.1029/2000JD000104, 2001.

Ikeda-Fukazawa, T., Kawamura, K., and Hondoh, T. (2004) Mechanism of Molecular Diffusion in Ice Crystals, Molecular Simulation, 30:13-15, 973-979, DOI: 10.1080/08927020410001709307

Kawamura, K., Parrenin, F., Lisiecki, L., Uemura, R., Vimeux, F., Severinghaus, J. P., Hutterli, M. A., Nakazawa, T., Aoki, S., Jouzel, J., Raymo, M. E., Matsumoto, K., Nakata, H., Motoyama, H., Fujita, S., Goto-Azuma, K., Fujii, Y., and Watanabe, O.: Northern Hemisphere forcing of climatic cycles in Antarctica over the past 360,000years, Nature, 448, 912-916, https://doi.org/10.1038/nature06015, 2007.

Libois, Q., Picard, G., Arnaud, L., Morin, S., and Brun, E.: Modeling the impact of snow drift on the decameter-scale vari- ability of snow properties on the Antarctic Plateau, Journal of Geophysical Research: Atmospheres, 119, 11,662–11,681, https://doi.org/10.1002/2014JD022361, 2014.

Lipenkov, V. Y., Raynaud, D., and Loutre, M. F.: On the potential of coupling air content and O2/N2 from trapped airfor establishing an ice core chronology tuned on local insolation, Quat. Sci. Rev., 30, 3280-3289, https://doi.org/10.1016/j.quascirev.2011.07.013, 2011.

Neff, P. D.: A review of the brittle ice zone in polar ice cores, Ann. Glaciol., 55, 72-82, 10.3189/2014AoG68A023, 2014.

Oyabu, I., Kawamura, K., Buizert, C., Parrenin, F., Orsi, A., Kitamura, K., Aoki, S., and Nakazawa, T.: The Dome Fuji ice core DF2021 chronology (0-207 kyr BP), Quat. Sci. Rev., 294, https://doi.org/10.1016/j.quascirev.2022.107754, 2022.

Picard, G., Domine, F., Krinner, G., Arnaud, L., and Lefebvre, E.: Inhibition of the positive snow-albedo feedback by precipitation in interior Antarctica, Nature Climate Change, 2, 795–798, https://doi.org/10.1038/nclimate1590, 2012.

Raynaud, D., Yin, Q., Capron, E., Wu, Z., Parrenin, F., Berger, A., and Lipenkov, V.: Past Antarctic summer temperature revealed by total air content in ice cores, EGUsphere [preprint], https://doi.org/10.5194/egusphere-2023-2360, 2023.

Severinghaus, J. P. and Battle, M. O.: Fractionation of gases in polar ice during bubble close-off: New constraints from firn air Ne, Kr and Xe observations, Earth and Planetary Science Letters, 244, 474–500, https://doi.org/https://doi.org/10.1016/j.epsl.2006.01.032, 2006. Yan et al., 2023.

---

## Author Comment (AC3)

**Response to Referee #2, Jochen Schmitt**

We are grateful to Jochen Schmitt for his time and effort in providing valuable feedback on the manuscript. These constructive comments have contributed to the improvement of the study. We have addressed all comments below and propose to implement the changes in a revised version of the manuscript.

Black = reviewer comment

blue = author's response / *"italic" = revised text.*

Review for Harris Stuart et al. 2023: ***Towards an understanding of the controls on δO2/N2 variability in ice core records*** by Jochen Schmitt, Bern

The paper, led by Romilly Harris Stuart, aims to improve our process understanding of a key ice core parameter - the O2/N2 ratio - that is used to orbitally date Antarctic ice core records. Decades ago it was discovered that the O2/N2 ratio resembles the orbitally-controlled solar insolation at the drill site. Already at that time, it was speculated how the upper firn layer is modulated by the amount of sunlight during summer. To affect the archived O2/N2 ratio in the bubbles, firn surface properties need to travel through the firn column to influence gas-specific (size-dependent) gas loss processes during the pore closure at the bottom of the firn. Over the years, a large number of studies suggested ideas to explain the observed O2/N2 variations, but we still lack an overall process understanding. While insolation apparently contributes a large fraction of the measured O2/N2 ratio, local temperature and accumulation rate modulate the orbital signal and lead to noise and uncertainty in the orbital tuning. At this target, Harris Stuart et al. place their study, which consists two approaches. Their study contributes to an important question relevant to the readers of The Cryosphere. Mostly, the paper is written clearly and provides the right depth of information and the figures are well-crafted and provide a visual support for the text. Overall, I support the publication of this study after minor revisions.

Their first approach is to apply an existing snowpack model to see if and to what extent differences in the solar radiation lead to changes in firn properties that might explain the observed O2/N2 ratios. Since the snowpack model was originally designed for alpine firn, applying it to low accumulation sites in Antarctica sets limitations. The authors became well aware of several limitations of their model (1-dimension, no wind compaction, merging box design) and thus interpreted their results with care. I got the impression that they used the model as much as it was possible for this study and then realized that no further insight could be obtained with this setup and that the model would need a significant improvement to capture the situation of low accumulation sites.

Their second approach is data-based and it was certainly a large effort to collect and screen all available O2/N2 records. The screening and data evaluation of the different cores and measurement campaigns is an important step and it would be crucial to provide a figure or two to allow the reader to see and understand the underlying problems of that step. Since it likely took a long time to collect all the records it would be helpful for others and the next generations of scientists to have easy access to these data sets and their meta information. So please spend some hours (perhaps more realistically days) of your time to bring all these data sets to a public database (both the already published and the new data). The analyses done on these 14 selected ice core sites conclude that factors other than insolation (accumulation rate and local temperature) have a sizable effect on the observed O2/N2 records and set limits to the precision and accurate of orbital tuning. This is a valuable outcome, but I feel that - in an ideal world with more time and resources - more can be done to disentangle the interplay between accumulation rate and temperature. As for the length of the diffusive firn column (i.e. d15N-N2), it might be the location on an accumulation vs temperature plot that determines if the firn column gets longer or shorter, or if the grain size within the first meter of firn increases or decreases. Since the temperature and accumulation rates are either known from present-day conditions or are output parameters of models (e.g. can be derived from delta age etc.), the team of this study might want to look a bit deeper into the interplay of temperature and accumulation rate in modulating O2/N2 ratios.

We would like to thank the reviewer for the general feedback and useful suggestions. A file containing all the published and unpublished $O_2/N_2$ data, accumulation rate reconstructions, temperature reconstructions, and SSI values will be published along with the paper. Regarding the disentanglement of accumulation rate and temperature effects, we propose to include an additional scatter plot in Figure 3 to show the relationship between accumulation rate and temperature and to facilitate a more comprehensive discussion. An updated version of Figure 3 is included in response to the reviewer's specific comments. Given the useful feedback from both reviewers, we propose to reorganise the discussion to enable a deeper look into the interplay between temperature and accumulation rate, and to develop the mechanistic links with $O_2/N_2$, while also improving readability.

I also wondered if more process understanding can be gained from analyzing the O2/N2 data from firn air studies. At least there should be some O2/N2 data from some drill sites available. The authors mention several times that one modulating factor of the O2/N2 imprint in the archived air bubbles is the degree by which the O2-enriched air that was expelled by the closing pores is advected upwards or diluted in firn. In other words, the O2/N2 fractionation during pore closure is only seen if it happens in an open system, ie. if the O2-rich air is removed from that layer. See e.g. lines 433 – 438. Perhaps using a full firn model that allows the simulation of permeability in the deep firn could help here?

This is a very valuable suggestion and would indeed contribute to the mechanistic understanding. There are data available from various drill sites which have been used to understand elemental fractionation during pore closure, alongside $O_2/N_2$ records from ice cores (e.g., Severinghaus and Battle, 2006). While we agree that analysing firn air data alongside ice core $\delta O_2/N_2$ records would be beneficial for the mechanistic understanding, it would require full representation of pore closure fractionation in firn models which is outside the scope of our study. For example, the ability of $O_2$-enriched air to be removed back to the atmosphere depends on deep firn properties, such as tortuosity and layering, which (to my knowledge) are not widely implemented in most firn models and as such, would be better suited for a future study. The purpose here is to present a new link between $\delta O_2/N_2$ and accumulation rate and temperature using the compiled data, and to identify how the surface forcing parameters modify snow properties. Rather than modelling the process, we aim to investigate the macro-scale mechanisms modulating the process. Future work will focus on combining firn air measurements with bubble ice measurements to better constrain the behaviour at distinct sites.

**Further suggestions and technical comments**:

Line 3: "trapped bubbles". I guess you want to say that the air in the bubble is sealed off from the open pore space; you can just say bubble since bubbles are closed anyway.

Yes, this is what we meant. This has been updated in the revised manuscript.

Line 4: write "... N2 molecules in extracted ice core air relative to the **modern** atmosphere - " Line 6: write "...and show a new additional link..." delete: ", in addition to the influence of the summer solstice insolation"

This has been modified in the updated version.

Line 8: "... forcings modulate snow physical properties near the surface "

Corrected in the revised version.

Line 10: "**a** mechanism**s**.."

Corrected in the revised version.

Line 16: firn...unconsolidated snow? Firn is the consolidated snow

Thanks for pointing this out. It has been corrected in the revised manuscript.

Line 18: rewrite "become sealed off from the firn air to form bubbles within the ice.

This will be re-written as:

"*seal off to form bubbles within the ice.*"

Line 18: "lock-in depth (LID)" actually you never use LID throughout the paper while you often use lock-in zone.

We propose to update the text to include a definition of lock-in zone.

*"Atmospheric air moves through porous networks within the firn until a critical depth (known as the lock-in depth) where vertical diffusion effectively stops, and pores gradually become closed off from the atmosphere. The lock-in depth and the depth at which all pores are closed (close-off depth) are largely determined by local accumulation rate, temperature, and possibly the degree of density layering (Schwander et al., 1997; Martinerie et al., 1994; Mitchell et al., 2015). The region between the lock-in depth and close-off depth in known as the lock-in zone (LIZ)."*

Line 22: komma after sites?

Added to the revised version.

Line 21 to 28: perhaps restructure this a bit. Essentially you describe two different kinds of dating approaches. O2/N2 and TAC are due to local effects of the firn column, thus these parameters are highly site-specific. On the other hand, d18O of O2 is a globally mixed atmospheric gas parameter that is not site-specific, and all ice cores yield the same record. Thus it can be used to wiggle-match different records but also relate the record to a certain orbital parameter.

Would be good to mention these two different approaches. Ideally, you could mention that d18O2 is used to date the gas phase of the ice core while O2N2 is an ice age parameter.

The paragraph introducing the dating techniques will be modified as written below.

*"Measurements of entrapped air can be used to reconstruct past atmospheric compositions, as well as to date the ice cores. One such dating technique - used primarily for deep ice cores from low accumulation sites - is orbital dating, which uses insolation curves at a given latitude directly calculated from astronomical variables (Laskar, et al., 2004). Such techniques can be used to construct gas-age chronologies by utilising the dependence of $\delta^{18}O$ of atmospheric $O_2$ ($\delta^{18}O_{atm}$) and precession (mid-June 65°N insolation) (Extier et al., 2018), and ice age chronologies based on the anti-phase relationship between total air content and integrated summer insolation (Raynaud et al., 2007), and $\delta(O_2/N_2)$ and summer solstice insolation (Kawamura et al., 2007; Suwa and Bender, 2008a; Landais et al., 2012; Bouchet et al., 2023). While $\delta^{18}O_{atm}$ is a direct atmospheric signal and may ultimately be used to align different ice core records, TAC and $\delta(O_2/N_2)$ are the result of processes within the firn column making the records site specific. The term $\delta(O_2/N_2)$ – hereafter, simply $\delta O_2/N_2$ - describes the relative difference between the ratio of $O_2$ to $N_2$ molecules trapped within the ice and that of the standard atmosphere and is expressed in the delta notation commonly used for stable isotope ratios."*

Line 28: delete "trapped within the ice" so it gets a bit more general

Corrected in the revised version.

Line 30/31: delete "vice versa"

This has been deleted.

Line 31: you could delete "numerous" as you already name quite a few sites...

Agreed, this has been corrected.

Line 34/35: you might rewrite this to convey that the modification due to insolation happens at the snow surface but the process that effectively alters the archived O2N2 ratio happens at the depth where the pores close off

We agree that this description can be improved and propose the text below to replace sentences in line 34-35. These updates require modifications to the paragraph starting line 46.

*"Over orbital timescales, $\delta O_2/N_2$ is in antiphase with local SSI when drawn on the ice-age time scale, which led Bender (2002) to discern that firn properties, containing an SSI signal retained throughout the firn column, modulate the fractionation of $O_2/N_2$ during pore closure. He, followed by many others, proposed that strong summer insolation drives temperature gradient metamorphism, thus increasing near-surface grain size which propagates through the firn during the firnification process down to the close-off depth (Bender, 2002; Severinghaus and Battle, 2006; Suwa and Bender, 2008a; Fujita et al., 2009)."*

Line 37: (COD) is just used twice ...just write it out in both cases

COD has been changed to close-off depth in both instances.

Line 40: replace ; with :

Corrected in the revised version.

Line 41: Why cite also her first name Tomoko?

There was a mistake in the BibTex file which has now been corrected.

Line 48: Why "They"? you refer to Bender (2002) so technically just Michael Bender although he acknowledges at the end of his paper that he profited a lot from the discussion with many giants in this field

This was an oversight, and "They" has been changed to "He".

Line 85: WAISD would be a new abbreviation, commonly used is WD or WAIS

The abbreviation has been changed to WAIS throughout the manuscript.

Line 102 Table 1 (and other tables): for better visibility please align numbers in columns on the right side, e.g. Table 3 in Petrenko et al. 2016 http://dx.doi.org/10.1016/j.gca.2016.01.004

The tables will be modified accordingly in the updated manuscript.

Line Table 1: If possible and available please also add other site characteristics to this table, e.g. close-off depth or ice age at close-off depth (delta age) they might be useful as well

We propose to include present-day close-off depth (or lock-in depth) for all available sites.

Line 157: "gas loss during coring", can you explain a bit more here?

We acknowledge that this is unclear. Here we should really only refer to gas loss effects during storage in reference to $O_2$ depletion through time. The idea of gas loss during coring comes from the apparent enrichment in $O_2$ in the bubble phase compared to clathrates, and therefore, we propose to instead mention this in the following section (Section 2.2.3) alongside the BCTZ.

Line 163: Note that the brittle zone does not always correspond to the BCTZ, while for most of the ice cores, this is the case. I guess some ice cores have a technically defined brittle zone while they do not have the conditions to form clathrates at a certain set of depth or temperature; thus, without this coexistence of clathrate and bubbles, there shouldn't be a strong fractionation. Perhaps elaborate shortly on that.

Line 165: O2/N2 measurements within the brittle (or BCTZ) ice are not per se unreliable; it requires a post-coring gas loss, so the fractionated air in the bubbles escapes and thus induces scattered results. Also, small sample sizes resolve individual layers of bubbes vs clathrates

Line 167: see above, does Berkner have a chlathrate zone? Perhaps this explains good data within the brittle zone.

(In response to the previous three comments combined) Indeed, Berkner does not appear to have a clathrate zone (Schaefer et al., 2011), so this may explain the good data. Information in Table 1 (and corresponding text) now uses BCTZ instead if brittle zone. The majority of measurements within the published 'brittle zones' were indeed also the BCTZ, with the exception of Berkner Island, Siple Dome, Roosevelt Island, and Law Dome (Neff, 2014). We propose to include the following additional text to improve clarity.

*"At the bubble-clathrate transition zone (BCTZ: where the high hydrostatic pressure in the bubbles cause entrapped gases to form clathrate hydrates (Schaefer et al., 2011)) elemental fractionation occurs due to some gas species being preferentially incorporated into the clathrate structures (Ikeda-Fukazawa et al., 2001), thus making the interpretation of gas measurements unreliable at these depths (Bender, 2002)."*

Line 176: post-coring gas loss to differentiate between the gas loss happening during pore closure in the ice sheet

To clarify that we refer to post-coring gas loss we will include the following text:

*"The storage histories of the ice need to be considered before interpreting the data to account for post-coring gas loss effects which disturb the signal (Section 2.2.2)."*

Line 214: you mention the black carbon content. How sensitive is the model to the black carbon? What about a similar effect of mineral dust during glacial times (OK, mineral dust is mostly light quartz but there are also darker particles...)

The model is sensitive to black carbon content, as shown by Libois et al. (2013). We did investigate the link between the black carbon loading and the surface snow temperature discrepancy (compared to measured temperature profiles) but found that the simulated temperature profile with depth was best represented using the ascribed value (3 ng/g; Libois et al., 2015). The radiative transfer scheme we used in Crocus, the Two-streAm Radiative TransfEr in Snow model (TARTES; Libois et al., 2013), also has an option to include dust content loading. We did not include dust as the effect of black carbon is much more potent than that of the dust particles (~50 time more absorptive: Warren et al., 2006). This is definitely a useful comment and would be important to consider for realistic simulations of glacial and interglacial conditions.

Line 277: are the dots after the permil and the m2 correct?

This mistake has been corrected.

Line 282: "integrated summer insolation": can say a few words on the difference between integrated summer insolation and SSI and why you use SSI?

$\delta O_2/N_2$ and total air content have slight differences in their spectral signature; $\delta O_2/N_2$ records are dominated by precession, while total air content records are dominated by obliquity (e.g. Lipenkov et al., 2011). Spectral analysis shows that integrated summer insolation (the annual sum of daily insolation above a certain threshold) is driven by obliquity, whereas SSI is driven by precession (e.g. Huybers 2006). Therefore, when investigating the mechanisms driving $\delta O_2/N_2$ we use SSI. This will be incorporated into the updated manuscript.

Line 290: Figure 2 caption: you don't need to say that Dome C is plotted in dark-blue and Dome F in mid-blue because you can indentify each panel with their name already. Please add a), b), c) as you do in Fig. 3

The labelling of Figure 2 has been modified accordingly.

Line 290: Figure 2 caption: why do you use r2 here, while in Fig. 3a, you use r for the same type of plot? perhaps always use r (as r2 can be calculated from that)

Both figures now show r, p-values, and the slope.

Line 312: Figure 3: I very much like your colour scheme, but here, it would also help to provide more visual hints to distinguish between some sites, e.g. LD and BI have quite similar colours (same for NEEM and WAIS). You could additionally use squares and diamonds.

As suggested, the markers for records from Greenland have been changed to squares.

Line 328: Table 3: The 5 EDML samples (596 – 860) are from the brittle zone. Are there no other samples measured at EDML, why just in the brittle zone?

In general, the $\delta O_2/N_2$ data compiled for our study are a biproduct from measurements of $\delta^{15}N$ or $\delta^{18}O_{atm}$. Therefore, the depth range of measured data was not chosen to assess $\delta O_2/N_2$.

Line 331: Fig. 4: Would the residuals look different if it would be plotted on the AICC2023?

This is a very useful point and both records are now presented on the AICC2023 chronology. The results are unchanged when applying the same analysis as in the original version of the manuscript (please see Figure A below). However, it was suggested by Reviewer #1 to instead remove the SSI signal by applying a high pass filter to the $O_2/N_2$ and $\delta D$ records. This has, of course, removed any 20 kyr variability observed in the original analysis, thus, reducing the strength of correlation, but the coherence in millennial-scale peaks remain.

[Figure]

**Figure A**. Comparison between the evolution of $\delta O_2/N_2$, SSI, and $\delta D$ from EDC on the AICC2012 ice age chronology (light colours: Bazin et al., 2013), and on the AICC2023 chronology (dark colours: Bouchet et al., 2023). Panel a) presents the $\delta O_2/N_2$ (blues) with SSI on the right y-axis (grey), b) $\delta D$ (green), and c) the $\delta O_2/N_2$-SSI residuals (orange), between 200 and 260 ka BP. Scatterplots show the correlation between the $\delta O_2/N_2$-SSI residuals and $\delta D$ using d) AICC2012 chronology, and e) the AICC2023 chronology.

Line Figure. 4 caption: the respectively structure always requires the reader to go to the end of the sentence while the classical way "Panel d shows the correlation" is often quicker to access

The caption for Figure 4 will be changed accordingly in the revised manuscript.

Line 354 Figure 5d: since there is no overlap between Jan and Jul, you could put both distributions into a single panel

This has been modified in the updated manuscript.

Line 381 Figure 7: it is not easy to see the difference between the faded line and the max line, perhaps increase the thickness of the line or use dashed lines etc

The contrast is increased between the faded and max lines in updated figure.

Line 399: the long list of references affects a bit the readability ....not sure if you need all the references here in the discussion section, perhaps write e.g. and two refs

The list of references has been replaced with the two most recent publications.

"*Our compilation of different deep ice core $\delta O_2/N_2$ records show the widely documented anti-correlation between SSI and $\delta O_2/N_2$ on the ice-age scale (e.g. Oyabu et al., 2021; Bouchet et al., 2023).*"

Line 406: Fig. 3c, I guess you mean Fig. 3a showing as well O2/N2 vs SSI while Fig. 3c shows temperature. Where is the slope for Fig. 3a to compare it with the slopes of Fig. 2?

Many thanks for this pointing this out. This has been corrected and the slopes of each panel in Fig 3 have been added for comparison.

Line 438-440: I am not so sure if this argument holds that the O2/N2 signal would then be on the gas age scale. Still it happens in the lock-in zone due to a process that was imprinted originally at the surface.

We agree that the lock-in zone characteristics will be set near the surface, but the residence time of a gas in the lock-in zone may be modified by changes at the surface such as to overburden pressure, hence, containing an integrated signal over the firn column. However, we acknowledge that this is overlapping the transient effects described by Eicher et al. (2016) and is therefore a different mechanism. As mentioned in response to general comments, we propose to modify the discussion which involves integrating Section 4.2 (a general review of mechanisms) into the interpretation of our results. As such, this argument will be corrected in the updated manuscript and used to support interpretations of the positive correlation between $\delta O_2/N_2$ and accumulation rate.

Line 490: I am puzzled a bit about the term bulk ice...

The term "bulk ice" is used here to incorporate multiple ice layers with potentially different characteristics. We propose to modify this sentence for clarity to the following:

*"Based on the results presented in our study, the anti-correlation between SSI and $\delta O_2/N_2$ is coherent with an increase in near-surface grain size for a given density, ultimately leading to a decrease $\delta O_2/N_2$."*

Line 490: "The opposite – a decrease" not sure if this sentence describing the opposite effect is necessary I guess the sensitivity of grain size for a given density works in both directions

We agree, and this sentence has been removed for the revised manuscript.

Line 493: yes, temperature and accumulation rate do generally covary, but they are not super tightly correlated, and there are sites that are above or below the expected line for the temperature– accumulation relationship. Perhaps you can derive some useful information from the deviations from this temperature-accumulation relation, i.e. a site that has too little accumulation rate for a given temperature. A scatter plot showing all sites with their accumulation vs temperature might help to identify sites that deviate from others in Figure 3. This requires O2/N2 data for the present-day conditions for accumulation and surface temperature that are likely more accurate than the reconstructed values based on modelling via water isotopes.

Many thanks for this useful suggestion. Given that we currently do not have a lot of modern $\delta O_2/N_2$ data from ice cores, we propose to include a scatter plot of reconstructed temperature vs accumulation rate to Figure 3, as suggested. A version of the updated Figure 3 is shown below. Sites from Greenland are represented by squares to distinguish them from Antarctic sites, as suggested in a previous comment. The following text is added to results Section 3.1.1, along with a draft of the updated Figure 3:

*"Berkner Island and Siple Dome fall below the regression line in Fig 3c and 3d, potentially indicating gas loss. Indeed, the measurements from the Siple Dome core were carried out ~8 years after coring, and the values become more depleted with depth. While it has been reported that the Sipe Dome core does not contain clathrate ice (Neff, 2014), the low values may link to gas loss during storage within the more brittle ice at depth in the core. Similarly, measurements from the Berkner Island core were performed on bubbly ice ~7 years after coring and fall within the depths of reported brittle ice (Mulvaney et al., 2007)."*

[Figure]

**Figure B** (Figure 3 in manuscript). Scatterplots showing the relationship between $\delta O_2/N_2$ and surface conditions for each site. Panel a) shows the temperature-accumulation rate relationship, followed by three plots presenting the dependence of $\delta O_2/N_2$ on the b) SSI, c) accumulation rate (A), and d) annual average temperature ($T_{air}$). Each point represents the mean values for each site over the depth interval of included $\delta O_2/N_2$ data for each site (Table 3). Error bars represent the range of values over the depth interval. Error bars on y-axis show the standard deviation of $\delta O_2/N_2$ measurements. Linear regressions are show in black, along with the associated correlation coefficient (r) and p-value. (Note any slight differences in mean $\delta O_2/N_2$ comes from adjustments in the brittle zone definition.)

We will also add some additional text to the discussion Section 4.1:

*"Inter-site analysis reveals that mean $\delta O_2/N_2$ is strongly correlated with accumulation rate and temperature for a given site. While mean $\delta O_2/N_2$ versus SSI falls on the same regression slope as the average of the temporal slopes for EDC, Dome F, and South Pole (Fig 2), the anti-correlation is very weak, suggesting that mean $\delta O_2/N_2$ at a given site is determined mostly by the local accumulation rate and temperature.*

*Deviations from inter-site temperature-accumulation rate relationship (Fig 3a) may explain anomalies in mean $\delta O_2/N_2$ (Fig 3c and 3d) given that such deviations would influence firn characteristics at the site that ultimately modulate $O_2/N_2$ fractionation. For example, compared to a site falling directly on the regression line, it may be expected that sites with a relatively high accumulation rate for their temperature would have higher $\delta O_2/N_2$ values due to increased snow burial rate near the surface resulting in smaller grain size (Vionnet et al., 2012). Indeed, this is supported by a relative decrease in grain size under increased accumulation rate in our sensitivity tests (A min), and by relatively high $\delta O_2/N_2$ at South Pole corresponding to a relatively high accumulation rate for its temperature. In contrast, such sites would also have a deeper lock-in depth leading to greater fractionation (Severinghaus and Battle, 2006). Our study cannot test the latter, but the relatively increased $\delta O_2/N_2$ at South Pole compared to EDC and Dome F (Fig 2) supports the grain size hypothesis."*

Finally, a contour plot indicating the expected $\delta O_2/N_2$ values as a function of temperature and accumulation rate may also be included to assist the interpretation. Here we reiterate that mean $\delta O_2/N_2$ is particularly sensitive to accumulation rate, especially at low accumulation sites. In contrast, even rather major changes in temperature would have little influence on $\delta O_2/N_2$ at low accumulation sites.

[Figure]

**Figure C**. Contour plot of $\delta O_2/N_2$ as a function of temperature and accumulation rate. Contours are calculated based on the multiple linear regression of $\delta O_2/N_2$ from accumulation rate (A) and temperature ($T_{air}$) and their coefficients ($\delta O_2/N_2 = 2.78 \log(A) + -0.03 T_{air} -13.76$). Annotated values state the mean $\delta O_2/N_2$ for each site, indicated in the legend.

Line 500: this sentence is a bit unclear to me

The first sentences of Section 4.3.2 may be modified to the following:

*"Density stratification in deep firn has also been invoked to modulate $\delta O_2/N_2$ variability (Fujita et al., 2009). Results from the Crocus model are used to infer the sensitivity of near-surface density and grain size variability to perturbations in input forcing parameters."*

Line 545: could you spend a few words on how the local SSI at EDC is linked to local accumulation rates since this is a larger-scale weather phenomenon and involves low-pressure systems entering the continent, etc.? Perhaps elaborate a bit on that?

The purpose of this sentence was intended to reiterate the covariance between temperature and accumulation rate based on the Clausius-Clapeyron relationship, but also to acknowledge that local temperature is not completely independent from SSI (Yan et al., 2023). We did not intend to suggest that local SSI is directly linked to local accumulation rate, and we propose to modify the sentence in line 544-545 as follows:

*"While the single-parameter sensitivity tests presented here provide useful insights for understanding physical mechanisms, they do not account for the expected complex compound effects associated with the covariance of accumulation rate and temperature."*

Line 575: "local climate (accumulation rate" I understand what you mean but accumulation rate might be largely determined by the circulation patterns in the Southern Ocean region.

To avoid confusion, we propose to change the sentence in line 575 to:

*"Analysis of both spatial (multi-site) and temporal (single site) variability in $\delta O_2/N_2$ revealed new evidence of a dependence on accumulation rate and temperature, in addition to the well-documented insolation dependence."*

Line 576: I guess this statement also holds for AICC2023 (although there seems to be a small circularity hidden into that because the age scale is constructed using the O2/N2 orbital tuning)

This statement is still true when using the new AICC2023 chronology. To avoid the circularity (and discrepancies in the chronology), we confirmed the coherence in millennial-scale variability by comparing $\delta O_2/N_2$ and $\delta D$ on the depth scale, shown below. The timings directly align giving us confidence in our results.

[Figure]

**Figure D**. $\delta O_2/N_2$ and $\delta D$ plotted on the depth scale from the EDC core (Bouchet et al., 2023; Landais et al., 2021).

Line 582: this sentence misses some words...support the idea...

This sentence now reads:

*"Our findings support the hypothesis that a grain size mechanism is the dominant driver of elemental fractionation at low accumulation sites, such that increased grain size for a given density facilitates $O_2$ expulsion via enhanced permeability."*

**References**

Bazin, L., Landais, A., Lemieux-Dudon, B., Toyé Mahamadou Kele, H., Veres, D., Parrenin, F., Martinerie, P., Ritz, C., Capron, E., Lipenkov, V., Loutre, M.-F., Raynaud, D., Vinther, B., Svensson, A., Rasmussen, S. O., Severi, M., Blunier, T., Leuenberger, M., Fischer, H., Masson- Delmotte, V., Chappellaz, J., and Wolff, E.: An optimized multi-proxy, multi-site Antarctic ice and gas orbital chronology (AICC2012): 120ndash;800 ka, Climate of the Past, 9, 1715–1731, https://doi.org/10.5194/cp-9-1715-2013, 2013.

Bouchet, M., Landais, A., Grisart, A., Parrenin, F., Prié, F., Jacob, R., Fourré, E., Capron, E., Raynaud, D., Lipenkov, V. Y., Loutre, M.-F., Extier, T., Svensson, A., Legrain, E., Martinerie, P., Leuenberger, M., Jiang, W., Ritterbusch, F., Lu, Z.-T., and Yang, G.-M.: The Antarctic Ice Core Chronology 2023 (AICC2023) chronological framework and associated timescale for the European Project for Ice Coring in Antarctica (EPICA) Dome C ice core, Clim. Past, 19, 2257–2286, https://doi.org/10.5194/cp-19-2257-2023, 2023.

Eicher, O., Baumgartner, M., Schilt, A., Schmitt, J., Schwander, J., Stocker, T. F., and Fischer, H.: Climatic and insolation control on the high-resolution total air content in the NGRIP ice core, Climate of the Past, 12, 1979–1993, https://doi.org/10.5194/cp-12-1979-2016, 2016.

Peter, H.: Early Pleistocene Glacial Cycles and the Integrated Summer Insolation Forcing, Science, 313, 508-511, https://doi.org/10.1126/science.1125249, 2006.

Landais, A., Stenni, B., Masson-Delmotte, V. et al. Interglacial Antarctic–Southern Ocean climate decoupling due to moisture source area shifts. Nature Geoscience, 14, 918–923, https://doi.org/10.1038/s41561-021-00856-4, 2021.

Libois, Q., Picard, G., France, J. L., Arnaud, L., Dumont, M., Carmagnola, C. M., and King, M. D.: Influence of grain shape on light penetration in snow, The Cryosphere, 7, 1803–1818, https://doi.org/10.5194/tc-7-1803-2013, 2013.

Libois, Q., Picard, G., Arnaud, L., Dumont, M., Lafaysse, M., Morin, S., and Lefebvre, E.: Summertime evolution of snow specific surface area close to the surface on the Antarctic Plateau, The Cryosphere, 9, 2383–2398, https://doi.org/10.5194/tc-9-2383-2015, 2015.

Lipenkov, V. Y., Raynaud, D., and Loutre, M. F.: On the potential of coupling air content and O2/N2 from trapped airfor establishing an ice core chronology tuned on local insolation, Quat. Sci. Rev., 30, 3280-3289, https://doi.org/10.1016/j.quascirev.2011.07.013, 2011.

Neff, P. D.: A review of the brittle ice zone in polar ice cores, Ann. Glaciol., 55, 72-82, https://doi.org/10.3189/2014AoG68A023, 2014.

Schaefer, H., Lourantou, A., Chappellaz, J., Lüthi, D., Bereiter, B., Barnola, J-M.: On the suitability of partially clathrated ice for analysis of concentration and δ13C of palaeo-atmospheric CO2, Earth and Planetary Science Letters, 307, 3-4, 334-340, https://doi.org/10.1016/j.epsl.2011.05.007, 2011.

Warren, S. G., Brandt, R. E., and Grenfell, T. C.: Visible and near-ultraviolet absorption spectrum of ice from transmission of solar radiation into snow, Appl. Opt., 45, 5320–5334, https://doi.org/10.1364/AO.45.005320, 2006.

Yan, Y., Kurbatov, A.V., Mayewski, P.A., Shackleton, S., Higgins, J.: Early Pleistocene East Antarctic temperature in phase with local insolation. Nature Geoscience, 16, 50–55, https://doi.org/10.1038/s41561-022-01095-x, 2023.

---

## Author Response (AR1)

**Response to anonymous Referee #1**

**Harris Stuart et al.: Towards an understanding of the controls on $\delta O_2/N_2$ variability in ice core records**

We are grateful to the reviewer for their time and effort in providing valuable feedback on the manuscript. These constructive comments have contributed to the improvement of the study. We have addressed all comments below and have now implemented these changes into the revised version of the manuscript.

In the following responses, the line numbers refer to those in the revised manuscript. Figures which are numbered refer to figures in the revised paper and supplement, while figures ordered by letters are included in this response only.

Black = reviewer comment
blue = author's response / *"italic" = revised text.*

This paper presents the relationship between environmental conditions of the ice sheet surface and $O_2/N_2$ fractionation in combination with $O_2/N_2$ records from 14 ice cores and results from a snowpack model. The relationship between $O_2/N_2$ and local summer solstice insolation is well known, while the physical mechanism to create the insolation signal is poorly understood. The paper provides new $O_2/N_2$ data from Greenland and Antarctica. It qualitatively demonstrates the role of accumulation rate, surface temperature, and solar radiation on surface snow properties with the snowpack model, which contributes to understanding the mechanism of close-off $O_2/N_2$ fractionation. Overall, the subject is appropriate to *The Cryosphere*. However, there are several points which require some major revision. It appears that there are several areas with insufficient explanations throughout the manuscript.

**General comments:**

1. The analysis investigating the relationship between $O_2/N_2$ and SSI, A, or T (section 3.1) and the sensitivity experiments with the snowpack model (section 3.2) provide new results that deserve a thorough examination. In the current manuscript, the model results are well discussed, while the discussion connecting the observation of $O_2/N_2$ and the model results (i.e., surface conditions) is weak. The relationship between grain size and $O_2/N_2$ is not well described. I understand that the focus of this study is not to investigate how the surface snow conditions affect the densification processes/metamorphisms in the deeper firn, but readers may expect that the discussion would address the link between surface snow metamorphism and subsequent firn properties and how they are related to $O_2/N_2$ Section 4.1 contains a recitation of the results and gives an example of Greenland, but the main argument of this section is unclear. Section 4.2 is almost entirely a review of previous studies. In Section 4.3, I expect the author to develop their discussion here. The model results are discussed, but the arguments connecting to $O_2/N_2$ are unclear. In the revised manuscript, I expect to add some discussion connecting the results of Sections 3.1 and 3.2, for example, by introducing arguments from previous studies such as Fujita et al. (2009) and Hutterli et al. (2009). See also my specific comments.

Many thanks to the reviewer for this useful suggestion. We acknowledge that the discussion connecting the $\delta O_2/N_2$ and snowpack modelling results ought to be further developed. Given the useful feedback from both reviewers, we have reorganised the discussion to enable a deeper look into the interplay between temperature and accumulation rate, and to develop the mechanistic links with $\delta O_2/N_2$. Modifications to discussion are included in response to specific comments, and we point out that the neighbouring discussion sections have been modified in the revised manuscript to accommodate the new text.

2. I am not satisfied with the analysis of the link between $O_2/N_2$ and water isotope ratios of the EDC core, which is one of the bases for the discussion linking the $O_2/N_2$ fractionation mechanism with the

model results. First, the authors found positive correlations between δD and O₂/N₂ in the EDC core, but they are all separated by relatively short time periods where the amplitudes of the 20 kyr cycle of δD are large (interglacials). Therefore, it seems to be not surprising to find correlations within those specific periods because O₂/N₂ has a strong precession component. The absence of a positive correlation in MIS11 may be because this interglacial period has a length of 2 precession cycles. If there is indeed a correlation between O₂/N₂ and T and/or A, a positive correlation would be expected even in MIS11. What would happen if correlations were examined over longer continuous periods, including periods with a smaller amplitude of 20 kyr cycle, such as MIS6? The validation may be possible by using data from the South Pole and Dome Fuji cores.

It is true that the high-resolution measurements from the EDC core are mainly during deglaciations and interglacial periods, but this was actually the objective of the study. We wanted to see if during the important local change in temperature, such as during deglaciations and associated sub-orbital variability (e.g. linked to Heinrich events), there is an δO₂/N₂ signal which is not only linked to local summer solstice insolation since δD itself is not in phase with change in local solstice insolation at EDC (e.g. Raynaud et al., 2023).

We agree that it would be beneficial to apply this analysis over the entire core, but we are limited by the availability of high-resolution data leading to over-sampling issues (please see Fig. A below). The entire time series are presented in the figure accompanying the comment after next to show the non-uniform measurement resolution. The key observation here is the coherent millennial-scale variability in δD and δO₂/N₂ from the EDC core which we cannot identify in low-resolution records. The same analysis applied to periods of high-resolution data from Dome Fuji and South Pole (included below and in the Supplement) highlights the site dependence of millennial-scale δO₂/N₂ variability. As we mention below, it appears that sub-orbital variability is linked to accumulation rate such that different mechanisms are influencing δO₂/N₂ at high accumulation sites. This aligns with previous studies reporting alternative phasing between δO₂/N₂ and temperature (accumulation rate) potentially depending on the climate state and time scale considered. For example, Kobashi et al. (2015) show an inverse correlation between δAr/N₂ (on gas-age scale) and temperature at GISP2 over the last 6000 years, whereas a positive correlation was observed by Suwa and Bender (2008) at GISP2 during Dansgaard-Oeschger events. Our response to the reviewers following comments will further address the millennial-scale variability.

[Figure]

**Figure A.** δO₂/N₂ over the last 800 ka from EDC. The low-resolution data, shown in grey, has an average resolution of around 2500 years and therefore cannot be used to assess millennial-scale variability (Landais et al., 2012; Bazin et al., 2016; Extier et al., 2018). On the other hand, the relatively high-resolution data, shown in blue, has an average resolution of between 300-400 years (Bouchet et al., 2023). Periods of high-resolution data are limited and thus we focus on the longest period available (200-200 ka BP).

Second, if the authors want to investigate whether the 1000-year scale variations shown in grey shadings in Fig. 4 are related to the O₂/N₂ signal, it is necessary to remove orbital variations with appropriate methods. For this purpose, I suggest applying the low-pass filter used in orbital tuning to both δD and O₂/N₂ and extracting residuals from low-pass filtered curves, or applying a high-pass filter to δD and O₂/N₂ to cur off the insolation frequency. The current approach can introduce artificial variations due

to the potential mismatch between the insolation and ice core ages. If the AICC2012 age scale is perfect in terms of absolute age, the long-term variation in Fig. 4b is real. However, as Extier et al. (2018) pointed out, the AICC2012 chronology tends to deviate from the U-Th chronology, especially during deglaciations. This raises the question about the accuracy of this chronology. Therefore, the orbital-scale variation shown in Fig. 4b may just reflect the phase difference between the AICC2012 chronology and insolation. Using the most recent AICC2023 chronology aligned with the U-Th chronology over the last ~600 kyr (Bouchet et al., 2023), or the DF2021 chronology (Oyabu et al., 2022) synchronized with the local summer insolation, could potentially alter the appearance of the residuals in $O_2/N_2$.

We thank the reviewer for raising this and appreciate the suggestions. The aim was not necessarily to remove all orbital variations but to investigate the non-SSI drivers of $\delta O_2/N_2$ variability. We do acknowledge that separating the SSI signal from orbital-scale local climate variability is not trivial and this is why we now focus on the millennial-scale variability. However, the fact that the results are unchanged with the ACC2023 chronology suggests a climate signal in the EDC $\delta O_2/N_2$ record. Our response to the reviewers later comment addresses the persistence of sub-orbital scale variability when using different chronologies.

Therefore, I suggest to conduct the following analyses, and add the results and corresponding discussion as needed.

1. Apply a low-pass filter to both $\delta D$ and $O_2/N_2$ and extract residuals from low-pass filtered curves to eliminate the potential discrepancy between the ice core and insolation ages. Alternatively, apply a high-pass filter to cut off the insolation signal.

The suggestion from the reviewer to apply a filter is very useful. A low pass filter is applied to the 100-year interpolated $\delta O_2/N_2$ and $\delta D$ records with a 10-kyr cut off to isolate low frequency signals. The filtered curves are subtracted from the original curve to extract the high frequency (sub-orbital scale) signals, and both curves are smoothed using a 5-point moving average to reduce noise. As in the original paper, we focus on the longest available period of relatively high-resolution data (between 200-260 ka BP) and reach the same conclusion that rapid changes (on the order of 1000's of years) in $\delta O_2/N_2$ often correspond with changes in $\delta D$ (updated Fig. 4). The linear regression of the high pass filtered $\delta D$ vs high pass filtered $\delta O_2/N_2$ is much lower than we found from the original analysis (r = 0.3, p < 0.0001), but this is expected given the absence of any low-frequency coherence in the records.

As mentioned previously, the same analysis cannot be applied to the entire record given that the filtering requires interpolated data which leads to oversampling during periods with low measurement resolution. Unfortunately, the high-resolution measurements do not cover the entire of MIS11 (just the onset) and thus we are unable to fully assess the non-SSI variability in the $\delta O_2/N_2$ data. We do acknowledge that the coherence between $\delta O_2/N_2$ and $\delta D$ is not constant in time or space, but instead highlight the importance of considering this millennial-scale variability when using $\delta O_2/N_2$ for orbital tuning. Moreover, these non-orbital signals are useful to improve the mechanistic understanding. We have updated Section 3.1.2 to incorporate the additional analysis using the filtered curves.

Lines 299-313: *"Two approaches are used to extract the non-SSI signals in the $\delta O_2/N_2$ records. First, we interpolate SSI onto the $\delta O_2/N_2$ ages and take the deviations from the $\delta O_2/N_2$-SSI linear regression to isolate the $\delta O_2/N_2$ variability not explained by SSI (hereafter, $\delta O_2/N_2$-SSI residual). The second approach directly investigates the millennial-scale variability by applying a low pass filter to the $\delta O_2/N_2$ and $\delta D$ records (interpolated onto a 100-year time step), using a 10-kyr cut-off to isolate the low-frequency signals associated with SSI. The filtered curves are then subtracted from the original curves to remove the orbital (SSI) signal, and a 5-point moving average is applied to the residuals to reduce noise but retain millennial-scale variability in the record.*

*Figure 4 primarily shows the dominant SSI cyclicity in the $\delta O_2/N_2$ record, as has been documented previously (e.g., Landais et al., 2012). Superimposed onto this signal are millennial-scale peaks in*

*δO₂/N₂ which appear to coincide with peaks in δD, highlighted by grey bars. This high frequency variability is more clearly identified from the filtered curves in Fig. 4d and 4e. The coherence between millennial-scale peaks in δO₂/N₂ and δD suggests a positive correlation with accumulation rate and temperature, which shares analogy with the spatial positive correlation between δO₂/N₂ and both accumulation rate and temperature (Fig. 3). We note a remnant 20-kyr variability in the δO₂/N₂-SSI residual which is in phase with δD. This orbital-scale signal is present when using either the AICC2012 chronology or the AICC2023 chronology (Bazin et al., 2013; Bouchet et al., 2023). Given that the new AICC2023 age scale (used here) has been orbitally tuned using this data (Bouchet et al., 2023), this signal is not expected to be the result of phase difference between the EDC chronology and SSI."*

[Figure]

***Figure 4****. Evolution of δO₂/N₂, SSI and δD on the AICC2023 ice age timescale from EDC (Bouchet et al., 2023). Panel (a) presents the 100-year interpolated δO₂/N₂ overlain with the low pass filtered curve (blue). The right y-axis in panel (a) shown SSI at 75.1∘S (grey). The predicted δO₂/N₂ as a function of SSI (δO₂/N₂ -SSI predicted) is shown on the right axis of panel (b), and the residuals of the observed δO₂/N₂ and predicted are on the left axis. Panel (c) shows the 100-year interpolated δD overlain with the low pass filtered curve (green). The bottom two panels present the residuals of the low pass filtered curves for δO₂/N₂ (d) and δD (e). Grey bars highlight the pronounced millennial-scale variability.*

2.  Use the AICC2023 chronology to examine SSI-residuals.

Using the AICC2023 chronology instead of AICC2012 does not significantly change the results of the SSI-O₂/N₂ residual analysis from the original manuscript. This is shown in Fig. B below which compares Fig. 4 from the original manuscript with data drawn on the AICC2012 time scale to the same analysis with data on the AICC2023 timescale. The δD and SSI data were interpolated onto the δO₂/N₂ scale in both cases. Figure 4 in the revised manuscript includes both the δO₂/N₂-SSI residuals, as defined above, and the high-frequency analysis. The updated text and Fig. 4 are included in our response to the previous comment.

[Figure]

**Figure B**. Comparison between the evolution of δO₂/N₂, SSI, and δD from EDC on the AICC2012 ice age chronology (light colours: Bazin et al., 2013), and on the AICC2023 chronology (dark colours: Bouchet et al., 2023). Panel a) presents the δO₂/N₂ (blues) with SSI on the right y-axis (grey), b) δD (green), and c) the δO₂/N₂-SSI residuals (orange), between 200 and 260 ka BP. Scatterplots show the correlation between the δO₂/N₂-SSI residuals and δD using d) AICC2012 chronology, and e) the AICC2023 chronology.

3.  Use the Dome Fuji O₂/N₂ and water isotope data, and use the DF2021 chronology to examine SSI-residuals.

Applying the same analysis to the Dome Fuji records similarly reveals some coherence in millennial-scale variability between residuals from the low-pass filtered curve of δD and δO₂/N₂ records (Fig. S3 below). However, the δD maximum around 130 ka (corresponding to MIS 5) is not present in the δO₂/N₂ data, as may be expected considering findings from Kawamura et al. (2007). During the glacial periods there appears to be some coherence. However, the δO₂/N₂ record has been influenced by scattering effects below the BCTZ (Oyabu et al., 2021) making any interpretation difficult. Data older than 170 ka are also of too low resolution for analysis of millennial-scale variability. We have included the following text in the Supplement.

Lines 27-33 (Supplement): *"Figure S3 presents the data from Dome F, with the high frequency signals of δO₂/N₂ and δD in the bottom two panels. Data older than 170 ka are of too low resolution for analysis of millennial-scale variability. Between 100 ka BP and 120 ka BP the filtered δO₂/N₂ records from Dome F show a similar millennial-scale coherence to the filtered δD record. However, the rapid, high-frequency variability in δO₂/N₂ over this period may be attributed to scattering effects below the BCTZ, as was documented by Oyabu et al. (2021). The rapid δD peak around 130 ka BP (corresponding to MIS 5) is not present in the δO₂/N₂ data. The absence of a climate signal in δO₂/N₂ records from Dome F has been previously identified by Kawamura et al. (2007)."*

[Figure]

***Figure S3***. *Time series of δO₂/N₂, δD, and SSI on the DF-2021 ice-age chronology from Dome F (Oyabu et al., 2022a). Panel (a) presents the 100-year interpolated δO₂/N₂ overlain with the low pass filtered curve (blue). The right y-axis in panel (a) shown SSI at 77∘S (grey). Panel (b) shows the 100-year interpolated δD overlain with the low pass filtered curve (green). The bottom two panels present the residuals of the low pass filtered curves for δO₂/N₂ (c) and δD (d).*

4. If applicable, use the South Pole core $O_2/N_2$ and water isotope data, and use the SP19 chronology to examine SSI-residuals.

Links between $\delta O_2/N_2$ and $\delta D$ in the South Pole core highlight the site dependence of $\delta O_2/N_2$ variability. The $\delta O_2/N_2$ and $\delta D$ filtered curves appear to resemble each other to varying degrees throughout the record. It appears that the in-phase coherence is most pronounced when $\delta D$ is low (over the LGM), while the older section of the records become shifted, potentially indicating a mechanism linked to the gas-age chronology (given that the $\delta D$ filtered record appears to lag $\delta O_2/N_2$). The analysis from Dome F and South Pole has been included in the Supplement, and this is briefly mentioned in the discussion. Millennial-scale $\delta O_2/N_2$ variability at South Pole poses further questions about the mechanisms at high accumulation sites. However, addressing these questions is beyond the scope of this study and they will instead be considered in future work.

Lines 34-41 (Supplement): *"Figure S4 shows the data from South Pole. While the data only covers ~35 kyr, the high-resolution measurements reveal interesting signals which appear to be present in both the $\delta O_2/N_2$ and $\delta D$ records. The $\delta O_2/N_2$ and $\delta D$ filtered curves appear to resemble each other to varying degrees throughout the record, with the younger section (18-25 ka BP) varying largely in-phase, while the older section appears to be anti-phased, and soften with $\delta D$ lagging $\delta O_2/N_2$ (e.g., 46-52 ka BP). The in-phase coherence during the younger period also corresponds to relatively low $\delta D$, and hence, accumulation rate. Indeed, during the last glacial maximum at South Pole the accumulation rate was closer to that of EDC today (~3 cm w.eq yr$^{-1}$; Kahle et al., 2020). $\delta D$ is higher during the older period, potentially suggesting that a different mechanism is dominant under high accumulation conditions."*

[Figure]

***Figure S4.*** *Time series of δO₂/N₂, δD, and SSI on the SP19 ice-age chronology from South Pole (Winski et al., 2019). Panel (a) presents the 100-year interpolated δO₂/N₂ overlain with the low pass filtered curve (blue). The right y-axis in panel (a) shown SSI at 90°S (grey). Panel (b) shows the 100-year interpolated δD overlain with the low pass filtered curve (green). The bottom two panels present the residuals of the low pass filtered curves for δO₂/N₂ (c) and δD (d).*

**Specific comments:**

Title: I suggest changing the title to be more specific. This paper focuses mainly on inland Antarctica (Dome C), although the Greenland records are used, and discusses the relationship between $O_2/N_2$ and the surface environments. Thus, I suggest including the idea of "the relationship between $O_2/N_2$ and the ice sheet surface environment" and "inland Antarctica" in the title.

We agree and the title has been changed accordingly. However, while we acknowledge that the snowpack modelling and temporal analysis focusses on Dome C, our compilation includes several Antarctic coastal sites. The title has therefore been changed to:

*"On the relationship between δO₂/N₂ variability and ice sheet surface conditions in Antarctica."*

Line 24: Please consider adding two more references: Lipenkov et al. (2011) for the Vostok chronology and Oyabu et al. (2022) for the DF chronology. Both used $O_2/N_2$ for orbital dating.

The suggested references have now been included.

Around line 60: Please consider adding a hypothesis by Kawamura et al. (2007) that the absence of a climatic signal may result from the cancellation of temperature and accumulation effects on $O_2/N_2$.

The hypothesis by Kawamura et al. (2007) has now been included.

Line 57-60*: "There is a growing body of evidence for a local climatic imprint on ice core δO₂/N₂ records. Firstly, spectral analysis has revealed climate related 100-ka cyclicity at EPICA Dome C (Bazin et al., 2016). However, this 100 ka-cyclicity is not apparent in the Dome F δO₂/N₂ record, which*

*Kawamura et al. (2007) attribute to the idea that temperature and accumulation effects cancel each other out."*

Line 100: I suggest also providing a brief explanation of the methodology at Scripps in a similar manner as that for LSCE.

The melt-refreeze technique at Scripps is very similar to that used at LSCE. We have modified the text in Section 2.2 as follows:

Lines 88-96: "*The previously unpublished $\delta O_2/N_2$ datasets were measured at the Laboratoire des Sciences du Climat et de l'Environnement (LSCE), with the addition of some GISP2 data measured at Scripps Institution of Oceanography (Scripps). At both LSCE and Scripps, gases are extracted from the ice using a melt-refreeze technique based on the method described by Sowers et al. (1989) with modifications to the LSCE method as described by Landais et al. (2003), and to the Scripps method as outlined in Petrenko et al. (2006). In short, ice samples are placed into glass flasks at -20◦C while the atmospheric air is evacuated. Samples are then left to slowly melt to release the trapped gases, before being refrozen. The extracted air samples are then passed through a $CO_2$ and water vapour trap, before being trapped in the stainless-steel dip-tube submerged in liquid helium. An alternative semi-automated extraction technique is now more commonly used at LSCE which removes the need for refreezing of the samples (Bazin et al., 2016; Bouchet et al., 2023).*"

Line 103: "An over view of ..." the same information is written in Section 2.1.

Many thanks for pointing out this repetition. The reference to Tables S1 and S2 in Section 2.2 has been removed.

Section 2.2.1: It is not clear which samples were measured at LSCE and which at Scripps. If only the GISP2 samples were measured at Scripps, it would be better to switch the order and write it at the end.

Only GISP2 were measured at Scripps and the description for these measurements have been moved to the end of Section 2.2.1.

"measured on the 10-collector Thermo Delta V Plus" is not necessary for all samples. This information should be shown at once in the first paragraph of Section 2.2.

The following sentences have been added at the start of Section 2.2:

Lines 98-100: *"Samples from the following sites were measured at LSCE, France on a 10-collector Thermo Delta V Plus, unless otherwise stated. In all cases, the values used in this study are the average of at least two replicate measurements and have an average analytical uncertainty of 0.5 ‰ for $\delta O_2/N_2$."*

Line 163-165: The brittle zone is a zone of poor ice-core quality and not necessarily consistent with the bubble-clathrate hydrate transition zone (BCTZ) (Neff, 2014). Thus, I suggest using "BCTZ" (or a similar term) as in the supplement text. If you did not find the BCTZ information in some cores and thus employed the brittle zone of Neff (2014), you should write it in the text. Also, "the air in the gas phase has a very different composition to that in the clathrate hydrates" should be "the air in the **bubble** has a very different composition to that in the clathrate hydrates" (e.g., Ikeda-Fukazawa et al., 2001).

Thank you for pointing this out. Information in Table 1 (and corresponding text) now uses BCTZ depth range instead of the more general brittle zone. The majority of measurements within the published 'brittle zones' were also the BCTZ, and those that differ (intermediate depth ice cores from Berkner

Island, Siple Dome, Roosevelt Island, and Law Dome) have been updated in the text and for the data rejection. We have also updated the sentence.

Lines 157-162: "*Several measurements were performed on ice within the bubble-clathrate transition zone (BCTZ: where the high hydrostatic pressure in the bubbles cause entrapped gases to form clathrate hydrates (Schaefer et al., 2011)). At these depths, elemental fractionation occurs due to some gas species being preferentially incorporated into the clathrate structures (Ikeda-Fukazawa et al., 2001), thus making the interpretation of gas measurements unreliable (Bender, 2002). While brittle ice is reported at intermediate depth sites, such as BI, JRI, DE08, DSSW20k, RICE, and SD, we do not consider these BCTZs due to the absence of clathrate hydrates (Neff, 2014; Rubino et al., 2019).*"

Line 166: $O_2/N_2$ is not always increased in the BCTZ (e.g., Oyabu et al., 2021).

Many thanks for highlighting this. The sentence has been corrected in the revised manuscript.

Lines 162-164*: "Measurements from the BCTZ may either have increased mean $\delta O_2/N_2$ (usually in excess of 0 ‰) or strong data scattering, expressed as a high standard deviation (Oyabu et al., 2021). To avoid adding biases to our analysis, measurements from BCTZ are removed.*"

Lines 178-183: Regarding the data rejection criteria, it is unclear what was rejected with and without gas loss correction, and what was employed with and without gas loss correction. It seems that some of the data you utilized was affected by the gas loss but was included in the analysis because the gas loss correction worked. Also, the data with correction should be written as "corrected" since it is not unaffected                                   by                              gas                                        loss.

I would like to see a figure that displays all $O_2/N_2$ data, including rejected and employed data with and without gas loss corrections, with different colors or symbols.

Thank you for bringing this to our attention. There was no attempt to apply a gas loss correction given the uncertainty of storage history for all cores, as mentioned in Section 2.2.2. Consecutive measurements over a ~15-year period at LSCE revealed that bubbly ice is largely unaffected by gas loss during storage while clathrate ice becomes strongly depleted in $O_2$ (Figs S1 and S2; Bouchet et al., 2023). Measurements from bubbly ice are therefore not rejected based on the storage time. We do acknowledge that previous studies have suggested that $\delta O_2/N_2$ in bubbly ice is more susceptible gas loss (Yan et al., 2023) which is clearly observable in the Vostok core, and hence bubble ice measurements from Vostok are excluded here. However, this was not observed at EDC, GISP2, and TALDICE.

Due to the vastly different depth ranges for all the sites, we include the figure below to show the $\delta O_2/N_2$ data from each site before and after data rejection. The data presented in panel (b) below does not necessarily correspond to the data used in Fig. 3 in the manuscript given that we consider the time periods which overlap with temperature and accumulation rate reconstructions for the analysis.

[Figure]

*Figure S1. Boxplots showing each dataset before and after removing the scattered data from the brittle zone and the gas loss affected data. Panel (a) shows the entire dataset, while panel (b) shows the remaining data after applying the data rejection criteria. Individual points overlying the box plots provide an illustration of the number and distribution of individual data points from each site.*

Lines 279-280: I didn't understand what the authors meant.

This sentence refers to the fact that $\delta O_2/N_2$ data from the South Pole ice core are higher than EDC and Dome F. Higher accumulation rate and temperature at South Pole may explain the shift, supported by the positive correlation between $\delta O_2/N_2$ and both temperature and accumulation rate (Fig. 3 in revised paper). The following sentence is used instead to improve clarity.

*Lines 264-266: "$\delta O_2/N_2$ values from South Pole are higher than for EDC and Dome F for the same SSI, suggesting additional factors are influencing the records, such as accumulation rate, which at South Pole is over double that of both EDC and Dome F."*

Line 288: "1000 year averaged SSI". Did you average for the last 1000 years?

The values presented in the original paper are the average SSI over the last 1000 years for the latitude of each site. However, we realise that it would be more robust to compare to the mean SSI over the same time period as the $\delta O_2/N_2$ data. Please see response to comment after next for the updates relating to mean SSI for each site.

Line 292: "bias toward cold, low-accumulation conditions" is unclear. Does it mean that A and T are averaged between the LGM and the present and do not include past interglacials that were warmer than the present?

Apologies for the confusion. This statement was in reference to the averaging of present day and glacial maximum temperature and accumulation rates from the two sites, James Ross Island and Berkner Island. For the majority of sites, we ensure that we have $\delta O_2/N_2$ data, accumulation rate, and temperature reconstructions overlapping the same time period. We then take the mean values of each parameter during this overlapping period. However, some sites have no published accumulation rate and temperature reconstructions and therefore we chose to take either the present-day values (e.e., for Law Dome sites given the young ages covering a limited time frame), or the mean value between present day and last glacial maximum (in the case of Berkner Island and James Ross Island). The bias we mentioned was due to a large proportion of the $\delta O_2/N_2$ data from James Ross Island falling within the Holocene (while measurements from Berkner Island span the LGM to early/mid Holocene). However,

we were able to find temperature reconstructions from the James Ross Island ice core overlapping the time period of the $\delta O_2/N_2$ data, hence removing this potential bias.

Lines 301-302: I am curious whether this regression would be better if you used SSI for the same time period as the $O_2/N_2$ data. Have you confirmed before?

The regression in Fig. 3a is not improved when comparing to SSI over the same time period as the $\delta O_2/N_2$ data. In any case, we now use the mean SSI values over the same time period as the data for continuity of comparison between variables (where age scales are available). Scatter plots showing the original and new $\delta O_2/N_2$ vs SSI values are included below.

[Figure]

**Figure C.** Scatterplots showing the dependence of $\delta O_2/N_2$ on a) the average SSI for each site over the last 100 years, (as in the original manuscript), and b) the mean SSI at each site's present latitude over the time period of the $\delta O_2/N_2$ measurements. Note that any differences in mean $\delta O_2/N_2$ comes from adjustments in the brittle zone definition, and colours differ from Figure S1 in these responses because the sites in this figure are sorted by accumulation rate.

The following text has been included for the description of Fig. 3 in the manuscript:

Lines 270-278: "*In addition to the $\delta O_2/N_2$ datasets, we compile SSI, accumulation rate, and temperature reconstructions from each site covering the same age range as the $\delta O_2/N_2$ data (where available). The data are presented in Fig. 3 for (a) SSI, (b) accumulation rate, and (c) temperature. The range of accumulation rates and temperatures over the given time period are included as error bars in Fig. 3 to indicate the climate histories for each site. We do not include the SSI ranges at each site given that they are substantially larger than the range in mean SSI between sites. Due to limited data availability from the Berkner Island ice core, we take the average temperature and accumulation rate using present day values and last glacial maximum values (Capron et al., 2013). This is justified given that the data spans approximately the last glacial maximum to the start of the Holocene (Massam, 2018). Present-day values are used for accumulation rate and temperature at DE08 and DSSW20k due to the relatively young ages corresponding to the $\delta O_2/N_2$ data (~90 a BP).*"

Lines 315-317: I don't think these descriptions are necessary here. It is obvious that this paper does not use $O_2/N_2$ for orbital dating.

We have removed these sentences and replaced them with the description of the filtering techniques (included in response to general comments) used for the temporal analysis of the EDC records (as well as Dome F and South Pole in the supplement).

Line 317: If all EDC $O_2/N_2$ data were published in Bouchet et al. (2023), I think it would no longer be new in this paper.

The sentence has been modified accordingly.

Lines 296-298: *"The following analysis uses the longest period of relatively high-resolution $\delta O_2/N_2$ measurements from the EDC ice core between 190-259 ka BP (1980-2350 m) covering MIS 7 (Bouchet et al., 2023)."*

Section 2.3: What is the depth range or thickness of each layer considered in this model? Also, the authors mention that the maximum number of layers was increased from 50 to 80; was this a new modification made specifically for this study?

Snow layer thickness in Crocus dynamically evolves through time with densification. In the version of the model used here, the layer thickness ranges from 2 mm at the surface (Libois et al., 2014) to over a metre in some circumstances. The range is mentioned in Section 2.3.1, but we now explicitly state the minimum and maximum in our simulations instead of the current "... millimetres to metres…".

Lines 189-190: *"Briefly, the initial number of layers is defined by the user, with the thickness of each layer allowed to change along the simulation (layer thickness ranging from 2 mm at the surface to metres thick)."*

For our Dome C simulations where the accumulation rate is very low and simulations are run with a 100-year spin up, the thickest layers are likely to be at base of the snowpack owing to the aggregation of layers when the snowpack is already comprised of 80 layers. We chose to increase the number of layers to 80 to maximise resolution with depth compared to the 50 layers ascribed in Libois et al. (2014).

Lines 190-192: *"The maximum number of layers available in the model was increased from 50 (Libois et al., 2014) to 80 (this study) to maximise the resolution with depth owing to the higher number of thin layers forming at Dome C than at Alpine sites."*

Line 323:325: As I pointed out above, this long-term variability may just reflect the phase difference between the AICC2012 chronology and insolation.

Please see the response to general comments above.

Lines 342-343, Figure 5a and 5b: What factors contributed to the increased SSA for the top 10 cm and the subsequent dips in both SSA and density around the 10 cm depth in the model results?

There are various possible explanations for the high simulated SSA in the top 10 cm. Three possibilities are included here, and we touch on these possibilities in the revised manuscript.
1) One possibility relates to the 3∘C bias in snow temperature in the model compared to observations (Fig. 5d), resulting in less grain growth in the upper snowpack.
2) Alternatively, the high values may be linked to large precipitation events adding new snow with high SSA. During large precipitation events, the fresh snow will be buried rapidly and undergo less metamorphism than the topmost snow given that most metamorphism occurs in the top 5 cm where solar radiation drives strong temperature gradients (Picard et al., 2012). This could result in sustained high SSA in the top 10 cm.
3) Finally, the snowpack was initialised using a 100-year spin up where we ran the forcing file 10 times between 2000 and 2010, followed by the period from 2000 to 2020. The distinct snow properties in the top 10 cm may potentially relate to a transition from repeated atmospheric forcing file.

The following text has been added to Section 3.2.1:

Lines 334-339: *"The overestimation of simulated SSA in the top 10 cm may be linked to the cold bias in modelled snow temperature during summer via the reduced rate of snow metamorphism in near-surface snow. Alternatively, high SSA may be the result of a recent precipitation event. During large*

*precipitation events the fresh snow will be buried rapidly and undergo less metamorphism than the topmost snow given that most metamorphism occurs in the top 5 cm where solar radiation drives strong temperature gradients (Picard et al., 2012). This could result in sustained high SSA in the top 10 cm.*"

Lines 371-373: "These opposing influences of accumulation rate and temperature on snow properties at first appears to contradict the observation in Figure 3" I agree that the opposing influences appear to contradict, while this is consistent with the hypothesis by Kawamura et al. (2007) (cancellation of temperature and accumulation effects on $O_2/N_2$). How about mentioning this consistency?

This is a useful suggestion from the reviewer, and the consistency with the hypothesis of Kawamura et al. (2007) is indeed important to mention. In order to keep the interpretations out of the results section as much as possible, we have included this hypothesis later in the discussion. We also point out that here we are referring to the dominant effect of accumulation rate over temperature on $\delta O_2/N_2$ between sites. In this case, it appears as though accumulation and temperature effects are not muted by one another, but that accumulation rate is dominant.

The following text has been included in Section 4.3.1:

Lines 461-467: *"Identifying the mechanisms driving the accumulation rate dependence of $\delta O_2/N_2$ is complicated by the fact that, at polar sites, accumulation rate tends to covary with temperature (Fig. 8). Thus, an increase in grain size with increased near-surface residence time (decreased accumulation rate) will be countered by reduced snow metamorphism rates with a decrease in temperature (Fig. 6). These counteracting effects were invoked by Kawamura et al. (2007) to explain the absence of a 100-ka periodicity in $\delta O_2/N_2$ records at Dome Fuji. However, the stronger correlation between $\delta O_2/N_2$ and accumulation rate than temperature in Fig. 3 suggests a dominant role of accumulation. Therefore, we propose that a grain size mechanism – like that of SSI – may explain the positive correlation between accumulation rate and $\delta O_2/N_2$ in the EDC core (Fig. 3b)."*

Line 373-375: I didn't understand the sentence. "most evident is the sensitivity of grain size to accumulation rate" Why can you say that?

Figure 6c shows that the mean grain size over the top 20 cm (and deeper) is more sensitive to decreased accumulation rate (-10% in grain size compared to the reference simulation) than to increased (31%). This sentence aims to highlight this non-linearity.

Lines 357-361: *"Mean density and grain size respond non-linearly to perturbations in all forcing parameters–clearly documented by the magnitude of increase in grain size from decreased accumulation rate being three times greater than the decrease induced by an increase in accumulation rate. This is in line with the dependence of $\delta O_2/N_2$ to the logarithm of accumulation rate documented in Fig. 3."*

Line 386: It is not clear why the decrease in mean density and increase in density variability with A max and T min is "surprising". Some more explanation is needed.

We agree that surprising is not the correct word to use here and it has been removed to avoid confusion.

Line 389: I suggest inserting "increase in" or similar words between "with" and "SSI".

The sentence has been updated and now reads:

Lines 373-374: *"$\sigma_{gs}$ appears to increase with an increase in SSI and accumulation rate throughout the top metre."*

Lines 421-423: "The observations from…." The sentence is unclear. Please clarify.

The intention of this sentence is to highlight that the dominant mechanisms driving $\delta O_2/N_2$ variability in Antarctica may differ from those in Greenland due to their different climates. Indeed, Kobashi et al. (2015) show a multi-decadal scale variability in $\delta Ar/N_2$ from GISP2 which is anti-correlated with accumulation rate – opposite what we observe on millennial timescales at Dome C between $\delta O_2/N_2$ and $\delta D$. In parallel, total air content studies have shown an anti-correlation between TAC and accumulation rate in Greenland (Eicher et al., 2016) but a positive correlation in Antarctica (Epifanio et al., 2023). Section 4.2 has been modified for clarification as follows:

Lines 425-435: *"Transient effects in the firn column in response to rapid climatic changes (linked to overburden pressure) are expected to be largely absent from Antarctic sites due to reduced magnitude in climate variability compared to Greenland. However, an accumulation-dependent grain size mechanism may explain the positive correlation between $\delta D$ and $\delta O_2/N_2$ at EDC. Epifanio et al. (2023) proposed that the contradictory behaviour of TAC between NGRIP and South Pole may be explained by varying responses of the firn to changes in accumulation rate for sites with different surface climatic conditions. They suggest that a grain size mechanism dominates TAC modulation at low accumulation sites while transient effects from rapid climatic changes are more important at warm, high accumulation sites. We expect that the mechanisms driving $\delta O_2/N_2$ are also modulated by accumulation rate. Indeed, observations of $\delta O_2/N_2$ records from South Pole appear to support the in-phase coherence between $\delta O_2/N_2$ and $\delta D$ in the EDC core, but only when $\delta D$ is sufficiently low (Fig. S5 in Supplement). A shifted, anti-phase relationship is apparent when $\delta D$ is higher, which suggests that different mechanisms may be dominant under high accumulation conditions."*

Lines 423-425: This section is for discussion and I don't think this statement fits here.

We agree and have removed this sentence from the updated manuscript.

Lines 469-470: I would suggest to delete the sentence "Our findings…" and move the contents of lines 470-472 to the last paragraph of section 4.3.1. If you would keep the sentence here, more words are needed (it is unclear what "our findings" and "a density-dependent grain size mechanism" refer to).

Please see response to the comment after next which outlines the modifications to Section 4.3.

Lines 473-474: Not necessary. This sentence is a repetition of the sentence in lines 468-469.

The repetition has been removed from the updated manuscript.

Lines 489-492: Hard to understand. "leading to bulk ice with decreased $\delta O_2/N_2$", but the sentence before this phrase alone does not yet clarify the causal relationship. Your analysis of the ice core data shows that $O_2/N_2$ decreases as SSI increases (anti-correlation), and your model results show that grain size decreases as SSI increases. This would mean that there should be an anti-correlation between grain size and $O_2/N_2$. In addition, there seems to be a lack of explanation why/how the increased grain size depletes $O_2/N_2$. You may consider adding discussion, drawing arguments from previous studies as described in the Introduction section, to explain why a decreased (increased) $O_2/N_2$ is associated with a larger (smaller) grain size. One idea may be bring the discussion of Calonne et al. (2022) and Gregory et al. (2014), which appeared in lines 470-473, to here.

Many thanks for this useful comment. Actually, our model results show that grain size increases as SSI increases, thus, a positive correlation between grain size and $\delta O_2/N_2$ would be expected. This aligns with previous studies (e.g. Bender, 2002). To improve clarity on this discussion section we have restructured Section 4.3.1, and we also propose to include a schematic of the interactions between forcing parameters and both surface properties and $\delta O_2/N_2$, as observed in our study, as well as hypotheses and understanding from previous studies (Fig. 9 below). We acknowledge that the schematic

is an oversimplification and, as such, leave it to the Reviewer and Editor to decide whether it be included. Below is a paragraph from Section 4.3.1 in the revised manuscript, but the Reviewer is referred to Section 4.3.1 for the full updated text.

Lines 453-460: "*It is widely accepted that permeation is the process by which small molecules escape during pore closure (Fig. 9 [1]; e.g., Ikeda-Fukazawa et al., 2004; Huber et al., 2006; Severinghaus and Battle, 2006). Gregory et al. (2014) found that permeability is increased in high-density, large-grained firn due to a less complex pore structure. Indeed, Calonne et al. (2022) showed that grain size has a strong influence on permeability, such that for a given density, permeability is increased with grain size (Fig. 9 [2]). Our results show that high SSI and low accumulation rate are associated with larger grain size and lower δO₂/N₂. This observation supports the hypothesis that increased SSI (decreased accumulation rate) ultimately leads to lower δO₂/N₂ due to the increased permeability associated with increased grain size (e.g., Severinghaus and Battle, 2006; Fujita et al., 2009), and suggests a similar mechanism can explain the accumulation rate dependence.*"

[Figure]

***Figure 9**. Schematic to show interactions between surface forcings, physical properties, and close-off fractionation. Plus signs show a positive relationship (i.e., increase in SSI corresponds to increase in grain size), and minus signs indicate an inverse relationship. We use full lines to show results from our study, and dashed lines to show links based on previous studies, the numbers for which can be found in the text with associated references.*

Lines 498: What is the link between your findings and residence time in the LIZ? The model results show only near the ice sheet surface, and there seems to be no discussion of how the results relate to the O₂/N₂ fractionation in the LIZ (deep firn).

Our modelling results do not reveal anything about the LIZ in themselves. This statement was in reference to the mechanism proposed by Severinghaus and Battle (2006), but it is true that our results do not provide a direct link to the LIZ residence time. We instead draw on previous studies to infer links to LIZ characteristics, as is outlined below (from Section 4.3.1):

Lines 468-476: "*Previous studies propose that firn column characteristics will also modulate δO₂/N₂ via changes in accumulation rate and temperature (Severinghaus and Battle, 2006; Bazin et al., 2016). One mechanism links to the time for pores to close off. At low accumulation sites, pores will take longer to fully close, and thus, experience more elemental fractionation (Severinghaus and Battle, 2006). Additional hypotheses point to LIZ thickness and pore space geometry. Low accumulation sites tend to have thinner LIZs (Fig. 9 [3]; e.g., Landais et al., 2006; Witrant et al., 2012) and less tortuous pore structure in deep firn, characterised by large, rounded grains (Gregory et al., 2014). Such*

*characters are associated with increased gas diffusivity that would facilitate the removal of O₂-enriched gas back to the atmosphere (Fig. 9 [4]). In contrast, at high accumulation sites, the associated decrease in gas diffusivity in the LIZ would lead to a build of a stagnant air enriched in O₂ resulting in the trapping of gas with relatively high $\delta O_2/N_2$ (Fig. 9 [5]).*"

Lines 481-482: "*The aforementioned mechanisms relating to firn column characteristics indicate a positive correlation between $\delta O_2/N_2$ and accumulation rate. This is consistent with Fig. 3b but cannot be directly supported by our sensitivity results.*"

Lines 519-520: What does "the variability– and bulk mean – differences" refer to? I didn't understand what the authors meant.

"bulk mean" here should be in brackets as we are saying that both the variability, and mean density and grain size values are sensitive to the ascribed accumulation rate and temperature values. This has been corrected in revised discussion.

Lines 556-567: Need to explain why elongated pores lead to a greater fractionation of O₂/N₂.

The following sentence has been added to the discussion Section 4.3.1:

Lines 477-480: "*Alternatively, Hutterli et al. (2009) proposed that varying near-surface temperature gradients under different SSI intensities modulate anisotropy of the snow, as was confirmed by Leinss et al. (2020), leading to vertically elongated pores under high SSI (or low accumulation) conditions. They argue that elongated pores facilitate vertical diffusivity in the LIZ, leading to greater fractionation of $\delta O_2/N_2$ (Hutterli et al., 2009), an alternative but complimentary mechanism to LIZ thickness.*"

Lines 583-585 (We argue that the…): I don't see this argument in Discussion. This is the conclusion section and not a good way to introduce a new argument. The argument should be addressed in the Discussion section.

Apologies for the lack of continuity. The cancellation of accumulation and temperature effects in $\delta O_2/N_2$ records has been included in discussion Section 4.3.1.

Table 3: A max of the Dome Fuji core seems to be too large (even larger than at EDML). The accumulation rate of the Dome Fuji core over the last 720 ka can be found at NOAA Paleo Data Search.

Thank you for pointing out this mistake. The maximum value has been corrected to 4.1 cm w.eq yr⁻¹ and the associated reference has been updated (Kawamura et al., 2017).

**Technical corrections:**

Line 18: "LID" only appears here, but "LIZ" appears without abbreviation (e.g., line 69).

Thank you for pointing this out. The abbreviations have been corrected and the definition of LIZ has been included as follows:

Lines 17-20: "*The lock-in depth and the depth at which all pores are closed (close-off depth) are largely determined by local accumulation rate, temperature, and possibly the degree of density layering (Schwander et al., 1997; Martinerie et al., 1994; Mitchell et al., 2015). The region between the lock-in depth and close-off depth in known as the lock-in zone (LIZ).*"

Line 41: "Tomoko Ikeda-Fukazawa and Hondoh, 2004" is "Ikeda-Fukazawa et al., 2004".

The citation has now been corrected in the revised manuscript.

Line 154: Add "slope" after "Chemical"

This has been updated in the text.

Line 277 and 3rd line of the Fig. 2 caption: ‰.m$^2$.W$^{-1}$ Remove periods.

The units have been corrected in the updated version.

Line 297: Figure 2b may be 3b.

Yes, apologies, this has been corrected.

Line 300: panels (a) and (b) may be panels (b) and (c).

Many thanks for this pointing this out. We have corrected this is the text.

Line 301: Large residuals in "Figure 2a" should be in "Figure 3a". I suggest replacing "residual" with another term, such as deviation from the regression line.

The figure reference has been updated. We also include a clear definition of $\delta O_2/N_2$-SSI residual to as follows:

Lines 299-301: *"First, we interpolate SSI onto the $\delta O_2/N_2$ ages and take the deviations from the $\delta O_2/N_2$-SSI linear regression to isolate the $\delta O_2/N_2$ variability not explained by SSI (hereafter, $\delta O_2/N_2$-SSI residual)."*

Line 314: EPICA Dome C is already shortened in Line 82.

EPICA Dome C has been replaced with "EDC".

Line 363: Figure 6c and 6d may be 6a and 6b.

This has been corrected in the revised text.

Line 388: 50cm. , Remove period and space.

Corrected in the revised version.

Line 417: (Suwa and Bender, 2008b) -> Suwa and Bender (2008b)

The citation formatting has been corrected.

Table 1: Brittle zone should be bubble-clathrate transition zone (BCTZ) or a similar term.

Changed to BCTZ (also throughout the text).

Figure 5 caption: Density (a) and SSA (b)

We thank you for pointing the out. The caption has been corrected.


The paper, led by Romilly Harris Stuart, aims to improve our process understanding of a key ice core parameter - the O2/N2 ratio - that is used to orbitally date Antarctic ice core records. Decades ago it was discovered that the O2/N2 ratio resembles the orbitally-controlled solar insolation at the drill site. Already at that time, it was speculated how the upper firn layer is modulated by the amount of sunlight during summer. To affect the archived O2/N2 ratio in the bubbles, firn surface properties need to travel through the firn column to influence gas-specific (size-dependent) gas loss processes during the pore closure at the bottom of the firn. Over the years, a large number of studies suggested ideas to explain the observed O2/N2 variations, but we still lack an overall process understanding. While insolation apparently contributes a large fraction of the measured O2/N2 ratio, local temperature and accumulation rate modulate the orbital signal and lead to noise and uncertainty in the orbital tuning. At this target, Harris Stuart et al. place their study, which consists two approaches. Their study contributes to an important question relevant to the readers of The Cryosphere. Mostly, the paper is written clearly and provides the right depth of information and the figures are well-crafted and provide a visual support for the text. Overall, I support the publication of this study after minor revisions.

Their first approach is to apply an existing snowpack model to see if and to what extent differences in the solar radiation lead to changes in firn properties that might explain the observed O2/N2 ratios. Since the snowpack model was originally designed for alpine firn, applying it to low accumulation sites in Antarctica sets limitations. The authors became well aware of several limitations of their model (1-dimension, no wind compaction, merging box design) and thus interpreted their results with care. I got the impression that they used the model as much as it was possible for this study and then realized that no further insight could be obtained with this setup and that the model would need a significant improvement to capture the situation of low accumulation sites.

Their second approach is data-based and it was certainly a large effort to collect and screen all available O2/N2 records. The screening and data evaluation of the different cores and measurement campaigns is an important step and it would be crucial to provide a figure or two to allow the reader to see and understand the underlying problems of that step. Since it likely took a long time to collect all the records it would be helpful for others and the next generations of scientists to have easy access to these data sets and their meta information. So please spend some hours (perhaps more realistically days) of your time to bring all these data sets to a public database (both the already published and the new data). The analyses done on these 14 selected ice core sites conclude that factors other than insolation (accumulation rate and local temperature) have a sizable effect on the observed O2/N2 records and set limits to the precision and accurate of orbital tuning. This is a valuable outcome, but I feel that - in an ideal world with more time and resources - more can be done to disentangle the interplay between accumulation rate and temperature. As for the length of the diffusive firn column (i.e. d15N-N2), it

might be the location on an accumulation vs temperature plot that determines if the firn column gets longer or shorter, or if the grain size within the first meter of firn increases or decreases. Since the temperature and accumulation rates are either known from present-day conditions or are output parameters of models (e.g. can be derived from delta age etc.), the team of this study might want to look a bit deeper into the interplay of temperature and accumulation rate in modulating O2/N2 ratios.

Many thanks for the general feedback and useful suggestions. A file containing all the published and unpublished $\delta O_2/N_2$ data, accumulation rate reconstructions, temperature reconstructions, and SSI values will be published along with the paper. Regarding the disentanglement of accumulation rate and temperature effects, we have included an additional figure in the discussion to show 1) the correlation between reconstructed accumulation rate and temperature from our compilation, and 2) the expected mean $\delta O_2/N_2$ values as a function of accumulation rate and temperature. The new figure is included in response to the reviewer's specific comments. Given the useful feedback from both reviewers, we have also reorganised the discussion to enable a deeper look into the interplay between temperature and accumulation rate, and to develop the links between the results from the snowpack sensitivity tests and the data compilation.

I also wondered if more process understanding can be gained from analyzing the O2/N2 data from firn air studies. At least there should be some O2/N2 data from some drill sites available. The authors mention several times that one modulating factor of the O2/N2 imprint in the archived air bubbles is the degree by which the O2-enriched air that was expelled by the closing pores is advected upwards or diluted in firn. In other words, the O2/N2 fractionation during pore closure is only seen if it happens in an open system, ie. if the O2-rich air is removed from that layer. See e.g. lines 433 – 438. Perhaps using a full firn model that allows the simulation of permeability in the deep firn could help here?

This is a very valuable suggestion and would indeed contribute to the mechanistic understanding. There are data available from various drill sites which have been used to understand elemental fractionation during pore closure, alongside $\delta O_2/N_2$ records from ice cores (e.g., Severinghaus and Battle, 2006). While we agree that analysing firn air data alongside ice core $\delta O_2/N_2$ records would be beneficial for the mechanistic understanding, it is outside the scope of this current study. The purpose here is to show the link between $\delta O_2/N_2$, and accumulation rate and temperature using the compiled data, and to identify how the surface forcing parameters modify snow properties. Future work will focus on combining firn air measurements with bubble ice measurements to better constrain the behaviour at distinct sites.

**Further suggestions and technical comments**:

Line 3: "trapped bubbles". I guess you want to say that the air in the bubble is sealed off from the open pore space; you can just say bubble since bubbles are closed anyway.

This has been updated in the revised manuscript.

Line 4: write "... N2 molecules in extracted ice core air relative to the **modern** atmosphere - " Line 6: write "...and show a new additional link..." delete: ", in addition to the influence of the summer solstice insolation"

This has been modified in the updated version.

Line 8: "... forcings modulate snow physical properties near the surface "

Corrected in the revised version.

Line 10: "**a** mechanism**s**.."

Corrected in the revised version.

Line 16: firn...unconsolidated snow? Firn is the consolidated snow

Thanks for pointing this out. It has been corrected in the revised manuscript.

Line 18: rewrite "become sealed off from the firn air to form bubbles within the ice.

This has been re-written as:

"*seal off to form bubbles within the ice.*"

Line 18: "lock-in depth (LID)" actually you never use LID throughout the paper while you often use lock-in zone.

Many thanks for spotting this. We have modified the text accordingly.

Lines 15-20*: "Atmospheric air moves through porous networks within the firn until a critical depth (known as the lock-in depth) where vertical diffusion effectively stops, and pores gradually become closed off from the atmosphere. The lock-in depth and the depth at which all pores are closed (close-off depth) are largely determined by local accumulation rate, temperature, and possibly the degree of density layering (Schwander et al., 1997; Martinerie et al., 1994; Mitchell et al., 2015). The region between the lock-in depth and close-off depth in known as the lock-in zone (LIZ)."*

Line 22: komma after sites?

Added to the revised version.

Line 21 to 28: perhaps restructure this a bit. Essentially you describe two different kinds of dating approaches. O2/N2 and TAC are due to local effects of the firn column, thus these parameters are highly site-specific. On the other hand, d18O of O2 is a globally mixed atmospheric gas parameter that is not site-specific, and all ice cores yield the same record. Thus it can be used to wiggle-match different records but also relate the record to a certain orbital parameter.

Would be good to mention these two different approaches. Ideally, you could mention that d18O2 is used to date the gas phase of the ice core while O2N2 is an ice age parameter.

This is indeed important to mention, and we have modified this paragraph as below:

Lines 21-31: "*Gas records from ice cores provide a vital dating tool, especially at low accumulation sites where other methods are unsuitable. One such tool is orbital dating, which enables certain gas records to be tuned to insolation curves directly calculated from astronomical variables (Laskar, et al., 2004). Ice core $\delta^{18}O_{atm}$ records ($\delta^{18}O$ of atmospheric $O_2$) are strongly correlated with precession (mid-June 65°N insolation) (Bender et al., 1994). $\delta^{18}O_{atm}$ provides a direct atmospheric signal which may ultimately be used to align different ice core records (Extier et al., 2018). The two alternative proxies for orbital dating are 1) total air content, which is anti-correlated with integrated summer insolation, i.e., the annual sum of daily insolation above a certain threshold (Raynaud et al., 2007), and 2) $\delta(O_2/N_2)$, which is anti-correlated with summer solstice insolation intensity (e.g., Kawamura et al., 2007; Bouchet et al., 2023). Unlike $\delta^{18}O_{atm}$, TAC and $\delta(O_2/N_2)$ reflect processes within the firn column making the records site specific. The term $\delta(O_2/N_2)$ – hereafter, simply $\delta O_2/N_2$ - describes the relative difference between the ratio of $O_2$ to $N_2$ molecules trapped within the ice and that of the standard atmosphere and is expressed in the delta notation commonly used for stable isotope ratios.*"

Line 28: delete "trapped within the ice" so it gets a bit more general

Corrected in the revised version.

Line 30/31: delete "vice versa"

This has been deleted.

Line 31: you could delete "numerous" as you already name quite a few sites...

Agreed, this has been corrected.

Line 34/35: you might rewrite this to convey that the modification due to insolation happens at the snow surface but the process that effectively alters the archived O2N2 ratio happens at the depth where the pores close off

We agree that this description can be improved, and we now include the text below to replace sentences starting line 35.

Lines 35-40: *"Over orbital timescales, $\delta O_2/N_2$ is in anti-phase with local SSI when drawn on the ice-age chronology, indicating that the firn properties controlling the $\delta O_2/N_2$ fractionation are set near the surface. The consensus is that temperature gradients are increased with high summer insolation leading to enhanced near surface snow metamorphism, thus increasing near-surface grain size which persists throughout the firnification process down to the close-off depth (Bender, 2002; Severinghaus and Battle, 2006; Suwa and Bender, 2008a; Fujita et al., 2009)."*

Line 37: (COD) is just used twice ...just write it out in both cases

COD has been changed to close-off depth in both instances.

Line 40: replace ; with :

Corrected in the revised version.

Line 41: Why cite also her first name Tomoko?

Apologies, there was a mistake in the BibTex file which has now been corrected.

Line 48: Why "They"? you refer to Bender (2002) so technically just Michael Bender although he acknowledges at the end of his paper that he profited a lot from the discussion with many giants in this field

Thanks for pointing this out. This was an oversight and has now been corrected.

Line 85: WAISD would be a new abbreviation, commonly used is WD or WAIS

The abbreviation has been changed to WAIS throughout the manuscript.

Line 102 Table 1 (and other tables): for better visibility please align numbers in columns on the right side, e.g. Table 3 in Petrenko et al. 2016 http://dx.doi.org/10.1016/j.gca.2016.01.004

The tables have been modified accordingly in the updated manuscript.

Line Table 1: If possible and available please also add other site characteristics to this table, e.g. close-off depth or ice age at close-off depth (delta age) they might be useful as well

Thank you for this suggestion. We have included present-day close-off depth for all sites in Table 1.

Line 157: "gas loss during coring", can you explain a bit more here?

We acknowledge that this is unclear. The intention was to refer to gas loss effects linked to micro-cracks and preferential loss of $N_2$ in the bubble phase in the BCTZ. Here we now only refer to gas loss effects during storage in reference to $O_2$ depletion through time. Fractionation effects relating to the BCTZ and microcracks are mentioned in S1 in the Supplement.

Lines 7-10 (Supplement): "*Instead, measurements on clathrate ice stored above -50°C for more than two years are rejected from our analysis. However, Bender (2002) observed characteristics of gas loss effected δO₂/N₂ values in bubble ice from Vostok which they attribute to selective loss of O₂ via microcracks in the ice following coring. This led to the removal of the majority of δO₂/N₂ data from Vostok.*"

Line 163: Note that the brittle zone does not always correspond to the BCTZ, while for most of the ice cores, this is the case. I guess some ice cores have a technically defined brittle zone while they do not have the conditions to form clathrates at a certain set of depth or temperature; thus, without this coexistence of clathrate and bubbles, there shouldn't be a strong fractionation. Perhaps elaborate shortly on that.

Line 165: O2/N2 measurements within the brittle (or BCTZ) ice are not per se unreliable; it requires a post-coring gas loss, so the fractionated air in the bubbles escapes and thus induces scattered results. Also, small sample sizes resolve individual layers of bubbes vs clathrates

Line 167: see above, does Berkner have a chlathrate zone? Perhaps this explains good data within the brittle zone.

(In response to the previous three comments combined) Indeed, Berkner does not appear to have a clathrate zone (Schaefer et al., 2011), so this may explain the good data. Information in Table 1 (and corresponding text) now uses BCTZ instead if brittle zone. The majority of measurements within the published 'brittle zones' were indeed also the BCTZ, with the exception of Berkner Island, Siple Dome, Roosevelt Island, and Law Dome, the intermediate depth cores (Neff, 2014). We include the following additional text to improve clarity.

Lines 157-164: "*Several measurements were performed on ice within the bubble-clathrate transition zone (BCTZ: where the high hydrostatic pressure in the bubbles cause entrapped gases to form clathrate hydrates (Schaefer et al., 2011)). At these depths, elemental fractionation occurs due to some gas species being preferentially incorporated into the clathrate structures (Ikeda-Fukazawa et al., 2001), thus making the interpretation of gas measurements unreliable (Bender, 2002). While brittle ice is reported at intermediate depth sites such as BI, JRI, DE08, DSSW20k, RICE, and SD, we do not consider these BCTZs due to the absence of clathrate hydrates (Neff, 2014; Rubino et al., 2019). Measurements from the BCTZ may either have increased mean δO₂/N₂ (usually in excess of 0 ‰) or strong data scattering, expressed as a high standard deviation (Oyabu et al., 2021). To avoid adding biases to our analysis, measurements from BCTZ are removed.*"

Line 176: post-coring gas loss to differentiate between the gas loss happening during pore closure in the ice sheet

To clarify that we refer to post-coring gas loss we include the following text:

Lines 151-152: *"Ice core storage histories need to be considered before interpreting the data to account for post-coring gas loss effects which disturb the signal (Section 2.2.2)."*

Line 214: you mention the black carbon content. How sensitive is the model to the black carbon? What about a similar effect of mineral dust during glacial times (OK, mineral dust is mostly light quartz but there are also darker particles...)

The model is sensitive to black carbon content, as shown by Libois et al. (2013). We did investigate the link between the black carbon loading and the surface snow temperature discrepancy (compared to measured temperature profiles) but found that the simulated temperature profile with depth was best represented using the ascribed value (3 ng g$^{-1}$; Libois et al., 2015). The radiative transfer scheme we used in Crocus, the Two-streAm Radiative TransfEr in Snow model (TARTES; Libois et al., 2013), also has an option to include dust content loading. We did not include dust as the effect of black carbon is much more potent than that of the dust particles (~50 time more absorptive: Warren et al., 2006). This is definitely a useful comment and would be important to consider for realistic simulations of glacial and interglacial conditions.

Line 277: are the dots after the permil and the m2 correct?

This mistake has been corrected.

Line 282: "integrated summer insolation": can say a few words on the difference between integrated summer insolation and SSI and why you use SSI?

$\delta O_2/N_2$ and total air content have slight differences in their spectral signature; $\delta O_2/N_2$ records are dominated by precession, while total air content records are dominated by obliquity (e.g., Lipenkov et al., 2011). Spectral analysis shows that integrated summer insolation (the annual sum of daily insolation above a certain threshold) is driven by obliquity, whereas SSI is driven by precession (e.g., Huybers, 2006). Therefore, when investigating the mechanisms driving $\delta O_2/N_2$ we use SSI. The following text has been included in the introduction, and we also reiterate these differences in the discussion Section 4.2.

Lines 25-28: *"The two alternative proxies for orbital dating are; 1) total air content, which is anti-correlated with integrated summer insolation, i.e., the annual sum of daily insolation above a certain threshold (Raynaud et al., 2007), and 2) $\delta(O_2/N_2)$, which is anti-correlated with summer solstice insolation intensity (e.g.,Kawamura et al., 2007; Bouchet et al., 2023)."*

Lines 407-413: *"Previous studies suggest an overlap between the drivers of $\delta O_2/N_2$ variability and those of total air content (TAC) variability (Fujita et al., 2009; Lipenkov et al., 2011). Ice core $\delta O_2/N_2$ and TAC records exhibit slight differences in their spectral signals, whereby $\delta O_2/N_2$ in dominated by precession (hence SSI pacing), whereas TAC is dominated by obliquity (hence the integrated summer insolation pacing). However, variability in both records is linked to local insolation via the modulation of near-surface snow properties, which ultimately influence pore closure processes. Fujita et al. (2009) hypothesised that the permeation mechanism driving fractionation of $\delta O_2/N_2$ can explain half the variation in TAC, with the rest being driven by effusion. Therefore, we draw on TAC studies to help interpret millennial-scale variability in $\delta O_2/N_2$."*

Line 290: Figure 2 caption: you don't need to say that Dome C is plotted in dark-blue and Dome F in mid-blue because you can indentify each panel with their name already. Please add a), b), c) as you do in Fig. 3

The labelling of Fig. 2 has been modified accordingly.

Line 290: Figure 2 caption: why do you use r2 here, while in Fig. 3a, you use r for the same type of plot? perhaps always use r (as r2 can be calculated from that)

Both figures now show r, p-values, and the slope.

Line 312: Figure 3: I very much like your colour scheme, but here, it would also help to provide more visual hints to distinguish between some sites, e.g. LD and BI have quite similar colours (same for NEEM and WAIS). You could additionally use squares and diamonds.

As suggested, the markers for records from Greenland have been changed to squares.

Line 328: Table 3: The 5 EDML samples (596 – 860) are from the brittle zone. Are there no other samples measured at EDML, why just in the brittle zone?

In general, the $\delta O_2/N_2$ data compiled for our study are a biproduct from measurements of $\delta^{15}N$ or $\delta^{18}O_{atm}$. Therefore, the depth range of measured data was not chosen to assess $\delta O_2/N_2$.

Line 331: Fig. 4: Would the residuals look different if it would be plotted on the AICC2023?

This is a very useful point and both records are now presented on the AICC2023 chronology. The results are unchanged when applying the same analysis as in the original version of the manuscript (please see Fig.A below). However, it was suggested by Reviewer #1 to instead remove the SSI signal by applying a high pass filter to the $\delta O_2/N_2$ and $\delta D$ records. This has, of course, removed any 20 kyr variability observed in the original analysis, thus, reducing the strength of correlation, but the coherence in millennial-scale peaks remain.

[Figure]

**Figure A**. Comparison between the evolution of $\delta O_2/N_2$, SSI, and $\delta D$ from EDC on the AICC2012 ice age chronology (faded colours: Bazin et al., 2013), and on the AICC2023 chronology (dark colours: Bouchet et al., 2023). Panel a) presents the $\delta O_2/N_2$ (blues) with SSI on the right y-axis (grey), b) $\delta D$ (green), and c) the $\delta O_2/N_2$-SSI residuals (orange), between 200 and 260 ka BP. Scatterplots show the correlation between the $\delta O_2/N_2$-SSI residuals and $\delta D$ using d) AICC2012 chronology, and e) the AICC2023 chronology.

Line Figure. 4 caption: the respectively structure always requires the reader to go to the end of the sentence while the classical way "Panel d shows the correlation" is often quicker to access

The caption for Fig. 4 has been changed accordingly in the revised manuscript.

Line 354 Figure 5d: since there is no overlap between Jan and Jul, you could put both distributions into a single panel

This has been modified in the updated manuscript.

Line 381 Figure 7: it is not easy to see the difference between the faded line and the max line, perhaps increase the thickness of the line or use dashed lines etc

The contrast is increased between the faded and max lines in updated figure.

Line 399: the long list of references affects a bit the readability ....not sure if you need all the references here in the discussion section, perhaps write e.g. and two refs

The list of references has been replaced with the two most recent publications.

Lines 383-384: "*The compilation of deep ice core $\delta O_2/N_2$ records in Fig. 2 reinforces the widely documented anti-correlation between SSI and $\delta O_2/N_2$ (e.g. Oyabu et al., 2022; Bouchet et al., 2023).*"

Line 406: Fig. 3c, I guess you mean Fig. 3a showing as well O2/N2 vs SSI while Fig. 3c shows temperature. Where is the slope for Fig. 3a to compare it with the slopes of Fig. 2?

Many thanks for this pointing this out. This has been corrected and the slopes of each panel in Fig. 3 have been added for comparison.

Line 438-440: I am not so sure if this argument holds that the O2/N2 signal would then be on the gas age scale. Still it happens in the lock-in zone due to a process that was imprinted originally at the surface.

We agree that the lock-in zone characteristics will be set near the surface. Here we meant to highlight that the residence time of a gas in the lock-in zone may be modified by changes at the surface such as due to overburden pressure, hence, containing an integrated signal over the firn column. However, we acknowledge that this is overlapping the transient effects described by Eicher et al. (2016) and is therefore a different mechanism. As mentioned in response to general comments, we have restructured the discussion which involved integrating Section 4.2 (a general review of mechanisms) into the interpretation of our results. As such, this mechanism is mentioned in Section 4.3.1 in the revised manuscript.

Lines 468-471: "*Previous studies propose that firn column characteristics, such as delta-age, will also modulate $\delta O_2/N_2$ via changes in accumulation rate and temperature (Severinghaus and Battle, 2006; Bazin et al., 2016). One mechanism links to the time for pores to close off. At low accumulation sites, pores will take longer to fully close, and thus, experience more elemental fractionation (Severinghaus and Battle, 2006).*"

Lines 481-482: "*The aforementioned mechanisms relating to firn column characteristics indicate a positive correlation between $\delta O_2/N_2$ and accumulation rate. This is consistent with Fig. 3b but cannot be directly supported by our sensitivity results.*"

Line 490: I am puzzled a bit about the term bulk ice...

The term "bulk ice" was used here to incorporate multiple ice layers with potentially different characteristics. Given the reworking of the discussion, this sentence has been removed.

Line 490: "The opposite – a decrease" not sure if this sentence describing the opposite effect is necessary I guess the sensitivity of grain size for a given density works in both directions

We agree, and this sentence has been removed for the revised manuscript.

Line 493: yes, temperature and accumulation rate do generally covary, but they are not super tightly correlated, and there are sites that are above or below the expected line for the temperature–accumulation relationship. Perhaps you can derive some useful information from the deviations from this temperature-accumulation relation, i.e. a site that has too little accumulation rate for a given temperature. A scatter plot showing all sites with their accumulation vs temperature might help to identify sites that deviate from others in Figure 3. This requires O2/N2 data for the present-day conditions for accumulation and surface temperature that are likely more accurate than the reconstructed values based on modelling via water isotopes.

Many thanks for this useful suggestion. Given that we currently do not have a lot of modern $\delta O_2/N_2$ data from ice cores, we instead include a new figure in the discussion showing accumulation rate-temperature relationship using the compiled datasets (Fig. 8). In addition, contour lines indicate the expected mean $\delta O_2/N_2$ values based on a multiple regression to predict mean $\delta O_2/N_2$ for each site as a function of accumulation rate and temperature. The figure highlights that mean $\delta O_2/N_2$ is particularly sensitive to accumulation rate, especially at low accumulation sites. In contrast, even rather major changes in temperature would have little influence on $\delta O_2/N_2$ at low accumulation sites. The following text has been added to the discussion Section 4.1 alongside the new Fig.8:

Lines 392-399: *"To further constrain the drivers of $\delta O_2/N_2$ variability, we construct a multiple regression to parameterize mean $\delta O_2/N_2$ as a function of accumulation rate and temperature (Fig. 8). Combined, accumulation rate and temperature can explain up to 70% of the total variance in mean $\delta O_2/N_2$. Adding SSI does not improve the adjusted $r^2$ of the multiple regression. Results are summarised in Fig. 8 and reiterate that $\delta O_2/N_2$ is highly sensitive to accumulation rate, especially at low accumulation sites. The site dependence of $\delta O_2/N_2$ we observe has been noted in previous studies. Indeed, Bazin et al. (2016) identified an offset in absolute $\delta O_2/N_2$ values in the Vostok core compared to Dome F and EDC, similar to our observation in Fig. 2. By including records from numerous ice cores, we can show that the absolute values are systematically linked to site accumulation rate and temperature."*

[Figure]

***Figure 8***. *Contour plot of $\delta O_2/N_2$ as a function of temperature and accumulation rate. Contours are calculated based on the multiple linear regression of $\delta O_2/N_2$ from accumulation rate (A) and temperature ($T_{air}$) and their coefficients ($\delta O_2/N_2 = 2.78 \log(A) + -0.03 T_{air} -13.76$). The accumulation rate-temperature regression (dashed, black line) is defined using a regression on temperature and accumulation rate for all sites in the data compilation. Annotated values state the mean $\delta O_2/N_2$ for each site.*

Line 500: this sentence is a bit unclear to me

The first sentences of Section 4.3.2 have been modified to the following:

Lines 487-488: "*Density stratification in deep firn has also been invoked to modulate $\delta O_2/N_2$ variability (Fujita et al., 2009). Results from the Crocus model are used to infer the sensitivity of near-surface density and grain size variability to perturbations in input forcing parameters.*"

Line 545: could you spend a few words on how the local SSI at EDC is linked to local accumulation rates since this is a larger-scale weather phenomenon and involves low-pressure systems entering the continent, etc.? Perhaps elaborate a bit on that?

Thank you for this comment. Our intention here was to reiterate the covariance between temperature and accumulation rate based on the Clausius-Clapeyron relationship. We do acknowledge that local temperature is not completely independent from SSI, and that they appear to have been in phase during the Early Pleistocene (Yan et al., 2023). However, we did not intend to suggest that local SSI is directly linked to local accumulation rate.

Line 575: "local climate (accumulation rate" I understand what you mean but accumulation rate might be largely determined by the circulation patterns in the Southern Ocean region.

To avoid confusion, we have modified the sentence as follows:

Lines 557-559: "*Analysis of both spatial (multi-site) and temporal (single site) variability in $\delta O_2/N_2$ presents new evidence of a dependence on accumulation rate and temperature, in addition to the well-documented SSI dependence.*"

Line 576: I guess this statement also holds for AICC2023 (although there seems to be a small circularity hidden into that because the age scale is constructed using the O2/N2 orbital tuning)

This statement is still true when using the new AICC2023 chronology. To avoid the circularity (and discrepancies in the chronology), we confirmed the coherence in millennial-scale variability by comparing $\delta O_2/N_2$ and $\delta D$ on the depth scale, shown below. The timings directly align giving us confidence in our results.

[Figure]

**Figure D**. $\delta O_2/N_2$ and $\delta D$ plotted on the depth scale from the EDC core (Bouchet et al., 2023; Landais et al., 2021).

Line 582: this sentence misses some words...support the idea...

This sentence now reads:

Lines 566-568: "*Our findings support the hypothesis that a grain size mechanism is the dominant driver of elemental fractionation at low accumulation sites, such that increased grain size for a given density facilitates $O_2$ expulsion via enhanced permeability.*"